# Learnability of Linear Thresholds
# from Label Proportions

**Rishi Saket**
Google Research, India
rishisaket@google.com

## Abstract

We study the problem of properly learning linear threshold functions (LTFs) in the learning from label proportions (LLP) framework. In this, the learning is on a collection of bags of feature-vectors with only the proportion of labels available for each bag.

First, we provide an algorithm that, given a collection of such bags each of size at most two whose label proportions are consistent with (i.e., the bags are satisfied by) an unknown LTF, efficiently produces an LTF that satisfies at least $(2/5)$-fraction of the bags. If all the bags are *non-monochromatic* (i.e., bags of size two with differently labeled feature-vectors) the algorithm satisfies at least $(1/2)$-fraction of them. For the special case of OR over the $d$-dimensional boolean vectors, we give an algorithm which computes an LTF achieving an additional $\Omega(1/d)$ in accuracy for the two cases.

Our main result provides evidence that these algorithmic bounds cannot be significantly improved, even for learning monotone ORs using LTFs. We prove that it is NP-hard, given a collection of non-monochromatic bags which are all satisfied by some monotone OR, to compute any function of constantly many LTFs that satisfies $(1/2 + \varepsilon)$-fraction of the bags for any constant $\varepsilon > 0$. This bound is tight for the non-monochromatic bags case.

The above is in contrast to the usual supervised learning setup (i.e., unit-sized bags) in which LTFs are efficiently learnable to arbitrary accuracy using linear programming, and even a trivial algorithm (any LTF or its complement) achieves an accuracy of $1/2$. These techniques however, fail in the LLP setting. Indeed, we show that the LLP learning of LTFs (even for the special case of monotone ORs) using LTFs dramatically increases in complexity as soon as bags of size two are allowed. Our work gives the first inapproximability for LLP learning LTFs, and a strong complexity separation between LLP and traditional supervised learning.

## 1 Introduction

A *linear threshold function* (LTF) over the $d$-dimensional feature-vectors $\mathbf{x}$ is given by $\mathsf{pos}(g(\mathbf{x}))$ for some linear function $g(x_1, \ldots, x_d) = \sum_{i=1}^{d} c_i x_i + c_{d+1}$, where $\mathsf{pos}(z) := \mathbb{1}_{\{z>0\}}$. Also known as *linear classifiers* or *halfspaces*, LTFs are one of the most fundamental classes studied in computational learning and lie at the core of several machine learning algorithms such as Perceptron [25, 21] and SVM [7]. It is known [4] that LTFs are *properly* learnable in the supervised learning setup: given a training set of labeled feature-vectors which are consistent with some LTF, using polynomial-time linear programming one can efficiently find one such LTF.

In this work we initiate an investigation into the proper learnability of LTFs in the framework of *learning from label proportions* (LLP). In this, instead of labels for each feature-vector, the training data consists of *bags* (subsets) of feature-vectors along with the proportion of labels in each bag.

35th Conference on Neural Information Processing Systems (NeurIPS 2021).

Given a collection of bags with their respective label proportions which are consistent with (i.e, are satisfied by) some unknown feature-vector level classifier, the goal is to find a classifier (from the same concept class for proper learning) satisfying the most number of bags. Such a model was formalized by Yu et. al [30] who showed bounds on the generalization error for predicting the label proportion on unseen bags.

When all bags are unit-sized this becomes the usual supervised learning in which as we noted above LTFs are efficiently properly learnable. A natural question is whether this remains true even when larger bag sizes are allowed. In particular, we consider the simplest setting with bags of size at most two and study the following question: *Given a collection of bags, all of size at most two and guaranteed to be satisfied by some unknown* LTF*, what is the maximum fraction of bags that can be satisfied using an efficiently computable* LTF*?*

There naturally are two types of bags: *monochromatic* bags whose feature vectors have the same label, and the *non-monochromatic* ones whose feature-vectors have different labels and are necessarily of size two. Clearly, the monochromatic bags determine the labels of their constituent feature-vectors. Therefore, for a collection of bags consistent with some LTF, one can efficiently find an LTF using linear programming to satisfy all the monochromatic bags. However, such an algorithm is not guaranteed to satisfy *any* non-monochromatic bag.

In traditional supervised learning of LTFs, the trivial algorithms of taking the best of any LTF or its complement, or a random homogeneous LTF, both achieve an accuracy of half. Unfortunately, these algorithms provide no such guarantees in the LLP setting, even for bags of size at most two. At a high level, in LLP the objective is given by comparing the predicted labels of a bag's feature-vectors with each other, unlike supervised learning where they are only compared with observed ones. This seems to qualitatively change the nature of the problem in terms of the applicable techniques.

We first provide an algorithm that satisfies (in expectation) at least half of the non-monochromatic bags and one-fourths of the monochromatic ones. This can be combined with the linear programming algorithm which satisfies all monochromatic bags (and possibly none of the non-monochromatic ones) to obtain an algorithm which outputs an LTF satisfying $(2/5)$-fraction of the bags. We also show that for the special case of the feature vectors being $d$-dimensional boolean and the bags being consistent with an OR (boolean disjunction) the algorithmic guarantees improve by an $\Omega(1/d)$ additional factor.

## 1.1 Previous Work on LLP

The LLP problem naturally arises in many real world scenarios where the labels are not available individually but only as proportions for bags of feature-vectors due to privacy [28] and legal [26] reasons, high label supervision cost [6] or technical limitations of labeling mechanisms [9]. Some of the earlier works [8, 16, 22, 26] applied supervised learning methods such as MCMC, clustering, and linear classifiers to the LLP problem. The work of [24] assumed an exponential generative model and proved performance bounds under assumptions on the distribution of the bags, and was further generalized by [23] while the work of [29] proposed novel *proportional* SVM based algorithms. Subsequently, approaches based on deep neural nets for large-scale and multi-class data [18, 10, 19], as well as bag pre-processing techniques [27] have been developed for LLP.

In contrast to the long line above of application focused research in LLP, the theoretical study of algorithmic and complexity issues in LLP has been rather limited. On the complexity side, [13] studied a *one-bag* version of this problem where the observed global label proportion is given over all the feature-vectors and the goal is to find a classifier to minimize the deviation from this. They showed, among other results that common concept classes such as monotone-ORs cannot be learnt in their model to arbitrary accuracy.

The LLP model which we study – of predicting the bag label proportions – was first formalized in the work of [30] who also bounded the generalization error when taking the (bag, label-proportion)-pairs as instances sampled iid from some distribution. Their bound has the same dependence on the feature-vector classifier's VC-dimension as in supervised learning, with an additional logarithmic dependence on the bag size. While the loss function they consider is slightly different, their bound can be applied to minimizing the number of unsatisfied bags of size two (see Appendix E of the supplemental). In the rest of this section we define the problem of learning LTFs from label proportions, describe our results and the previous related work along with an informal description of our methods and proof techniques.

## 1.2 Problem Definition

Let $\mathcal{X} = \{\mathbf{x}_1, \ldots, \mathbf{x}_n\} \subseteq \mathbb{R}^d$ be the set of feature-vectors, and bags $\mathcal{B} = \{B_1, \ldots, B_m\} \subseteq \binom{\mathcal{X}}{1} \cup \binom{\mathcal{X}}{2}$, i.e. each bag is of size at most 2. The label set is binary i.e., $\{0,1\}$, and for each bag $B_k$ there is a proportion $\sigma_k$ which is the average of the labels of the vectors in the bag, satisfying $\sigma_k \in \{0,1\}$ if $|B_k| = 1$ and $\sigma_k \in \{0, 0.5, 1\}$ if $|B_k| = 2$. If $\sigma_k \neq 0.5$ then $B_k$ is said to be *monochromatic* i.e., bags which have same labels for their feature-vectors. The remaining bags $B_k$ necessarily of size 2 having $\sigma_k = 0.5$ are called *non-monochromatic*.

A bag $B_k \in \mathcal{B}$ is satisfied by some $f : \mathcal{X} \to \{0,1\}$ if $\left( \sum_{\mathbf{x} \in B_k} f(\mathbf{x}) \right) / |B_k| = \sigma_k$.

An instance of LLP-LTF is given $(\mathcal{X}, \mathcal{B} = \{B_k\}_{k=1}^m, \{\sigma_k\}_{k=1}^m)$. It is said to be *satisfiable* if there exists an LTF that satisfies all the bags. The goal is to find an LTF that satisfies the most bags.

We denote by LLP-OR the special case when $\mathcal{X}$ has boolean vectors and the guaranteed LTF for satisfiable instances is an OR. A further special case is when the guaranteed OR is monotone i.e., it has no negated variables.

## 1.3 Our Results

We provide the following algorithmic guarantees for LLP-LTF on bags of size at most 2.

**Theorem 1.1.** *There is a polynomial time algorithm which given a satisfiable instance of* LLP-LTF *on bags of size at most 2, computes an* LTF *that satisfies at least* $(2/5)$*-fraction of the bags. If all bags are non-monochromatic this satisfies at least half of the bags. For satisfiable* LLP-OR *on bags of size at most 2 with $d$-dimensional boolean feature-vectors, there is a polynomial-time algorithm computing an* LTF *satisfying at least* $(2/5 + \gamma_0/d)$*-fraction of the bags in general, and* $(1/2 + \gamma_0/d)$*-fraction if all bags are non-monochromatic, for some absolute constant $\gamma_0 > 0$.*

The above theorem for LLP-LTF is proved in Sec. 2 with the proof for LLP-OR provided in Sec. 2.1. The algorithms use random hyperplane rounding of vectors, so the above guarantees are in expectation. They can however be derandomized by the sophisticated procedure given in [20] and we provide an explanation in Appendix A of the supplemental.

Our result on the intractability of LLP-OR (an thereby of LLP-LTF) is stated below.

**Theorem 1.2.** *Given a satisfiable instance of* LLP-OR *consisting only of non-monochromatic bags each of size 2 s.t. there is a monotone* OR *that satisfies all bags, it is* NP-*hard to find any function of $\ell$* LTFs *that satisfies* $(1/2 + \delta)$*-fraction of the bags, for any constants $\ell \in \mathbb{Z}^+, \delta > 0$.*

The proof follows via a reduction from a variant of the Label Cover problem. We include the guarantees of the hardness reduction as Theorem 3.3 which directly implies Theorem 1.2. However, due to lack of space we defer the formal proof of Theorem 3.3 to Appendices C and D of the supplemental, while including an informal description in Sec. 1.5. We note that Theorem 1.2 shows the optimality of the approximation factor obtained in Theorem 1.1 for non-monochromatic bags.

**Previous Related Work.** As mentioned earlier, the proper supervised-learnability of LTFs [4] is well known without any distributional assumptions. In the presence of adverserial label noise however, the problem was shown to be NP-hard [17] and the works of [1, 2, 5] proved stronger inapproximability bounds with the optimal $(1/2 + \varepsilon)$-factor hardness obtained independently by [11] and [15], further generalized by [3] to apply for constant degree *polynomial thresholds* as hypotheses. Subsequently, [12] showed that this also holds for the special case of learning OR with an LTF which was recently strengthened by [14] to also rule out functions of constantly many LTFs as hypotheses.

## 1.4 Overview of Algorithmic Results

**Algorithm for LLP-LTF.** Given a satisfiable instance of LLP-LTF we can first assume by slightly translating the constant part of the unknown satisfying LTF that it classifies all the feature-vectors of the instance with a non-zero margin. Next, we observe that there are strict quadratic constraints (over the coefficients of the LTF) that distinguish the monochromatic from the non-monochromatic bags. Indeed, consider the unknown LTF $\mathsf{pos}(\langle \mathbf{c}_*, \mathbf{x} \rangle)$ (WLOG by appending a last coordinate with value 1 to all feature-vectors $\mathbf{x}$ of the instance) that satisfies all bags. Then, $\langle \mathbf{c}_*, \mathbf{x}_i \rangle \langle \mathbf{c}_*, \mathbf{x}_j \rangle$ is positive for monochromatic bags and negative for the non-monochromatic ones whose constituent vectors are $\mathbf{x}_i$

and $\mathbf{x}_j$. Taking $\mathbf{C} = \mathbf{c}_* \mathbf{c}_*^\mathsf{T}$ we obtain the same conditions for $\mathbf{x}_i^\mathsf{T} \mathbf{C} \mathbf{x}_j$, which are constraints linear in the entries of positive semi-definite (psd) $\mathbf{C}$, and can be efficiently solved for $\mathbf{C}$ using a semi-definite program (SDP). Using the psd decomposition $\mathbf{C} = \mathbf{L}^\mathsf{T} \mathbf{L}$ we map $\mathbf{x}_i$ to the vector $\mathbf{z}_i := \mathbf{L} \mathbf{x}_i$ so that $\langle \mathbf{z}_i, \mathbf{z}_j \rangle = \mathbf{x}_i^\mathsf{T} \mathbf{C} \mathbf{x}_j$. From here, the random hyperplane rounding of $\mathbf{z}_i$ to $\langle \mathbf{g}, \mathbf{z}_i \rangle$ ($\mathbf{g} \leftarrow N(0,1)^d$) yields an LTF $\mathsf{pos}(h(\mathbf{x}_i)) = \mathsf{pos}(\langle \mathbf{g}, \mathbf{z}_i \rangle)$ that separates (in expectation) the vectors of at least half of the non-monochromatic bags and at most half of the monochromatic ones. Taking the better of $\mathsf{pos}(h(\mathbf{x}))$ and $\mathsf{pos}(-h(\mathbf{x}))$ satisfies in expectation at least half of the non-monochromatic bags and at least one-fourths of the monochromatic ones. Combining this with the known linear programming algorithm that satisfies all the monochromatic bags we obtain the desired $(2/5)$-approximation.

**Algorithm** LLP-OR. Given a satisfying OR we can write an equivalent LTF with a margin property – the thresholded linear form has value either $(-1/2)$ if the OR evaluates to 0, and otherwise takes values in the range $[1/2, d]$. This allows us to add margin terms in the SDP constraints corresponding to the bags. The rest of the algorithm mapping $\mathbf{x}_i$ to $\mathbf{z}_i$ remains the same, while the strengthened constraints imply an extra $\Omega(1/d)$ in the probabilities that the random hyperplane rounding separates the vectors of non-monochromatic bags and does not separate those of monochromatic bags, yielding a corresponding improvement in the fraction of satisfied bags.

## 1.5 Overview of Hardness Result

For this exposition of the proof techniques we focus only on a weaker version of Theorem 1.2 for one LTF rather than functions of constantly many LTFs as hypothesis. The proof is obtained via a reduction from a variant of the (by now standard) Label Cover problem. This consists of a bi-regular bipartite graph $(V, U, E)$ where vertices in $V$ and $U$ have to be assigned labels from $[M]$ and $[m]$ respectively for some $M > m$. Each edge $e = (v, u) \in V \times U$ is a constraint with a projection $\pi_{v,u} : [M] \to [m]$ and is satisfied by the labels $l_v$ and $l_u$ for $v$ and $u$ respectively if $\pi_{v,u}(l_v) = l_u$. It is NP-hard to distinguish whether a Label Cover instance is completely satisfiable ("Yes" case) , or any labeling satisfies only an $o(1)$-fraction of edges ("No" case).

The goal of the reduction is to transform a Label Cover instance $\mathcal{L}$ to an LLP-OR instance of only non-monochromatic 2-sized bags s.t. (i) if $\mathcal{L}$ is a Yes instance there is an OR satisfying all the bags, (ii) if $\mathcal{L}$ is a No instance then there is no LTF satisfying more than $(1/2 + o(1))$-fraction of the bags.

The above template of reduction from Label Cover and its variants has been widely used to prove hardness results, including in computational learning by the previous works mentioned in this section. The main technical ingredient for such reductions is a *dictatorship test* tailored to transform a specific local collection of variables/constraints of $\mathcal{L}$ into a sub-instance of the target problem. This is done by defining a coordinate in the feature-space for each variable of $\mathcal{L}$ and its label, and then constructing the feature-vectors and bags of the sub-instances in this space - zeroing out coordinates corresponding to variables not in the local collection. Taking the union the sub-instances for all local collections yields the hard instance of the target problem, in our case LLP-OR.

Our dictatorship test needs to satisfy the following meta-properties, (i) *completeness*: any labeling satisfying the local collection of Label Cover constraints yields a good solution to the sub-instance i.e., an OR that satisfies all its bags, and (ii) *soundness*: any good enough solution to the sub-instances i.e., an LTF satisfying more than half the bags on average over the sub-instances, should be *independently decodable* into a labeling satisfying the local constraints. This notion of decodability is crucial, it depends on the fact (by design) that the good solution LTF will have coefficients on the disjoint sets of coordinates corresponding to each Label Cover variable. An independent decoding is any randomized procedure selecting a label for each variable considering *only* the LTF coefficients of the coordinates corresponding to that variable. In the next few paragraphs we motivate the design of our dictatorship test and provide an overview of its completeness and soundness analyses.

### 1.5.1 Dictatorship test construction

We fix as our local collection any two vertices $v, w \in V$ which are neighbors of some $u \in U$ in $\mathcal{L}$ with $\pi_{v,u}$ and $\pi_{w,u}$ being the respective projections. Making sure that completeness holds is usually straightforward. For the soundness we want to ensure that if the LLP-OR sub-instances have an LTF satisfying more $(1/2 + \delta)$-fraction of the bags (on an average over the local collections) then there should be independent decodings of $v$ and $w$ assigning labels $l_v$ and $l_w$ s.t. $\pi_{v,u}(l_v) = \pi_{w,u}(l_w)$ with significant probability depending on $\delta$. We call such $l_v$ and $l_w$ consistent labels. To begin, we

provide a first cut approach and illustrate its shortcomings to provide the insights for developing the actual dictatorship test used in our reduction.

**First Cut Approach.** For $v$ and $w$ we have a separate set of $[M]$ boolean coordinates denoted by the boolean variables $\{X_{v,i}\}_{i=1}^{M}$ and $\{X_{w,i}\}_{i=1}^{M}$. Let us construct a distribution that samples bags with two feature-vectors $\mathbf{X}^{(1)}$ and $\mathbf{X}^{(2)}$, the former supported only on $\{X_{v,i}\}_{i=1}^{M}$ and the latter other only on $\{X_{w,i}\}_{i=1}^{M}$, such that the OR formula $(X_{v,l_v} \vee X_{w,l_w})$ for any consistent labels $l_v$ and $l_w$ evaluates to 1 exactly on one of them. One way to do this is as follows. Sample a random set $J^v$ by choosing each $j \in [m]$ w.p. $1/2$ and let $J^w := [m] \setminus J^v$. Then, (i) for each $j \in J^v$ set $X_{v,i}^{(1)} = 1$ for all $i \in \pi_{v,u}^{-1}(j)$ and $X_{w,i}^{(2)} = 0$ for all $i \in \pi_{w,u}^{-1}(j)$, and (ii) for each $j \in J^w$ set $X_{v,i}^{(1)} = 0$ for all $i \in \pi_{v,u}^{-1}(j)$ and $X_{w,i}^{(2)} = 1$ for all $i \in \pi_{w,u}^{-1}(j)$; with the undefined coordinates in $\mathbf{X}^{(1)}$ and $\mathbf{X}^{(2)}$ set to 0. This distribution – call it $\mathcal{D}^0$ – then outputs a non-monochromatic bag $B$ containing $\mathbf{X}^{(1)}$ and $\mathbf{X}^{(2)}$. It is easy to verify that exactly one of $(X_{v,l_v}^{(1)} \vee X_{w,l_w}^{(1)}) = X_{v,l_v}^{(1)}$ and $(X_{v,l_v}^{(2)} \vee X_{w,l_w}^{(2)}) = X_{w,l_w}^{(2)}$ evaluate to 1 for any consistent labels $l_v$ and $l_w$. Therefore, this dictatorship test given by $\mathcal{D}^0$ satisfies the completeness property. Unfortunately however, it *does not* have the soundness property. To see this, suppose that the pre-images of the two projections are all of the same size $d$ i.e, $\left|\pi_{v,u}^{-1}(j)\right| = d$ and $\left|\pi_{v,u}^{-1}(j)\right| = d$ for all $j \in [m]$, implying $M = md$. Then, the linear threshold $\mathsf{pos}(h(\mathbf{X}))$ where $h(\mathbf{X}) := \sum_{i=1}^{m}(X_{v,i} + X_{w,i}) - M/2$, evaluates to 1 exactly on one of $\mathbf{X}^{(1)}$ and $\mathbf{X}^{(2)}$ w.p. 1 when $m$ is odd and w.p. $1 - O(1/\sqrt{m})$ if $m$ is even. Thus, $\mathsf{pos}(h)$ which is a majority-threshold is a good solution. However, $h$ has equal weight on each of its $\{X_{v,i}\}_{i=1}^{M}$ and $\{X_{w,i}\}_{i=1}^{M}$ coordinates, so any independent decoding of $v$ and $w$ using the corresponding coefficients of $h$ will yield consistent labels with probability only around $O(1/M)$, much lower than desired.

**Final Dictatorship Test.** The problem with $\mathcal{D}^0$ is that the total number of 1s in $\mathbf{X}^{(1)}$ and $\mathbf{X}^{(2)}$ together is exactly $M$ and therefore the majority threshold works. If the coordinates were real-valued this could be mitigated by choosing the non-zero coordinates from some range of values randomly, introducing enough variability in the sum to *fool* the majority threshold. While this of course is not possible in the boolean case, we employ the following trick to achieve a similar effect: replace each original coordinate with some large number of new coordinates. When the original coordinate was set to zero set all the corresponding new ones to zero, and when the original was set to 1, sample the new ones from some distribution such that (i) at least one of the new coordinates is always 1, and (ii) the number of 1s sampled has a large variation. In particular, we replace $X_{v,i}$ ($i \in [M]$) with $\{X_{v,i,b,q} \mid b \in \{0,1\}, q \in [Q]\}$ for some large $Q$. The vector $\mathbf{X}^{(1)}$ is constructed as follows. For all $i$ s.t. $\pi_{v,u}(i) \in J^v$ independently, we first sample $b = (1 - a) \in \{0,1\}$ u.a.r. then set exactly one u.a.r. chosen value from $\{X_{v,i,b,r}^{(1)}\}_{r=1}^{Q}$, to 1 and rest to zero. Further, after choosing $J^v$, a random subset $J_\varepsilon^v \subseteq J^v$ is chosen by including every $j \in J^v$ with probability $\varepsilon$. For those $i$ s.t. $\pi_{v,u}(i) \in J_\varepsilon^v$ the variables $\{X_{v,i,a,r}^{(1)}\}_{r=1}^{Q}$ are sampled iid unbiased $\{0,1\}$-Bernoulli. We can think of $J_\varepsilon^v$ be those blocks of coordinates which introduce the variability in the number of 1s due the fact that while a given $j \in J^v$ is included in $J_\varepsilon^v$ with probability only $\varepsilon$, once it is then roughly half of the coordinates in $\mathbf{X}^{(1)}$ corresponding to $i \in \pi_{v,u}^{-1}(j)$ are sampled (as iid Bernoulli) to be 1, rather than $\sim 1/Q$ fraction. For a choice $\varepsilon > 0$ letting $Q$ to be sufficiently larger than $1/\varepsilon$ yields the desired variation.

The coordinates $X_{w,i}$ ($i \in [M]$) are analogously replaced with $\{X_{w,i,b,q} \mid b \in \{0,1\}, q \in [Q]\}$, and a similar process using $J_\varepsilon^w \subseteq J^w$ is followed to sample the new variables for the point $\mathbf{X}^{(2)}$.

With the above transformations, $\mathcal{D}^0$ is replaced by the distribution $\mathcal{D}$ which samples vectors $\mathbf{X}^{(1)}$ and $\mathbf{X}^{(2)}$ now over the extended coordinate sets corresponding to $v$ and $w$ respectively. Since we maintain the property that if the original coordinate $X_{v,i}$ was 1 then at least one of $\{X_{v,i,0,r}, X_{v,i,1,r}\}_{r=1}^{Q}$ is set to 1, and when the former was 0 then the latter are all 0 as well (similarly for $X_{w,i}$), we still have that $\left(\bigvee_{b \in \{0,1\}} \bigvee_{q=1}^{Q} X_{v,l_v,b,q}\right) \vee \left(\bigvee_{b \in \{0,1\}} \bigvee_{q=1}^{Q} X_{w,l_w,b,q}\right)$ for consistent labels $l_v$ and $l_w$, satisfies all the bags sampled from $\mathcal{D}$. On the other hand, the soundness analysis for $\mathcal{D}$ is quite lengthy and technical, and we provide below an informal description of the main steps involved and the techniques used for them, using some convenient simplified notation.

Consider a linear form $h$ on the above set of coordinates and a candidate solution $\mathsf{pos}(h(\mathbf{X}))$ to the dictatorship test. For simplicity let us assume that it does not have any constant term. If $\mathbf{c}_v$ and $\mathbf{c}_w$

are the coefficient vectors of $h$ restricted to the coordinates corresponding to $v$ and $w$ separately, then $\mathsf{pos}(h(\mathbf{X}^{(1)}) = \mathsf{pos}(\langle \mathbf{c}_v, \mathbf{X}^{(1)} \rangle)$ and $\mathsf{pos}(h(\mathbf{X}^{(2)}) = \mathsf{pos}(\langle \mathbf{c}_w, \mathbf{X}^{(2)} \rangle)$. For $v$ let the weight of its label $i$ be given by $\|\mathbf{c}_{v,i}\|_2^2 = \sum_{b,q} c_{v,i,b,q}^2$ with the total weight of $\mathbf{c}_v$ being $\|\mathbf{c}_v\|_2^2$ normalized to 1. Using the notion of *critical index* with small parameter $\tau > 0$ we partition $[M]$ into subsets $\mathsf{reg}_v$ of *regular* coordinates which all have weight at most $\tau$, $\mathsf{top}_v$ of the top-$K$ (for some large enough $K$) non-regular coordinates in terms of their weight, and $\mathsf{torso}_v$ of the rest of the coordinates. Let $h^v$, $h^v_{\mathrm{top}}$, $h^v_{\mathrm{reg}}$ and $h^v_{\mathrm{tor}}$ be the linear forms on $\mathbf{X}^{(1)}$ given by the restrictions of $\mathbf{c}_v$ to coordinates in $[M]$, $\mathsf{top}_v$, $\mathsf{reg}_v$ and $\mathsf{torso}_v$ respectively. Similarly we obtain for $w$ the subsets $\mathsf{top}_w$, $\mathsf{torso}_w$, $\mathsf{reg}_w$ of coordinates and the restrictions $h^w$, $h^w_{\mathrm{top}}$, $h^w_{\mathrm{reg}}$ and $h^w_{\mathrm{tor}}$.

For the soundness analysis of $\mathcal{D}$ we assume that $h$ does not admit a consistent independent decoding of $v$ and $w$, and based on this wish to show that the distributions $\mathsf{pos}(h(\mathbf{X}^{(1)}))$ and $\mathsf{pos}(h(\mathbf{X}^{(2)}))$ are independent of each other. We do this through the following arguments.

*Truncation*: $h^v_{\mathrm{tor}}(\mathbf{X}^{(1)})$, $h^w_{\mathrm{tor}}(\mathbf{X}^{(2)})$ are too small to affect the value of $\mathsf{pos}(h(\mathbf{X}^{(1)})$ and $\mathsf{pos}(h(\mathbf{X}^{(2)})$ respectively, with any significant probability. This uses an anti-concentration result of [14] applied to $h^v_{\mathrm{top}}$ and $h^w_{\mathrm{top}}$.

*Non-Intersection.* $\pi_{v,u}(\mathsf{top}_v) \cap \pi_{w,u}(\mathsf{top}_w) = \emptyset$. Further, the total weights of coordinates in $\mathsf{reg}_v \cap \pi_{w,u}^{-1}(\pi_{w,u}(\mathsf{top}_w))$ and those in $\mathsf{reg}_w \cap \pi_{w,u}^{-1}(\pi_{v,u}(\mathsf{top}_v))$ are tiny. If not, one could select a label for $v$ from either w.p. $1/2$ choosing randomly from $\mathsf{top}_v$, or w.p. $1/2/$ sampling one from $\mathsf{reg}_v$ with probability proportional to its weight, and do a similar independent decoding for $w$, yielding a consistent labeling with significant probability, thereby contradicting our assumption above.

*Decoupling the regular parts.* We show that with high probability over the choice of $J^v$, the distribution of $h^v_{\mathrm{reg}}(\mathbf{X}^{(1)})$ is close to a fixed Gaussian random variable. This is via an application of the Berry-Esseen theorem after a series of estimates on the conditional (on choice of $J^v$) mean and variance of $h^v_{\mathrm{reg}}(\mathbf{X}^{(1)})$. We obtain the same for $J^w$ and $h^w_{\mathrm{reg}}$.

Using the truncation argument, one can ignore the contribution of $h^v_{\mathrm{tor}}$ and $h^w_{\mathrm{tor}}$. From the non-intersection and decoupling arguments we obtain that $\mathsf{pos}(h^v_{\mathrm{reg}}(\mathbf{X}^{(1)}) + h^v_{\mathrm{top}}(\mathbf{X}^{(1)})$ and $\mathsf{pos}(h^w_{\mathrm{reg}}(\mathbf{X}^{(2)}) + h^w_{\mathrm{top}}(\mathbf{X}^{(w)})$ are independent of each other. A short argument averaging over all pairs of neighbors $v, w$ of a given $u$ shows that the fraction of bags satisfied (on an average over the choice of $v$ and $w$) is at most $1/2 + o(1)$.

In several steps of the above arguments we are aided by the *smoothness* property of the the Label Cover variant (see Sec. 3) which allows us to assume that in any pre-image $\pi_{v,u}^{-1}(j)$ there is at most one coordinate either from $\mathsf{top}_v$ or of weight greater than $1/\mathrm{poly}(d)$, where $d$ is the pre-image size of the Label Cover projections. Overall, our analysis follows (at a high level) the analytical approach of [12] and [14], though our decoupling step is more involved as it requires sharper as well as conditional bounds to obtain invariance as a high probability statement over the choice of $J^v$.

Finally, extending our analysis to rule out functions of constantly many LTFs considerably complicates the above arguments. In particular, we apply the *multi-dimensional* version of the Berry-Esseen theorem to do the decoupling step. For this, we add additional noise to the $h_{\mathrm{reg}}$ parts to lower bound the smallest eigenvalue of a correlation matrix which determines the error bounds when applying the multi-dimensional version of this theorem.

## 2   Approximation for LLP-LTF on bags of size $2$

Let $\mathcal{I} = (\mathcal{X}, \mathcal{B} = \{B_1, \ldots, B_m\}, \{\sigma_k\}_{k=1}^m)$ be a satisfiable instance of LLP-LTF. We begin with the following simple lemma which implies that the existence of an LTF satisfying all the bags of $\mathcal{I}$ while classifying each point of the instance with non-zero margin.

**Lemma 2.1.** *There is an LTF* $\mathsf{pos}(f(\mathbf{x}))$ *that satisfies all the bags of* $\mathcal{I}$, *along with* $|f(\mathbf{x})| > 0$ *for all* $\mathbf{x} \in \cup_{j=1}^m B_j$.

*Proof.* By definition there is an LTF $\mathsf{pos}(g(\mathbf{x}))$ that satisfies all the bags of $\mathcal{I}$. Since $\mathsf{pos}(z) = \mathbb{1}_{\{z>0\}}$ and the total number of feature vectors in the bags is finite, the quantity $\kappa$ given by

$$\kappa := \min \left\{ g(\mathbf{x}) \mid \mathbf{x} \in \cup_{j=1}^m B_j, \ \mathsf{pos}(g(\mathbf{x})) = 1 \right\} \tag{1}$$

---

Algorithm $\mathcal{A}$. Input: satisfiable instance $\mathcal{I}$ of LLP-LTF.

    1. For each $\mathbf{x} \in \mathbb{R}^d$ define $\tilde{\mathbf{x}} := (x_1, \ldots, x_d, 1) \in \mathbb{R}^{d+1}$.

    2. Solve the following SDP for psd matrix $\mathbf{C} \in \mathbb{R}^{(d+1) \times (d+1)}$:

$$\tilde{\mathbf{x}}_i^\mathsf{T} \mathbf{C} \tilde{\mathbf{x}}_j \ \leq \ 0 \qquad\qquad \forall \{\mathbf{x}_i, \mathbf{x}_j\} = B_k \text{ s.t. } |B_k| = 2, \sigma_k = 0.5 \qquad (2)$$

$$\tilde{\mathbf{x}}_i^\mathsf{T} \mathbf{C} \tilde{\mathbf{x}}_j \ \geq \ 0 \qquad\qquad \forall \{\mathbf{x}_i, \mathbf{x}_j\} = B_k \text{ s.t. } |B_k| = 2, \sigma_k \neq 0.5 \qquad (3)$$

$$\tilde{\mathbf{x}}_i^\mathsf{T} \mathbf{C} \tilde{\mathbf{x}}_i \ > \ 0 \qquad\qquad\qquad\qquad \forall \mathbf{x}_i \in \cup_{j=1}^m B_j. \qquad (4)$$

    3. Let $\mathbf{C} = \mathbf{L}^\mathsf{T} \mathbf{L}$ be its psd decomposition.

    4. Sample $\mathbf{g}$ u.a.r from $N(0,1)^{d+1}$.

    5. Define the linear form $h(\mathbf{x}) := \langle \mathbf{L}\tilde{\mathbf{x}}, \mathbf{g} \rangle$.

    6. Let $h^* \in \{h, -h\}$ such that $\mathsf{pos}(h^*(.))$ satisfies more bags of $\mathcal{I}$. Output $\mathsf{pos}(h^*(.))$.

---

Figure 1: Algorithm $\mathcal{A}$ for LLP-LTF.

satisfies $\kappa > 0$. Thus, we obtain the desired linear form $f$ by decreasing the constant term of $g$ by exactly $\kappa/2$, ensuring that $f(\mathbf{x}) \geq \kappa/2$ if $g(\mathbf{x}) > 0$, and $f(\mathbf{x}) \leq -\kappa/2$ if $g(\mathbf{x}) \leq 0$, $\forall \mathbf{x} \in \cup_{j=1}^m B_j$. $\qquad\qquad\square$

We first provide in Fig. 1 a SDP based algorithm $\mathcal{A}$ and prove the following lemma.

**Lemma 2.2.** *Algorithm $\mathcal{A}$ (Fig. 1) satisfies in expectation half of the non-monochromatic bags and one-fourth of the monochromatic bags.*

*Proof.* Let $\mathsf{pos}(f(\mathbf{x}))$ be the LTF given by Lemma 2.1 satisfying all the bags of $\mathcal{I}$. Letting $\tilde{\mathbf{x}}$ as defined in Step 1 of $\mathcal{A}$ along with the non-zero margin of $\mathsf{pos}(f(\mathbf{x}))$ implies that there is some $\mathbf{c}_* \in \mathbb{R}^{d+1}$ such that:

$$\langle \mathbf{c}_*, \tilde{\mathbf{x}}_i \rangle \langle \mathbf{c}_*, \tilde{\mathbf{x}}_j \rangle = \tilde{\mathbf{x}}_i^\mathsf{T} \left( \mathbf{c}_* \mathbf{c}_*^\mathsf{T} \right) \tilde{\mathbf{x}}_j \ \leq \ 0 \qquad \forall \{\mathbf{x}_i, \mathbf{x}_j\} = B_k \text{ s.t. } |B_k| = 2, \sigma_k = 0.5 \quad (5)$$

$$\langle \mathbf{c}_*, \tilde{\mathbf{x}}_i \rangle \langle \mathbf{c}_*, \tilde{\mathbf{x}}_j \rangle = \tilde{\mathbf{x}}_i^\mathsf{T} \left( \mathbf{c}_* \mathbf{c}_*^\mathsf{T} \right) \tilde{\mathbf{x}}_j \ \geq \ 0 \qquad \forall \{\mathbf{x}_i, \mathbf{x}_j\} = B_k \text{ s.t. } |B_k| = 2, \sigma_k \neq 0.5 \quad (6)$$

$$\langle \mathbf{c}_*, \tilde{\mathbf{x}}_i \rangle \langle \mathbf{c}_*, \tilde{\mathbf{x}}_i \rangle = \tilde{\mathbf{x}}_i^\mathsf{T} \left( \mathbf{c}_* \mathbf{c}_*^\mathsf{T} \right) \tilde{\mathbf{x}}_i \ > \ 0 \qquad\qquad\qquad \forall \mathbf{x}_i \in \cup_{j=1}^m B_j. \quad (7)$$

Thus, $\mathbf{C} = \mathbf{c}_* \mathbf{c}_*^\mathsf{T}$ satisfies (2), (3) and (4). and the SDP solved by $\mathcal{A}$ is feasible. Let $\mathbf{C} = \mathbf{L}^\mathsf{T} \mathbf{L}$ denote a solution and its psd decomposition. For each $i \in [n]$ consider the vector $\mathbf{z}_i \in \mathbb{R}^{d+1}$ given by $\mathbf{z}_i := \mathbf{L}\tilde{\mathbf{x}}_i \in \mathbb{R}^{d+1}$. Observe that $\langle \mathbf{z}_i, \mathbf{z}_j \rangle = \langle \mathbf{L}\tilde{\mathbf{x}}_i, \mathbf{L}\tilde{\mathbf{x}}_j \rangle = \tilde{\mathbf{x}}_i^\mathsf{T} \mathbf{L}^\mathsf{T} \mathbf{L} \tilde{\mathbf{x}}_j = \tilde{\mathbf{x}}_i^\mathsf{T} \mathbf{C} \tilde{\mathbf{x}}_j$, for $i, j \in [n]$. This, along with (2), (3), and (4) yields,

$$\langle \mathbf{z}_i, \mathbf{z}_j \rangle \ \leq \ 0 \qquad\qquad \forall \{\mathbf{x}_i, \mathbf{x}_j\} = B_k \text{ s.t. } |B_k| = 2, \sigma_k = 0.5 \qquad (8)$$

$$\langle \mathbf{z}_i, \mathbf{z}_j \rangle \ \geq \ 0 \qquad\qquad \forall \{\mathbf{x}_i, \mathbf{x}_j\} = B_k \text{ s.t. } |B_k| = 2, \sigma_k \neq 0.5 \qquad (9)$$

$$\|\mathbf{z}_i\| \ > \ 0 \qquad\qquad\qquad\qquad \forall \mathbf{x}_i \in \cup_{j=1}^m B_j \qquad (10)$$

Let $\mathbf{g}$ be as in the algorithm and note that as defined in Step 4 of $\mathcal{A}$, $h(\mathbf{x}_i) = \langle \mathbf{L}\tilde{\mathbf{x}}_i, \mathbf{g} \rangle = \langle \mathbf{z}_i, \mathbf{g} \rangle$. Thus, $\{\mathsf{pos}(h(\mathbf{x}_i))\}_{i=1}^n$ is a random hyperplane rounding of the vectors $\{\mathbf{z}_i\}_{i=1}^n$. Using this observation along with (8), (9) and standard geometric facts (see Appendix A of the supplemental) we obtain:

$$\Pr\left[ \mathsf{pos}(h(\mathbf{x}_i)) \neq \mathsf{pos}(h(\mathbf{x}_j)) \right] \ \geq \ 1/2 \qquad \forall \{\mathbf{x}_i, \mathbf{x}_j\} = B_k \text{ s.t. } |B_k| = 2, \sigma_k = 0.5 \quad (11)$$

$$\Pr\left[ \mathsf{pos}(h(\mathbf{x}_i)) = \mathsf{pos}(h(\mathbf{x}_j)) \right] \ \geq \ 1/2 \qquad \forall \{\mathbf{x}_i, \mathbf{x}_j\} = B_k \text{ s.t. } |B_k| = 2, \sigma_k \neq 0.5 \quad (12)$$

$$\Pr\left[ \mathsf{pos}(h(\mathbf{x}_i)) = \mathsf{pos}(h(\mathbf{x}_i)) \right] \ = \ 1 \qquad\qquad\qquad \forall \{\mathbf{x}_i\} = B_k \quad (13)$$

where the last equation for bags of size 1 is trivially true. Further, (10) implies that over the choice of $\mathbf{g}$ the values of all the $h(\mathbf{x}_i)$ occurring in the above three equations are non-zero *almost surely* i.e., with probability 1. This implies that $\mathsf{pos}(h(\mathbf{x})) = 1 - \mathsf{pos}(-h(\mathbf{x}))$ for all $\mathbf{x} \in \cup_{j=1}^m B_j$, with probability 1. Thus, it can be seen that both $\mathsf{pos}(h)$ and $\mathsf{pos}(-h)$ satisfy all the non-monochromatic bags $B = \{\mathbf{x}_i, \mathbf{x}_j\}$ such that $\mathsf{pos}(h(\mathbf{x}_i)) \neq \mathsf{pos}(h(\mathbf{x}_j))$. Furthermore, at least one of $\mathsf{pos}(h)$ and $\mathsf{pos}(-h)$ satisfy half of the monochromatic bags $B$ which have vectors $\mathbf{x}_i, \mathbf{x}_j$ ($\mathbf{x}_i = \mathbf{x}_j$ if $|B| = 1$) for which $\mathsf{pos}(h(\mathbf{x}_i)) = \mathsf{pos}(h(\mathbf{x}_j))$. This implies that $\mathsf{pos}(h^*)$ returned by $\mathcal{A}$ satisfies in expectation at least half of the non-monochromatic bags and one-fourth of the monochromatic bags. $\qquad\square$

*Proof.* (of Theorem 1.1 for LLP-LTF) We use $\mathcal{A}$ in combination with the linear programming method $\overline{\mathcal{A}}$ that is guaranteed to satisfy all monochromatic bags. If the fraction of monochromatic bags $q$ is at least $2/5$ of the total bags, then $\overline{\mathcal{A}}$ finds an LTF to satisfy all of them. Otherwise, $\mathcal{A}$ finds one to satisfy (in expectation) $q/4 + 1/2(1-q) = 1/2 - q/4 \geq 1/2 - 2/20 = 2/5$ fraction of the bags. If $q = 0$ i.e., all bags are non-monochromatic the LTF satisfies $1/2$-fraction of the bags. $\qquad\square$

## 2.1 Special Case: LLP-OR

Let $\mathcal{I}$ be a satisfiable instance of LLP-OR. We have $\mathbf{x}_i \in \{0,1\}^d$ $(i \in [n])$ and there is boolean OR function $f$ of the form:

$$f(\mathbf{x}) = \bigvee_{s \in S_1^f} x_s + \bigvee_{s \in S_0^f} \neg x_s, \quad \mathbf{x} \in \{0,1\}^d, \tag{14}$$

for disjoint subsets $S_1^f, S_0^f \subseteq [d]$ such that $f$ satisfies each bag of the instance $\mathcal{I}$. Observe that for any $\mathbf{x} \in \{0,1\}^d$

$$\left[ f(\mathbf{x}) = 1 \Leftrightarrow \sum_{s \in S_1^f} x_s + \sum_{s \in S_0^f} (1 - x_s) \geq 1 \right], \quad \left[ f(\mathbf{x}) = 0 \Leftrightarrow \sum_{s \in S_1^f} x_s + \sum_{s \in S_0^f} (1 - x_s) = 0 \right] \tag{15}$$

Let us define $\mathbf{c}^f = (c_1^f, \ldots, c_d^f, c_{d+1}^f) \in \mathbb{R}^{d+1}$ as $\mathbf{c}_s^f = 1$ if $s \in S_1^f$, else $\mathbf{c}_s^f = -1$ if $s \in S_0^f$, else $\mathbf{c}_s^f = \left| S_0^f \right| - 1/2$ if $s = d+1$, and $\mathbf{c}_s^f = 0$ otherwise. Then, letting $\tilde{\mathbf{x}} = (x_1, \ldots, x_d, 1)$, (15) is transformed to

$$\left[ f(\mathbf{x}) = 1 \Leftrightarrow \langle \mathbf{c}^f, \tilde{\mathbf{x}} \rangle \in [1/2, d - 1/2] \right], \quad \left[ f(\mathbf{x}) = 0 \Leftrightarrow \langle \mathbf{c}^f, \tilde{\mathbf{x}} \rangle = -1/2 \right] \tag{16}$$

Given the above it can be seen that

$$\langle \mathbf{c}^f, \tilde{\mathbf{x}}_i \rangle \langle \mathbf{c}^f, \tilde{\mathbf{x}}_j \rangle \leq \frac{-1}{4d} \left( \langle \mathbf{c}^f, \tilde{\mathbf{x}}_i \rangle^2 + \langle \mathbf{c}^f, \tilde{\mathbf{x}}_j \rangle^2 \right) \quad \forall \{\mathbf{x}_i, \mathbf{x}_j\} = B_k \text{ s.t. } |B_k| = 2, \sigma_k = 0.5$$

$$\langle \mathbf{c}^f, \tilde{\mathbf{x}}_i \rangle \langle \mathbf{c}^f, \tilde{\mathbf{x}}_j \rangle \geq \frac{1}{4d} \left( \langle \mathbf{c}^f, \tilde{\mathbf{x}}_i \rangle^2 + \langle \mathbf{c}^f, \tilde{\mathbf{x}}_j \rangle^2 \right) \quad \forall \{\mathbf{x}_i, \mathbf{x}_j\} = B_k \text{ s.t. } |B_k| = 2, \sigma_k \neq 0.5$$

Thus, one modifies the algorithm in Fig. 1 to obtain $\mathcal{A}'$ by replacing (2) and (3) with

$$\tilde{\mathbf{x}}_i^\mathsf{T} \mathbf{C} \tilde{\mathbf{x}}_j \leq \frac{-1}{4d} \left( \tilde{\mathbf{x}}_i^\mathsf{T} \mathbf{C} \tilde{\mathbf{x}}_i + \tilde{\mathbf{x}}_j^\mathsf{T} \mathbf{C} \tilde{\mathbf{x}}_j \right) \quad \forall \{\mathbf{x}_i, \mathbf{x}_j\} = B_k \text{ s.t. } |B_k| = 2, \sigma_k = 0.5 \tag{17}$$

$$\tilde{\mathbf{x}}_i^\mathsf{T} \mathbf{C} \tilde{\mathbf{x}}_j \geq \frac{1}{4d} \left( \tilde{\mathbf{x}}_i^\mathsf{T} \mathbf{C} \tilde{\mathbf{x}}_i + \tilde{\mathbf{x}}_j^\mathsf{T} \mathbf{C} \tilde{\mathbf{x}}_j \right) \quad \forall \{\mathbf{x}_i, \mathbf{x}_j\} = B_k \text{ s.t. } |B_k| = 2, \sigma_k \neq 0.5 \tag{18}$$

Following the same analysis as in the LTF case, noting that $\|\mathbf{z}_i\|^2 = \tilde{\mathbf{x}}_i^\mathsf{T} \mathbf{C} \tilde{\mathbf{x}}_i > 0$ for any $i \in [m]$ (by (10)), and using the AM-GM inquality we obtain that:

$$\frac{\langle \mathbf{z}_i, \mathbf{z}_j \rangle}{\|\mathbf{z}_i\| \|\mathbf{z}_j\|} \leq \frac{-1}{4d} \left( \frac{\|\mathbf{z}_i\|^2 + \|\mathbf{z}_j\|^2}{\|\mathbf{z}_i\| \|\mathbf{z}_j\|} \right) \leq \frac{-1}{2d} \quad \forall \{\mathbf{x}_i, \mathbf{x}_j\} = B_k \text{ s.t. } |B_k| = 2, \sigma_k = 0.5 \tag{19}$$

$$\frac{\langle \mathbf{z}_i, \mathbf{z}_j \rangle}{\|\mathbf{z}_i\| \|\mathbf{z}_j\|} \geq \frac{1}{4d} \left( \frac{\|\mathbf{z}_i\|^2 + \|\mathbf{z}_j\|^2}{\|\mathbf{z}_i\| \|\mathbf{z}_j\|} \right) \geq \frac{1}{2d} \quad \forall \{\mathbf{x}_i, \mathbf{x}_j\} = B_k \text{ s.t. } |B_k| = 2, \sigma_k \neq 0.5 \tag{20}$$

Therefore, from standard geometric facts (see Appendix A of the supplemental), the random hyperplane rounding of the $\{\mathbf{z}_i\}_{i=1}^n$ yields probabilities $1/2 + \Omega(1/d)$ instead of $1/2$ in (11) and (12). Following the rest of the arguments as in the LLP-LTF case we obtain the following lemma.

**Lemma 2.3.** *The algorithm $\mathcal{A}'$ described above satisfies at least $(1/2 + \alpha_0/d)$-fraction of the non-monochromatic bags and $(1/4 + \alpha_0/2d)$-fraction of the monochromatic bags in expectation, for some positive constant $\alpha_0 \in (0,1]$.*

*Proof.* (of Theorem 1.1 for LLP-OR) The guarantee of the linear programming algorithm $\overline{\mathcal{A}}$ satisfying all monochromatic bags case holds for LLP-OR as well. Thus, if the fraction of monochromatic bags $x$ is greater than $2/5 + \alpha_0/(8d)$ then $\overline{\mathcal{A}}$ obtains an LTF that satisfies all the monochromatic bags. Otherwise, $\mathcal{A}'$ from Lemma 2.3 finds one to satisfy $(1/2 + \alpha_0/d)(1-x) + x(1/4 + \alpha_0/2d)$ fraction of the bags which is at least $(1/2 + \alpha_0/d) - (2/5 + \alpha_0/(8d))(1/4 + \alpha_0/2d) \geq (2/5 + \alpha_0/(4d))$ since $\alpha_0 \in (0,1]$ and $d \geq 1$. If $x = 0$ i.e., all bags are non-monochromatic the LTF satisfies $(1/2 + \alpha_0/d)$-fraction of the bags. $\qquad\square$

## 2.2 Experimental Evaluation

We evaluate Algorithm $\mathcal{A}$ (Fig. 1) on synthetically generated satisfiable LLP-LTF instances over dimension $d \in \{10, 40, 100\}$ and number of bags $m \in \{50, 100, 200\}$. For each $(d, m)$ the algorithm solves 100 independently generated instances - each by choosing a random satisfying linear form $f$ (with iid $N(0,1)$ coefficients) and $m$ bags independently, for each bag sampling two points iid from $N(0,1)^d$ with their label proportions given by $\mathsf{pos}(f(\mathbf{x}))$. Note that the bags and their constituent points are not correlated with $f$.

For each instance the SDP in (2)-(4) is solved and the best $h^*$ chosen using 100 independent Gaussians $\mathbf{g}$. With $m_0$ and $m_1$ denoting the number of monochromatic and non-monochromatic bags respectively, the theoretical performance threshold is given by $t := (m_0/4 + m_1/2)$ (Lemma 2.2). We measure $s$ as the number of bags satisfied by $\mathsf{pos}(h^*)$ and the average and minimum values of $(s/t)$ over the 100 instances per $(d, m)$, both of which are quite a bit larger than 1, showing that the algorithm performs much better than than the theoretical guarantee on such instances. We also compute $r$ as the number of bags satisfied by the best $\{\mathsf{pos}(h'), \mathsf{pos}(-h')\}$ out of 100 random linear forms $h'$, and the average value of $(r/s)$. The latter is below 1 in all cases indicating that our algorithm performs better on average than the random linear threshold with the gap being higher when $d$ is smaller. The results are presented in Table 1.

In the previous evaluation, for larger $d$ we observe only small gap between the performance of $\mathcal{A}$ and the random linear threshold. Our next experiment shows that this is not the case on instances with bags/points that are correlated to $f$. Here we choose $f$ passing through origin and random monochromatic bags having two points on the unit sphere which are nearly diametrically opposite and random non-monochromatic bags having two points on the unit sphere very close to each other. We reduce the number of trials to 5: taking the output of $\mathcal{A}$ from best of 5 samples of $\mathbf{g}$ as $\mathsf{pos}(h^*)$, and the best $\{\mathsf{pos}(h'), \mathsf{pos}(-h')\}$ out of 5 random linear forms $h'$. The results are in Table 2. Unsurprisingly, due to this geometry of points, the performance of the random linear thresholds (avg $r$, avg $(r/s)$) is much worse than before. On the other hand $h^*$ given by $\mathcal{A}$ performs better than before as the SDP is more discriminative and its solution $\mathbf{C}$ is highly correlated to $f$.

Table 1: Results on random 2-sized bags with iid points uncorrelated with satisfying linear form $f$. 100 instances per $(d, m)$ row.

| $d$ | $m$ | avg $m_0$ | avg $m_1$ | avg $t$ | avg $s$ | avg $\left(\frac{s}{t}\right)$ | min $\left(\frac{s}{t}\right)$ | avg $r$ | avg $\left(\frac{r}{s}\right)$ |
|---|---|---|---|---|---|---|---|---|---|
| 10 | 50 | 27.20 | 22.80 | 18.20 | 39.24 | 2.162 | 1.714 | 33.09 | 0.847 |
| 10 | 100 | 52.98 | 47.02 | 36.76 | 85.26 | 2.325 | 1.949 | 64.12 | 0.754 |
| 10 | 200 | 105.83 | 94.17 | 73.54 | 181.95 | 2.479 | 2.217 | 125.04 | 0.688 |
| 40 | 50 | 25.94 | 24.06 | 18.52 | 29.12 | 1.574 | 1.405 | 28.83 | 0.993 |
| 40 | 100 | 50.59 | 49.41 | 37.35 | 54.59 | 1.462 | 1.316 | 53.26 | 0.978 |
| 40 | 200 | 101.43 | 98.57 | 74.64 | 104.96 | 1.406 | 1.265 | 100.70 | 0.961 |
| 100 | 50 | 24.97 | 25.03 | 18.76 | 28.74 | 1.533 | 1.317 | 28.28 | 0.987 |
| 100 | 100 | 50.06 | 49.94 | 37.48 | 52.08 | 1.390 | 1.266 | 51.49 | 0.990 |
| 100 | 200 | 100.04 | 99.96 | 74.99 | 97.24 | 1.297 | 1.216 | 96.58 | 0.994 |

Table 2: Results on random 2-sized bags with points highly correlated with satisfying linear form $f$. 100 instances per $(d, m)$ row.

| $d$ | $m$ | avg $m_0$ | avg $m_1$ | avg $t$ | avg $s$ | avg $\left(\frac{s}{t}\right)$ | min $\left(\frac{s}{t}\right)$ | avg $r$ | avg $\left(\frac{r}{s}\right)$ |
|---|---|---|---|---|---|---|---|---|---|
| 10 | 50 | 25.73 | 24.27 | 18.57 | 48.74 | 2.632 | 1.200 | 2.37 | 0.049 |
| 10 | 100 | 49.94 | 50.06 | 37.52 | 98.17 | 2.619 | 2.222 | 5.19 | 0.053 |
| 10 | 200 | 98.58 | 101.42 | 75.36 | 199.61 | 2.650 | 2.500 | 8.86 | 0.044 |
| 40 | 50 | 24.81 | 25.19 | 18.80 | 47.19 | 2.517 | 1.639 | 2.54 | 0.055 |
| 40 | 100 | 49.36 | 50.64 | 37.66 | 96.77 | 2.574 | 1.176 | 4.36 | 0.045 |
| 40 | 200 | 99.26 | 100.74 | 75.18 | 195.42 | 2.601 | 2.135 | 8.59 | 0.044 |
| 100 | 50 | 24.92 | 25.08 | 18.77 | 46.05 | 2.460 | 1.105 | 2.62 | 0.060 |
| 100 | 100 | 50.70 | 49.30 | 37.33 | 93.20 | 2.498 | 1.139 | 4.21 | 0.046 |
| 100 | 200 | 99.26 | 100.74 | 75.18 | 189.85 | 2.527 | 1.462 | 8.69 | 0.046 |

# 3 Smooth Label Cover and Statement of Hardness Reduction

The Smooth-Label-Cover problem is defined as follows.

**Definition 3.1.** *A* Smooth-Label-Cover *instance* $\mathcal{L}((V_{\mathcal{L}}, U_{\mathcal{L}}, E_{\mathcal{L}} \subseteq V_{\mathcal{L}} \times U_{\mathcal{L}}), M, m, \{\pi_{v,u}\}_{(v,u) \in E_{\mathcal{L}}})$ *consists of a bi-regular connected bipartite graph with vertex sets* $V_{\mathcal{L}}, U_{\mathcal{L}}$, *a directed edge set* $E_{\mathcal{L}}$, *and a set of projections* $\{\pi_{v,u} : [M] \mapsto [m]\}_{(v,u) \in E_{\mathcal{L}}}$. *A labeling* $\sigma := (\sigma_V, \sigma_U)$ *s.t.* $\sigma_V : V_{\mathcal{L}} \mapsto [M]$ *and* $\sigma_U : U_{\mathcal{L}} \mapsto [m]$ *is said to* satisfy *an edge* $(v, u)$ *if* $\pi_{v,u}(\sigma_V(v)) = \sigma_U(u)$.

The hardness of Smooth-Label-Cover is given by the following theorem (see Appendix B.2 of the supplemental).

**Theorem 3.2.** *There exists an absolute constant* $\kappa_0 > 0$ *such that for all integer parameters* $z$ *and* $J$, *it is* NP-*Hard to distinguish whether an instance* $\mathcal{L}$ *of* Smooth-Label-Cover *with* $M = 7^{(J+1)z}$ *and* $m = 2^z 7^{Jz}$, *satisfies,*

- YES*: There exists a labeling* $\sigma := (\sigma_V, \sigma_U)$ *which satisfies all the edges.*
- NO*: There is no labeling* $\sigma$ *which satisfies more than* $2^{-\kappa_0 z}$-*fraction of the edges.*

*Additionally,* $\mathcal{L}$ *satisfies the following properties:*

- (Smoothness) *For every vertex* $v \in V$ *and for a randomly sampled edge* $(v, u)$ *incident on* $v$, $\Pr_{u \sim v}[\pi_{v,u}(i) = \pi_{v,u}(j)] \leq \frac{1}{J}$, *for any fixed pair of distinct labels* $i, j \in [M]$.
- *For any edge* $(v, u)$, *and any label* $j \in [m]$, $|\pi_{v,u}^{-1}(j)| \leq d$, *where* $d = 4^z$.

The hardness reduction from the above to LLP-OR is given below:

**Theorem 3.3.** *For any constants* $\delta > 0$ *and* $\ell \in \mathbb{Z}^+$, *there is a polynomial time reduction from an instance* $\mathcal{L}$ *of* Smooth-Label-Cover *given by Theorem 3.2 with* $J$ *and* $z$ *depending only on* $\delta$ *and* $\ell$, *to an instance* $\mathcal{I}$ *of* LLP-OR *with only non-monochromatic 2-sized bags such that:*

- (Completeness) *If* $\mathcal{L}$ *is a* YES *instance then there is a monotone* OR *that satisfies all the bags of* $\mathcal{I}$.
- (Soundness) *If* $\mathcal{L}$ *is a* NO *instance then there is no function of at most* $\ell$ *LTFs that satisfies more than* $1/2 + \delta$ *fraction of the bags of* $\mathcal{I}$

Due to lack of space, the proof of the above theorem is omitted and appears in Appendices C and D of the supplemental.

# 4 Conclusions and Future Work

We present in this work the first study of the proper learnability of LTFs from label proportions, on bags of size at most two. Defining this formally as the LLP-LTF problem and the special case of learning OR using LTFs as LLP-OR (in Section 1.2) we give efficient algorithms to compute an LTF satisfying $(2/5)$-fraction of the bags of a satisfiable LLP-LTF instance, and $(1/2)$-fraction of them if all bags are non-monochromatic. For satisfiable LLP-OR over $d$-dimensional boolean vectors we improve these factors to $(2/5 + \Omega(1/d))$ and $(1/2 + \Omega(1/d))$ respectively. In our main (and most technically challenging) result we prove the NP-hardness of satisfying using any function of constantly many LTFs more than $(1/2 + o(1))$-fraction of bags of an instance of LLP-OR with only non-monochromatic 2-sized bags all guaranteed to be satisfiable by some monotone-OR, thus establishing the optimality of our algorithms for the non-monochromatic bags case.

While the above we feel, is notable progress in understanding the complexity of LLP-LTF, there are many interesting related directions for future work. First is resolving the gap between the $(2/5)$-factor algorithm and the $(1/2 + o(1))$-factor hardness. Obtaining efficient proper learning algorithms for larger bag sizes is a natural followup problem. Whether using more complicated hypotheses – like polynomial-threshold functions (PTFs) – allows us to obtain better bounds for LLP learning LTFs is another important question. Proving NP-hardness results ruling PTFs out as good classifiers seems challenging – and is open even for agnostically learning noisy ORs in supervised-learning. All these questions can also be studied for other classes such as DNF-formulas. We hope that our work along with this rich landscape of open questions generates further interest in the study of learnability of common concept classes from label proportions.

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
