# Learnability of Linear Thresholds from Label Proportions (Supplemental Appendix)

**Rishi Saket**
Google Research, India
rishisaket@google.com

## A   Hyperplane Rounding and Derandomization

The following is a standard geometric fact related to hyperplane rounding.

**Fact A.1.** *Let $\mathbf{z}_1$ and $\mathbf{z}_2$ be two unit vectors in $d$-dimensions (for any positive integer $d$). If the angle subtended by them is $\theta$, then*

$$\Pr_{\mathbf{g}}\left[\mathsf{pos}\left(\langle \mathbf{z}_1, \mathbf{g}\rangle\right) \neq \mathsf{pos}\left(\langle \mathbf{z}_2, \mathbf{g}\rangle\right)\right] = \theta/\pi, \tag{1}$$

*where $\mathbf{g} \sim N(0,1)^d$ is a vector with each coordinate iid standard Gaussian.*

Note that (i) if $\langle \mathbf{z}_1, \mathbf{z}_2\rangle \geq 0$, then $\theta \leq \pi/2$, and therefore the probability in (1) is at most $1/2$, and (ii) if $\langle \mathbf{z}_1, \mathbf{z}_2\rangle \leq 0$, then $\theta \geq \pi/2$, and therefore the probability in (1) is at least $1/2$. This can be strengthened by observing that $d(\cos\theta)/d\theta = -\sin\theta \in [-1,0]$ for $\theta \in [0,\pi]$. From the above it follows that for any $\delta \in [0,1]$,

- If $\langle \mathbf{z}_1, \mathbf{z}_2\rangle \geq \delta$ then $\theta \leq \pi/2 - \delta$, and therefore the probability in (1) is at most $1/2 - \delta/\pi$.
- If $\langle \mathbf{z}_1, \mathbf{z}_2\rangle \leq -\delta$ then $\theta \geq \pi/2 + \delta$, and therefore the probability in (1) is at least $1/2 + \delta/\pi$.

Therefore, the constant $\alpha_0$ in the proof of Lemma 2.3 can be taken to be $1/2\pi$.

### A.1   Derandomization of Theorem 1.1 using [4]

Consider the algorithm given by Theorem 1.1 for LLP-LTF in Section 2. The objective for which randomized rounding is applied can be recast as a Max-Cut problem as follows. For each non-monochromatic bag consisting of $\{\mathbf{x}_i, \mathbf{x}_j\}$, we consider the pair $\{\mathbf{z}_i, \mathbf{z}_j\}$ to be separated in the random hyperplane, while for each monochromatic bag of size 2 consisting of $\{\mathbf{x}_i, \mathbf{x}_j\}$ we consider the pair $\{\mathbf{z}_i, -\mathbf{z}_j\}$ (arbitrarily negating one of them) to be separated by the random hyperplane rounding. The procedure of [4] (as applied to Max-Cut) will deterministically find one such $\mathbf{g}$ which preserves the expected number of pairs separated. The rest of the algorithm using the obtained $\mathbf{g}$ remains the same. A similar process applies to the algorithm given in Section 2.1 for LLP-OR.

## B   Preliminaries for Hardness Result

We will use the usual Berry-Esseen theorem below, as well as the multi-dimensional version.

**Theorem B.1** (Berry-Esseen Theorem, [5])**.** *Let $X_1, \ldots, X_n$ be independent random variables with $\mathsf{E}[X_i] = 0$ and $\mathrm{Var}[X_i] = \sigma_i^2$. Let $\sigma^2 = \sum_{i \in n} \sigma_i^2$. Then,*

$$\sup_{t \in \mathbb{R}} \left| \Pr_{X_1, \ldots, X_n} \left[ \sigma^{-1} \sum_{i \in [n]} X_i \leq t \right] - \Phi(t) \right| \leq c\gamma \tag{2}$$

35th Conference on Neural Information Processing Systems (NeurIPS 2021).

where $c$ is a universal constant, $\Phi$ is the CDF of the standard Gaussian $N(0,1)$, and $\gamma :=$ $\sigma^{-1} \max_{i \in [n]} (\mathsf{E}\left[|X_i|^3\right] / \sigma_i^2)$.

**Theorem B.2** (Multi-dimensional Berry-Esseen Theorem, [6]). *Let* $\mathbf{X}_1, \ldots, \mathbf{X}_n$ *be independent random vectors in* $\mathbb{R}^d$ *with* $\mathsf{E}[\mathbf{X}_i] = 0$. *Let* $\mathbf{S} = \sum_{i=1}^n \mathbf{X}_i$ *and* $\boldsymbol{\Sigma} = \mathrm{Cov}[\mathbf{S}]$. *Then for all convex sets* $A \subseteq \mathbb{R}^d$

$$|\mathsf{P}\left[\mathbf{S} \in A\right] - \Pr[\mathbf{Z} \in A| \leq Cd^{1/4}\gamma \tag{3}$$

*where $C$ is a universal constant,* $\mathbf{Z} \sim N(0, \boldsymbol{\Sigma})$ *and* $\gamma := \sum_{i=1}^n \mathsf{E}\left[\left\|\boldsymbol{\Sigma}^{-1/2}\mathbf{X}_i\right\|_2^3\right]$.

**Theorem B.3** (Chernoff-Hoeffding). *Let* $X_1, \ldots, X_n$ *be independent random variables, s.t.* $a_i \leq X_i \leq b_i$, $\Delta_i = b_i - a_i$ *for* $i = 1, \ldots, n$. *Then, for any* $t > 0$,

$$\Pr\left[\left|\sum_{i=1}^n X_i - \sum_{i=1}^n \mathsf{E}[X_i]\right| > t\right] \leq 2 \cdot \exp\left(-\frac{2t^2}{\sum_{i=1}^n \Delta_i^2}\right).$$

**Chebyshev's Inequality.** For any random variable $X$ and $t > 0$,

$$\Pr\left[|X| > t\right] \leq \mathsf{E}[X^2]/t^2. \tag{4}$$

We will also use the following anti-concentration lemma proved in [2].

**Lemma B.4.** *Let* $\mathbf{c}_1, \mathbf{c}_2, \ldots, \mathbf{c}_T \in \mathbb{R}^R$ *be such that* $\|\mathbf{c}_1\| \geq \|\mathbf{c}_2\| \geq \cdots \geq \|\mathbf{c}_T\|$. *Suppose* $\mathbf{X}_1, \mathbf{X}_2, \ldots, \mathbf{X}_T$ *are $R$-dimensional Bernoulli random variables i.e.,* $\mathbf{X}_i \stackrel{u.a.r}{\sim} \{0,1\}^R$, $i \in [T]$. *Then,*

$$\sup_{\theta \in \mathbb{R}} \Pr\left[\left|\sum_{i \in [T]} \langle \mathbf{c}_i, \mathbf{X}_i \rangle + \theta\right| \leq \frac{\|\mathbf{c}_T\|}{T^{1/2}}\right] \leq O(T^{-1/2}).$$

The total variation distance between two distributions $P, Q$ over $\mathbb{R}^d$ is $\mathrm{TV}(P,Q) := \sup_{A \subseteq \mathbb{R}^d} |P(A) - Q(A)|$. The following gives a bound on this distance between two multi-dimensional Gaussians with the same mean.

**Theorem B.5** ([1]). *If* $\boldsymbol{\Sigma}_1$ *and* $\boldsymbol{\Sigma}_2$ *are positive-definite* $d \times d$ *matrices, and let* $\lambda_1, \ldots, \lambda_d$ *be the eigenvalues of* $\boldsymbol{\Sigma}_1^{-1}\boldsymbol{\Sigma}_2 - \mathbf{I}_d$, *then for any* $\boldsymbol{\mu} \in \mathbb{R}^d$

$$\mathrm{TV}\left(N\left(\boldsymbol{\mu}, \boldsymbol{\Sigma}_1\right), N\left(\boldsymbol{\mu}, \boldsymbol{\Sigma}_2\right)\right) \leq (3/2)\left(\sum_{i=1}^d \lambda_i^2\right)^{1/2}$$

The total variation between one-dimensional Gaussians is as follows.

**Theorem B.6** ([1]).

$$\mathrm{TV}\left(N(\mu_1, \sigma_1^2), N(\mu_2, \sigma_2^2)\right) \leq \frac{3|\sigma_1^2 - \sigma_2^2|}{\sigma_1^2} + \frac{|\mu_1 - \mu_2|}{2\sigma_1}.$$

We also use the Gaussian anti-concentration bound (which is obtained by simple integration) that for $g \sim N(0, \sigma^2)$

$$\Pr[g \in [t, t+\delta]] \leq \delta/(\sigma\sqrt{2\pi}) \tag{5}$$

for any $t \in \mathbb{R}$ and $\delta > 0$.

## B.1 Critical Index

We first define the following notion of regularity.

**Regularity.** We say that a sequence of values $\{a_i \mid i \in I\}$ is $\nu$-*regular* if $a_i^2 \leq \nu \sum_{i' \in I} a_{i'}^2$ for all $i \in I$.

For some $M$ and $R$ consider a sequence of vectors $\{\mathbf{c}_i \in \mathbb{R}^R\}_{i=1}^M$ whose concatenation is $\mathbf{c} := \oplus_{i=1}^M \mathbf{c}_i$ so that $\|\mathbf{c}\|_2^2 = \sum_{i=1}^M \|\mathbf{c}_i\|_2^2$. We will work with linear forms $h = \sum_{i=1}^M \langle \mathbf{c}_i, \mathbf{X}_i \rangle$ where $\mathbf{X}_i \in \mathbb{R}^R$ ($1 \leq i \leq M$) are variables.

Let $\sigma : [M] \to [M]$ be the unique ordering s.t.

$$\|\mathbf{c}_{\sigma(1)}\|_2 \geq \|\mathbf{c}_{\sigma(2)}\|_2 \geq \cdots \geq \|\mathbf{c}_{\sigma(M)}\|_2, \tag{6}$$

and,

$$\sigma(i) < \sigma(i'), \qquad \forall i, i' \text{ s.t. } i < i' \text{ and } \|\mathbf{c}_{\sigma(i)}\|_2 = \|\mathbf{c}_{\sigma(i')}\|_2 \tag{7}$$

For a given $\tau \in (0, 1)$ define $i_\tau(\mathbf{c}) := \min\{i \in [M] \text{ s.t. } \|\mathbf{c}_{\sigma(i)}\|_2^2 \leq \tau \sum_{i' \geq i} \|\mathbf{c}_{\sigma(i')}\|_2^2\}$ be the $\tau$-critical index.

Define the following: $C_\tau(h) := \{\sigma(i) \mid i < i_\tau(\mathbf{c}), i \in [M]\}$, $C_\tau^{\leq K}(h) := \{\sigma(i) \mid i \leq K, i < i_\tau\}$, and $C_\tau^{\mathrm{reg}}(h) := [M] \setminus C_\tau(h)$.

Now, the sequence of coefficients corresponding to $C_\tau^{\mathrm{reg}}(h)$ are $\tau$-regular, since for any $i \in C_\tau^{\mathrm{reg}}(h)$, $\|\mathbf{c}_i\|_2^2 \leq \tau \sum_{i' \in C_\tau^{\mathrm{reg}}(h)} \|\mathbf{c}_{i'}\|_2^2$. This regular part of $\mathbf{c}$ is for convenience denoted by the following notation: $\mathbf{c}^{\mathrm{reg}} := \oplus_{i \in C_\tau^{\mathrm{reg}}(h)} \mathbf{c}_i$.

On the other hand, the coefficients corresponding to the non-regular part $C_\tau(h)$ are essentially geometrically decreasing. This is formalized in the following proposition.

**Proposition B.7** ([7]). *For the above setting the following condition hold:*

- *For any $i_1, i_2 \in C_\tau(h)$, s.t. $t_1 := \sigma^{-1}(i_1)$, $t_2 := \sigma^{-1}(i_2)$, $t_1 < t_2$,*

$$\|\mathbf{c}_{i_2}\|^2 \leq \frac{1}{\tau}(1-\tau)^{(t_2-t_1)}\|\mathbf{c}_{i_1}\|^2.$$

## B.2 Proof of Theorem 3.2

The proof of Theorem 3.2 is given in Appendix A of [3], in particular the intermediate instance $\mathcal{B}$ obtained in their proof is the hard instance of Theorem 3.2.

## B.3 Representing Boolean Functions

Finally, we have this simple lemma on the representation of boolean functions.

**Lemma B.8.** *Any boolean valued function $f$ over boolean variables $y_1, \ldots, y_\ell$ can be written as*

$$f(y_1, \ldots, y_\ell) = \sum_{H \subseteq [\ell]} \left( a_H \prod_{s \in H} y_s \right), \tag{8}$$

*where each $|a_H| \leq 2^\ell$ for each $H \subseteq [\ell]$.*

*Proof.* Using the truth table of $f$, it can be written as an OR of up to $2^\ell$ terms, one for each point at which $f$ evaluates to 1. Each such term is an AND of exactly $\ell$ literals which evaluates to 1 on a distinct point on which $f$ evaluates to 1. Any such term can be evaluated as a product of: $y$ for each positive literal $y$ in it, and $(1 - y)$ for each negated literal in the term. Distributing this product yields for each term an expression of the form on the RHS of (8) with coefficients of magnitude 1. Summing up the expressions for all the (up to $2^\ell$) terms gives us the expression on the RHS of (8) for $f$, with coefficients of magnitude at most $2^\ell$. □

# C Hardness Reduction and Proof of Theorem 3.3

Let $\delta, \ell$ be the constants in the statement of Theorem 3.3. We choose the following parameters for the reduction:

$$\varepsilon := \frac{\varepsilon_0}{10^{10\ell}} \left( \frac{\delta}{8\ell} \right)^{32}, Q := \frac{64}{\varepsilon^4}, \quad \tau := \varepsilon_0 \frac{\varepsilon^8}{Q^8}, \quad K := \frac{32}{\tau} \log \left( \frac{8Q}{\tau} \right), \tag{9}$$

where $\varepsilon_0 \in (0, 1)$ is a small enough absolute constant to be decided later. The hardness reduction is from an instance $\mathcal{L}$ of **Smooth-Label-Cover** given by Thm. 3.2 for a value of $z$ (and thereby

$d := 4^z$) to be chosen later, and $J_{\mathrm{param}}$ (we denote $J$ from Thm. 3.2 as $J_{\mathrm{param}}$ to avoid notation conflict) as defined as follows:

$$J_{\mathrm{param}} := \frac{4}{\varepsilon^2}\left(K\ell + \frac{16d^8\ell}{\tau^6}\right)^2. \tag{10}$$

We now define the set of boolean variables (feature space) over which the LLP-OR instance is defined.

**Variables.** Let

$$\bigcup_{i=1}^{M}\bigcup_{b\in\{0,1\}}\bigcup_{q=1}^{Q}\{X_{i,b,q}\} \tag{11}$$

be a set of variables. We use the following notations: $\mathbf{X}_{i,b} := (X_{i,b,1}, \dots, X_{i,b,Q})$, $\mathbf{X}_i = \mathbf{X}_{i,0} \oplus \mathbf{X}_{i,1}$ and $\mathbf{X} = \oplus_{i=1}^{M}\mathbf{X}_i$, for all $i \in [M], b \in \{0,1\}$. Further, let $S_1, \dots, S_m$ be a partition of $[M]$ such that $|S_j| \leq d$ for all $j \in [m]$. For each $j \in [m]$, let $\mathbf{X}^{(j)} = \oplus_{i\in S_j}\mathbf{X}_i$ (in any order). For coefficients $\{c_{i,b,q} \mid i \in [M], b \in \{0,1\}, q \in [Q]\}$, we analogously have $\mathbf{c}_{i,b}$, $\mathbf{c}_i$, $\mathbf{c}^{(j)}$ and $\mathbf{c}$.

For each vertex $v \in V_{\mathcal{L}}$ there is a separate copy $\mathbf{X}_v$ of $\mathbf{X}$ defined above. In particular, the total set of variables is

$$\bigcup_{v\in V_{\mathcal{L}}}\bigcup_{i=1}^{M}\bigcup_{b\in\{0,1\}}\bigcup_{q=1}^{Q}\{\overline{X}_{v,i,b,q}\}.$$

represented by the $(|V_{\mathcal{L}}| \cdot M \cdot 2 \cdot Q)$-dimensional boolean vector $\overline{\mathbf{X}}$. The goal is now to define a distribution over non-monochromatic bags of size at most two.

For this we first define a distribution $\mathcal{D}_X$ as given in Fig. 1 on $\mathbf{X}$ given in (11), parameterized by a partition $S_1, S_2, \dots, S_m$ of $[M]$. We use as building blocks the two distributions over $\{0,1\}^Q$: $\overline{\mathcal{U}}_Q := U(\{\mathbf{e}_1, \dots, \mathbf{e}_Q\})$ (i.e., uniform over the $Q$-dim coordinate vectors) and $\widehat{\mathcal{U}}_Q := U(\{0,1\}^Q)$ (uniform over $Q$-dim boolean vectors).

---

1. Randomly choose $J \subseteq [m]$ by sampling each $j \in [m]$ independently w.p. $1/2$.

2. Randomly choose $J_\varepsilon \subseteq J$ by sampling each $j \in J$ independently w.p. $\varepsilon$.

3. Independently for each $i \in [M]$ sample $b_i \in \{0,1\}$ u.a.r. Set $a_i = 1 - b_i$ for $i \in [M]$.

4. Independently for each $i \in \cup_{j\in J}S_j$, sample $\mathbf{X}_{i,a_i}$ from $\overline{\mathcal{U}}_Q$.

5. Independently for each $i \in \cup_{j\in J_\varepsilon}S_j$, sample $\mathbf{X}_{i,b_i}$ from $\widehat{\mathcal{U}}_Q$.

6. For $i \in \cup_{j\in J\setminus J_\varepsilon}S_j$, set $\mathbf{X}_{i,b_i}$ to $\mathbf{0}_Q$.

7. For all $j \notin J$, set all $\mathbf{X}_i$ ($i \in S_j$) to $\mathbf{0}_{2Q}$.

---

Figure 1: Distribution $\mathcal{D}_X\left(\{S_j\}_{j=1}^{m}\right)$ with partition $\{S_j\}_{j=1}^{m}$ of $[M]$.

For a given $u \in U_{\mathcal{L}}$ and $(v,u),(w,u) \in E_{\mathcal{L}}$, using $\mathcal{D}_X$ we define in Fig. 2 a distribution $\mathcal{D}_{u,v,w}$ on a pair of points – over the entire variable set (11) – $(\overline{\mathbf{X}}^{(1)}, \overline{\mathbf{X}}^{(2)})$. The final non-monochromatic bag distribution $\mathcal{D}_B$ is given in Fig. 3.

## C.1 Proof of Completeness of Theorem 3.3

Suppose that the instance $\mathcal{L}$ is a YES instance i.e., there is a labeling $\sigma = (\sigma_V, \sigma_U)$ that satisfies all the edges $E_{\mathcal{L}}$. Consider the monotone-OR $F^*$ given by:

$$F^*\left(\overline{\mathbf{X}}\right) = \bigvee_{v\in V_{\mathcal{L}}}\left(\bigvee_{b\in\{0,1\}}\left(\bigvee_{q=1}^{Q}\overline{X}_{v,\sigma_V(v),b,q}\right)\right). \tag{12}$$

The following lemma shows the completeness of the reduction.

1. Randomly choose $J^v \subseteq [m]$ by sampling each $j \in [m]$ independently w.p. $1/2$. Let $J^w = [m] \setminus J^v$.

2. Setting $S_j = \pi_{v,u}^{-1}(j)$ for $j \in [m]$, sample $\overline{\mathbf{X}}_v^{(1)}$ from $\mathcal{D}_X\left(\{S_j\}_{j=1}^m\right) \mid J = J^v$.

3. Setting $S_j = \pi_{w,u}^{-1}(j)$ for $j \in [m]$, sample $\overline{\mathbf{X}}_w^{(2)}$ from $\mathcal{D}_X\left(\{S_j\}_{j=1}^m\right) \mid J = J^w$.

4. For all $v' \in V_{\mathcal{L}} \setminus \{v\}$ set all $\{\overline{X}_{v',i,b,q}^{(1)} \mid i \in [M], b \in \{0,1\}, q \in [Q]\}$ to zero.

5. For all $v' \in V_{\mathcal{L}} \setminus \{w\}$ set all $\{\overline{X}_{v',i,b,q}^{(2)} \mid i \in [M], b \in \{0,1\}, q \in [Q]\}$ to zero.

6. Output $(\overline{\mathbf{X}}^{(1)}, \overline{\mathbf{X}}^{(2)})$.

Figure 2: Distribution $\mathcal{D}_{u,v,w}$.

1. Choose $u \in U_{\mathcal{L}}$ u.a.r.

2. Choose $v, w$ independently and u.a.r from $N_{\mathcal{L}}(u) := \{v' \in V_{\mathcal{L}} \mid (v', u) \in E_{\mathcal{L}}\}$.

3. Sample $(\overline{\mathbf{X}}^{(1)}, \overline{\mathbf{X}}^{(2)})$ from $\mathcal{D}_{u,v,w}$.

4. Output the bag $B = \{\overline{\mathbf{X}}^{(1)}, \overline{\mathbf{X}}^{(2)}\}$.

Figure 3: Bag Distribution $\mathcal{D}_B$.

**Lemma C.1.** *For each bag $B = \{\overline{\mathbf{X}}^{(1)}, \overline{\mathbf{X}}^{(2)}\}$ in the support of $\mathcal{D}_B$ in Fig. 3,*
$$F^*\left(\overline{\mathbf{X}}^{(1)}\right) \neq F^*\left(\overline{\mathbf{X}}^{(2)}\right).$$

*Proof.* Let $j^* := \pi_{v,u}(\sigma_V(v)) = \pi_{w,u}(\sigma_V(w))$. Suppose that in the generation of $\{\overline{\mathbf{X}}^{(1)}, \overline{\mathbf{X}}^{(2)}\}$ from $\mathcal{D}_B$, $(\overline{\mathbf{X}}^{(1)}, \overline{\mathbf{X}}^{(2)})$ were sampled from $\mathcal{D}_{u,v,w}$.

In $\mathcal{D}_{u,v,w}$, it can be seen that if $j^* \in J^v$ then $\overline{\mathbf{X}}_w^{(2)}$ satisfies $\overline{\mathbf{X}}_{w,i}^{(2)} = \mathbf{0}_{2Q}$ for all $i \in \pi_{w,u}^{-1}(j^*)$, in particular $\overline{\mathbf{X}}_{w,\sigma_V(w)}^{(2)} = \mathbf{0}_{2Q}$. Further, $\overline{\mathbf{X}}_{v'}^{(2)} = \mathbf{0}$ for any $v' \neq w$. Thus, $F^*$ evaluates to 0 on $\overline{\mathbf{X}}^{(2)}$. On the other hand, from $\mathcal{D}_X$ we know that one of $\{\overline{\mathbf{X}}_{v,\sigma_V(v),0}^{(1)}, \overline{\mathbf{X}}_{v,\sigma_V(v),1}^{(1)}\}$ is sampled from $\overline{\mathcal{U}}_Q$, therefore at least one of $\{\overline{X}_{v,\sigma_V(v),b,q} \mid b \in \{0,1\}, q \in [Q]\}$ is set to 1 and therefore $F^*$ evaluates to 1 on $\overline{\mathbf{X}}^{(1)}$.

An analogous argument shows that when $j^* \in J^w = [m] \setminus J^v$, $F^*$ evaluates to 1 on $\overline{\mathbf{X}}^{(2)}$, and to 0 on $\overline{\mathbf{X}}^{(1)}$. $\qquad\square$

## C.2 Proof of Soundness of Theorem 3.3

In this case $\mathcal{L}$ is a NO instance. Consider $\ell$ linear forms
$$g_s(\overline{\mathbf{X}}) = c_{s,0} + \sum_{v \in V_{\mathcal{L}}} \sum_{i=1}^{M} \sum_{b \in \{0,1\}} \sum_{q=1}^{Q} c_{s,v,i,b,q} \overline{X}_{v,i,b,q}, \quad s \in [\ell]. \tag{13}$$

For an given vertex $v \in V_{\mathcal{L}}$, define
$$h_{s,v}(\overline{\mathbf{X}}_v) := c_{s,v,i,b,q} \overline{X}_{v,i,b,q}. \tag{14}$$

for each $s \in [\ell]$. Let us fix a triple $(u,v,w)$ as a choice of $\mathcal{D}_B$ in Fig. 3. For convenience we let $\pi_v := \pi_{v,u}$ and $\pi_w := \pi_{w,u}$. Using this along with the notation in Sec. B.1 define for each $s \in [\ell]$:
$$B_\tau(h_{s,v}) := \{i \in C_\tau^{\text{reg}}(h_{s,v}) \mid \|\mathbf{c}_{s,v,i}\|_2^2 > (\tau^6/(16d^8))\|\mathbf{c}_{s,v}^{\text{reg}}\|_2^2\}, \tag{15}$$
$$B_\tau(h_{s,w}) := \{i \in C_\tau^{\text{reg}}(h_{s,w}) \mid \|\mathbf{c}_{s,w,i}\|_2^2 > (\tau^6/(16d^8))\|\mathbf{c}_{s,w}^{\text{reg}}\|_2^2\}, \tag{16}$$

and

$$B_\tau(v) := \bigcup_{s \in \ell} B_\tau(h_{s,v}) \qquad\qquad B_\tau(w) := \bigcup_{s \in \ell} B_\tau(h_{s,w}) \qquad (17)$$

$$C_\tau^{\leq K}(v) := \bigcup_{s \in \ell} C_\tau^{\leq K}(h_{s,v}) \qquad\qquad C_\tau^{\leq K}(w) := \bigcup_{s \in \ell} C_\tau^{\leq K}(h_{s,w}). \qquad (18)$$

Further, for each $s \in [\ell]$

$$S_{svv} := \left\{ i \in C_\tau^{\text{reg}}(h_{s,v}) \mid \pi_v(i) \in \pi_v\left(C_\tau^{\leq K}(v)\right) \right\}, \quad S_{vv} = \bigcup_{s=1}^{\ell} S_{svv} \qquad (19)$$

$$S_{sww} := \left\{ i \in C_\tau^{\text{reg}}(h_{s,w}) \mid \pi_w(i) \in \pi_w\left(C_\tau^{\leq K}(w)\right) \right\}, \quad S_{ww} = \bigcup_{s=1}^{\ell} S_{sww} \qquad (20)$$

$$S_{svw} := \left\{ i \in C_\tau^{\text{reg}}(h_{s,v}) \mid \pi_v(i) \in \pi_w\left(C_\tau^{\leq K}(w)\right) \right\}, \quad S_{vw} = \bigcup_{s=1}^{\ell} S_{svw} \qquad (21)$$

$$S_{swv} := \left\{ i \in C_\tau^{\text{reg}}(h_{s,w}) \mid \pi_w(i) \in \pi_v\left(C_\tau^{\leq K}(v)\right) \right\}, \quad S_{wv} = \bigcup_{s=1}^{\ell} S_{swv}. \qquad (22)$$

Using the above we have the following definition.

**Definition C.2.** *A triple $(u, v, w)$ chosen in $\mathcal{D}_B$ (Fig. 3) is said to be* good *if the it satisfies the following conditions:*

1. Top-$K$ Top-reg bijective.

$$\left| \pi_v\left(C_\tau^{\leq K}(v) \cup B_\tau(v)\right) \right| = \left| C_\tau^{\leq K}(v) \cup B_\tau(v) \right|, \qquad (23)$$

$$\left| \pi_w\left(C_\tau^{\leq K}(w) \cup B_\tau(w)\right) \right| = \left| C_\tau^{\leq K}(w) \cup B_\tau(w) \right| \qquad (24)$$

2. Top-$K$ non-intersection.

$$\pi_v\left(C_\tau^{\leq K}(v)\right) \cap \pi_w\left(C_\tau^{\leq K}(w)\right) = \emptyset. \qquad (25)$$

3. Top-$K$ heavy-reg non-intersection. *For each $s \in [\ell]$,*

$$\sum_{i \in S_{svw}} \|\mathbf{c}_{s,v,i}\|_2^2 \leq (\tau^6/16)\|\mathbf{c}_{s,v}^{\text{reg}}\|_2^2, \quad \text{and} \quad \sum_{i \in S_{swv}} \|\mathbf{c}_{s,w,i}\|_2^2 \leq (\tau^6/16)\|\mathbf{c}_{s,w}^{\text{reg}}\|_2^2. \qquad (26)$$

First we prove the following lemma.

**Lemma C.3.** *There is a choice of $z$ depending only on $\delta, \ell$ such that the probability over the choice of $(u, v, w)$ in $\mathcal{D}_B$ that $(u, v, w)$ is good is at least $1 - \varepsilon$.*

*Proof.* The following arguments use the bi-regulariy of $\mathcal{L}$. Suppose for a contradiction that at least $\varepsilon$-fraction of the triples $(u, v, w)$ are not good. From the smoothness property of Theorem 3.2, a random edge $(v, u)$ incident on $v$ for a given $v \in V_\mathcal{L}$ satisfies (23) is at least:

$$1 - \frac{\left(|C_\tau^{\leq K}(v)| + |B_\tau(v)|\right)^2}{J_{\text{param}}} \geq 1 - \left(\ell K + \ell \frac{16d^8}{\tau^6}\right)^2 / J_{\text{param}} \geq 1 - \varepsilon^2/4,$$

since $|B_\tau(h_{s,v})| \leq 16d^8/\tau^6$ for any $s \in [\ell]$ and by (10). Thus, we obtain that $(u, v, w)$ satisfies the *Top-$K$ Top-reg bijective* property with probability at least, $1 - \varepsilon^2/2$.

It follows that there must be at least $\varepsilon - \varepsilon^2/2 \geq \varepsilon/2$ fraction of triples $(u, v, w)$ which do not satisfy either (25) or (26) - call such triples *intersecting*. Given this, consider the following randomized labeling $\sigma$ to the vertices of $\mathcal{L}$:

1. For every $v \in V_\mathcal{L}$:

(a) With probability $1/2$ choose $\sigma_V(v)$ u.a.r from $C_{\tau}^{\leq K}(v)$.

(b) With probability $1/2$:

i. Choose $s \in [\ell]$ u.a.r.

ii. Choose $\sigma_V(v)$ to be $i \in C_{\tau}^{\text{reg}}(h_{s,v})$ with probability $\|\mathbf{c}_{s,v,i}\|_2^2/\|\mathbf{c}_{s,v}^{\text{reg}}\|_2^2$.

2. For each $u \in U_{\mathcal{L}}$, choose a $w$ u.a.r. from $N(u)$ and assign $\sigma_U(u) = \pi_{w,u}(\sigma_V(w))$.

Consider a random edge $(v, u)$, and let $w$ be the choice for $u$ in the above randomized labeling. With probability $\varepsilon/2$, $(u, v, w)$ is intersecting. In this case, if (25) is violated then with probability $1/(4\ell^2 K^2)$, $\pi_{v,u}(\sigma_V(v)) = \pi_{w,u}(\sigma_V(w))$. On the other hand, if one of the conditions in (26) is not satisfied for some $s \in [\ell]$ then with probability $\tau^6/(64K\ell^2)$, $\pi_{v,u}(\sigma_V(v)) = \pi_{w,u}(\sigma_V(w))$. In both cases, $(v, u)$ is satisfied. Thus, there is a labeling that satisfies at least,

$$\frac{\varepsilon}{2} \cdot \min\left\{\frac{1}{4\ell^2 K^2}, \frac{\tau^6}{64K\ell^2}\right\}$$

fraction of the edges. Choosing $z$ in Theorem 3.2 to be large enough depending only on $\tau, K, \varepsilon, \ell$ (which in turn depend only on $\delta, \ell$ for fixed $\varepsilon_0$ as given in (9)) yields a contradiction. $\qquad\square$

Now, for any boolean function $f$ over $\mathsf{pos}(g_1), \ldots, \mathsf{pos}(g_\ell)$ we have the following lemma.

**Lemma C.4.** *For any good triple $(u, v, w)$, under the distribution $\mathcal{D}_{u,v,w}$,*

$$\left| \mathsf{E}\left[f(\overline{\mathbf{X}}^{(1)})f(\overline{\mathbf{X}}^{(2)})\right] - \mathsf{E}\left[f(\overline{\mathbf{X}}^{(1)})\right] \mathsf{E}\left[f(\overline{\mathbf{X}}^{(2)})\right] \right| \leq O\left(\ell^2 2^{4\ell} \varepsilon^{1/4}\right).$$

*Proof.* From Lemma B.8, $f\left(\mathsf{pos}(g_1), \ldots, \mathsf{pos}(g_\ell)\right)$ can be written as:

$$\sum_{H \subseteq [\ell]} \left(a_H \prod_{s \in H} \mathsf{pos}(g_s)\right) \tag{27}$$

where each $|a_H| \leq 2^\ell$. The above expression is the sum of at most $2^\ell$ product terms. Thus, $f(\overline{\mathbf{X}}^{(1)})f(\overline{\mathbf{X}}^{(2)})$ has at most $2^{2\ell}$ product terms of the form

$$\left(a_H \prod_{s \in H} \mathsf{pos}(g_s)\left(\overline{\mathbf{X}}^{(1)}\right)\right)\left(a_{H'} \prod_{s \in H'} \mathsf{pos}(g_s)\left(\overline{\mathbf{X}}^{(2)}\right)\right) \tag{28}$$

for some $H, H' \subseteq [\ell]$. By Lemma C.5 however, we know that for a good triple $(u, v, w)$

$$
\begin{aligned}
&\left| \mathsf{E}\left[a_H \prod_{s \in H} \mathsf{pos}(g_s)\left(\overline{\mathbf{X}}^{(1)}\right) a_{H'} \prod_{s \in H'} \mathsf{pos}(g_s)\left(\overline{\mathbf{X}}^{(2)}\right)\right] \right. \\
&\quad \left. - \mathsf{E}\left[a_H \prod_{s \in H} \mathsf{pos}(g_s)\left(\overline{\mathbf{X}}^{(1)}\right)\right] \mathsf{E}\left[a_{H'} \prod_{s \in H'} \mathsf{pos}(g_s)\left(\overline{\mathbf{X}}^{(2)}\right)\right] \right| \\
&\leq |a_H a_{H'}| \cdot O\left(\ell^2 \varepsilon^{1/4}\right)
\end{aligned} \tag{29}
$$

Since $|a_H a_{H'}| \leq 2^{2\ell}$, bounding the error separately for the $2^{2\ell}$ terms as above, we obtain the bound in the lemma. $\qquad\square$

Over all choices of triples made in $\mathcal{D}_B$, the probability that $f$ satisfies the non-monochromatic bag consisting of $\left(\overline{\mathbf{X}}^{(1)}, \overline{\mathbf{X}}^{(2)}\right)$ is,

$$
\begin{aligned}
&\mathsf{E}_{u,v,w}\mathsf{E}_{\mathcal{D}_{u,v,w}}\left[f\left(\overline{\mathbf{X}}^{(1)}\right)\left(1 - f\left(\overline{\mathbf{X}}^{(2)}\right)\right) + f\left(\overline{\mathbf{X}}^{(2)}\right)\left(1 - f\left(\overline{\mathbf{X}}^{(1)}\right)\right)\right] \\
=\ &\mathsf{E}_{u,v,w}\mathsf{E}_{\mathcal{D}_{u,v,w}}\left[f\left(\overline{\mathbf{X}}^{(1)}\right) + f\left(\overline{\mathbf{X}}^{(2)}\right) - 2f\left(\overline{\mathbf{X}}^{(1)}\right)f\left(\overline{\mathbf{X}}^{(2)}\right)\right]
\end{aligned} \tag{30}
$$

Since at least $(1 - \varepsilon)$-fraction of triples are good and using Lemma C.4, from the above we bound the probability by

$$\mathsf{E}_u \left[ \mathsf{E} \left[ f \left( \overline{\mathbf{X}}^{(1)} \right) \right] + \mathsf{E} \left[ f \left( \overline{\mathbf{X}}^{(2)} \right) \right] - 2\mathsf{E} \left[ f \left( \overline{\mathbf{X}}^{(1)} \right) \right] \mathsf{E} \left[ f \left( \overline{\mathbf{X}}^{(2)} \right) \right] \right] + O \left( \ell^2 2^{4\ell} \varepsilon^{1/4} + \varepsilon \right) \quad (31)$$

where the inner expectations are over $v, w, \mathcal{D}_{u,v,w}$. For a fixed $u$, since $v$ and $w$ are chosen independently from $N(u)$ we have

$$\mathsf{E} \left[ f \left( \overline{\mathbf{X}}^{(1)} \right) \right] = \mathsf{E} \left[ f \left( \overline{\mathbf{X}}^{(2)} \right) \right] = p_u,$$

Using which (31) becomes,

$$\mathsf{E}_u \left[ 2p_u - 2p_u^2 \right] + O \left( \ell^2 2^{4\ell} \varepsilon^{1/4} \right) \quad (32)$$

Observe that

$$p_u = \mathsf{E}_{v \in N(u)} \left[ f \left( \overline{\mathbf{X}}^{(1)} \right) \right],$$

where $\overline{\mathbf{X}}^{(1)}$ in the above is sampled according to $\mathcal{D}_{u,v,w}$. The choice of $w$ is immaterial since the marginal distribution of $\overline{\mathbf{X}}^{(1)}$ after fixing $u$ and $v$ is the same. Note also that $p_f := \mathsf{E}_u[p_u]$ is the bias of $f$ over the choice of a random bag from $\mathcal{D}_B$ and a uniformly random feature vector from $B$. Now,

$$\mathsf{E}_u \left[ 2p_u - 2p_u^2 \right] = 2p_f - 2\mathsf{E}_u \left[ p_u^2 \right] \leq 2p_f - 2 \left( \mathsf{E}_u \left[ p_u \right] \right)^2 = 2p_f(1 - p_f),$$

using $\mathsf{E}[X^2] \geq \left( \mathsf{E}[X] \right)^2$. Thus,

$$\Pr_{B \leftarrow \mathcal{D}_B} \left[ \mathsf{pos}(f) \text{ is non-monochromatic on } B \right] \leq 2p_f(1 - p_f) + O \left( \ell^2 2^{4\ell} \varepsilon^{1/4} \right). \quad (33)$$

Since $p_f \in [0, 1]$, the value on the RHS of the above is at most $1/2 + O \left( \ell^2 2^{4\ell} \varepsilon^{1/4} \right) \leq 1/2 + \delta$ by the setting of our parameters (and small enough choice of the absolute constant $\varepsilon_0$) in (9).

## C.3 Analysis for a good triple

In this subsection we fix a good triple $(u, v, w)$, and for convenience as before let $\pi_v := \pi_{v,u}$ and $\pi_w := \pi_{w,u}$. Note that $\overline{\mathbf{X}}^{(1)}$ and $\overline{\mathbf{X}}^{(2)}$ in $\mathcal{D}_{u,v,w}$ are supported only on the coordinates corresponding to $v$ and $w$ respectively. Therefore, in this section we will think of $\overline{\mathbf{X}}^{(1)}$ defined only over $\{X_{v,i,b,q}\}$ and $\overline{\mathbf{X}}^{(2)}$ defined only over $\{X_{w,i,b,q}\}$ and let $\mathcal{D}$ denote this distribution over $\overline{\mathbf{X}}^{(1)}$ and $\overline{\mathbf{X}}^{(2)}$. For convenience, we let $\mathcal{D}_{X^{(1)}}$ and $\mathcal{D}_{X^{(2)}}$ denote the respective marginals.

For each $s \in [\ell]$ we let $g_{s,v}$ and $h_{s,v}$ be the restrictions of $g_s$ and $h_s$ to the coordinates $\{X_{v,i,b,q}\}$ respectively and similarly define $g_{s,w}$ and $h_{s,w}$.

For any $H \subseteq [\ell]$ define:

$$f_{v,H}(\mathbf{X}_v) := \prod_{s \in H} \mathsf{pos}(g_{s,v}(\mathbf{X}_v)), \qquad f_{w,H}(\mathbf{X}_w) := \prod_{s \in H} \mathsf{pos}(g_{s,w}(\mathbf{X}_w)) \quad (34)$$

In the remainder of this subsection we prove the following lemma.

**Lemma C.5.** *For any $H, H' \subseteq [\ell]$*

$$\left| \mathsf{E}_{\mathcal{D}} \left[ f_{v,H} \left( \mathbf{X}^{(1)} \right) f_{w,H'} \left( \mathbf{X}^{(2)} \right) \right] - \mathsf{E}_{\mathcal{D}_{X_v}} \left[ f_{v,H} \left( \mathbf{X}^{(1)} \right) \right] \mathsf{E}_{\mathcal{D}_{X_w}} \left[ f_{w,H'} \left( \mathbf{X}^{(2)} \right) \right] \right|$$
$$\leq O(\ell^2 \varepsilon^{1/4}) \quad (35)$$

*Proof.* For a given $s \in [\ell]$ let us consider the distribution of of $h_{s,v}(\mathbf{X}^{(1)}) = \sum_{i \in [M]} \langle \mathbf{c}_{v,i}, \mathbf{X}_{v,i}^{(1)} \rangle$. We divide this sum into four disjoint parts as follows. Let

$$h_{s,v}^{(0)}(\mathbf{X}^{(1)}) := \sum_{i \in C_{\bar{\tau}}^{\leq K}(h_{s,v})} \langle \mathbf{c}_{s,v,i}, \mathbf{X}_{v,i}^{(1)} \rangle \tag{36}$$

$$h_{s,v}^{(1)}(\mathbf{X}^{(1)}) := \sum_{i \in C_{\tau}(h_{s,v}) \backslash C_{\bar{\tau}}^{\leq K}(h_v)} \langle \mathbf{c}_{s,v,i}, \mathbf{X}_{v,i}^{(1)} \rangle \tag{37}$$

$$h_{s,v}^{(2)}(\mathbf{X}^{(1)}) := \sum_{i \in S_{svv} \cup S_{svw}} \langle \mathbf{c}_{v,i}, \mathbf{X}_{s,v,i}^{(1)} \rangle \tag{38}$$

$$h_{s,v}^{(3)}(\mathbf{X}^{(1)}) := \sum_{i \in C_{\tau}^{\mathrm{reg}}(h_{s,v}) \backslash (S_{svv} \cup S_{svw})} \langle \mathbf{c}_{s,v,i}, \mathbf{X}_{v,i}^{(1)} \rangle \tag{39}$$

Similarly, we define:

$$h_{s,w}^{(0)}(\mathbf{X}^{(2)}) := \sum_{i \in C_{\bar{\tau}}^{\leq K}(h_{s,w})} \langle \mathbf{c}_{s,w,i}, \mathbf{X}_{w,i}^{(2)} \rangle \tag{40}$$

$$h_{s,w}^{(1)}(\mathbf{X}^{(2)}) := \sum_{i \in C_{\tau}(h_{s,w}) \backslash C_{\bar{\tau}}^{\leq K}(h_w)} \langle \mathbf{c}_{s,w,i}, \mathbf{X}_{w,i}^{(2)} \rangle \tag{41}$$

$$h_{s,w}^{(2)}(\mathbf{X}^{(2)}) := \sum_{i \in S_{sww} \cup S_{swv}} \langle \mathbf{c}_{s,w,i}, \mathbf{X}_{w,i}^{(2)} \rangle \tag{42}$$

$$h_{s,w}^{(3)}(\mathbf{X}^{(2)}) := \sum_{i \in C_{\tau}^{\mathrm{reg}}(h_{s,w}) \backslash (S_{sww} \cup S_{swv})} \langle \mathbf{c}_{s,w,i}, \mathbf{X}_{w,i}^{(2)} \rangle \tag{43}$$

We will first prove the following lemma.

**Lemma C.6.** *For any $s \in [\ell]$*

$$\Pr_{\mathcal{D}}\left[ \mathsf{pos}\left( g_{s,v}(\mathbf{X}^{(1)}) \right) \neq \mathsf{pos}\left( h_{s,v}^{(0)}(\mathbf{X}^{(1)}) + h_{s,v}^{(3)}(\mathbf{X}^{(1)}) + c_{s,0} \right) \right] \leq O(\varepsilon^{1/4}) \tag{44}$$

*and similarly,*

$$\Pr_{\mathcal{D}}\left[ \mathsf{pos}\left( g_{s,w}(\mathbf{X}^{(2)}) \right) \neq \mathsf{pos}\left( h_{s,w}^{(0)}(\mathbf{X}^{(2)}) + h_{s,w}^{(3)}(\mathbf{X}^{(2)}) + c_{s,0} \right) \right] \leq O(\varepsilon^{1/4}) \tag{45}$$

*Proof.* We will prove (44), with (45) following analogously. We fix any $s \in [\ell]$. Lemma C.8 shows that,

$$\Pr\left[ \mathsf{pos}(g_{s,v}) \neq \mathsf{pos}\left( h_{s,v}^{(0)} + h_{s,v}^{(2)} + h_{s,v}^{(3)} + c_{s,0} \right) \right] \leq O(\varepsilon^{1/4}) \tag{46}$$

by our setting of the parameters and a small enough choice of $\varepsilon_0$ in (9). Let us now bound contribution of $h_{s,v}^{(2)}$. Note that by (15) and (23), each $i \in S_{svv}$ satisfies $\|\mathbf{c}_{v,i}\|_2 \leq \left( \tau^3/(2d^4) \right) \|\mathbf{c}_{s,v}^{\mathrm{reg}}\|_2$. Furthermore, by definition $|S_{svv}| \leq Kd\ell$. Thus,

$$\left| \sum_{i \in S_{svv}} \langle \mathbf{c}_{s,v,i}, \mathbf{X}_{v,i}^{(1)} \rangle \right| \leq \sum_{i \in S_{svv}} \|\mathbf{c}_{s,v,i}\|_1 \;\; \leq \;\; \sqrt{2Q} \sum_{i \in S_{svv}} \|\mathbf{c}_{s,v,i}\|_2$$

$$\leq \;\; \left( Kd\ell\sqrt{2Q} \right) \left( \frac{\tau^3}{2d^4} \right) \|\mathbf{c}_{s,v}^{\mathrm{reg}}\|_2$$

$$\leq \;\; (\tau^{3/2}/4)\|\mathbf{c}_{s,v}^{\mathrm{reg}}\|_2 \tag{47}$$

by our setting of the parameters and a small enough choice of $\varepsilon_0$ in (9). The above also implies (using the fact that $\|\mathbf{x}\|_2 \leq \|\mathbf{x}\|_1$),

$$\sum_{i \in S_{svv}} \|\mathbf{c}_{s,v,i}\|_2^2 \leq \sum_{i \in S_{svv}} \|\mathbf{c}_{s,v,i}\|_1^2 \leq \left( \sum_{i \in S_{svv}} \|\mathbf{c}_{s,v,i}\|_1 \right)^2 \leq (\tau^3/16)\|\mathbf{c}_{s,v}^{\mathrm{reg}}\|_2^2 \tag{48}$$

Now consider any $j \in \pi_v(S_{svw})$, and let $i^* := \text{argmax}\{\|\mathbf{c}_{s,v,i}\|_2 \mid i \in S_{svw}, \pi_v(i) = j\}$. By (26) $\|\mathbf{c}_{s,v,i^*}\|_2 \leq (\tau^3/4)\|\mathbf{c}_{s,v}^{\text{reg}}\|_2$ and by (15) for any $i \in S_{svw} \cap \pi_v^{-1}(j)$, $i \neq i^*$, $\|\mathbf{c}_{s,v,i}\|_2 \leq (\tau^3/(2d^4))\|\mathbf{c}_{s,v}^{\text{reg}}\|_2$.

Therefore, for a given $j \in \pi_v(S_{svw})$,

$$\sum_{i \in S_{svw} \cap \pi_v^{-1}(j)} \|\mathbf{c}_{s,v,i}\|_1 \leq \sqrt{2Q} \sum_{i \in S_{svw} \cap \pi_v^{-1}(j)} \|\mathbf{c}_{s,v,i}\|_2 \quad \leq \quad \sqrt{2Q}\left((\tau^3/4) + d(\tau^3/(4d^4))\right)\|\mathbf{c}_{s,v}^{\text{reg}}\|_2$$

$$\leq \quad \tau^3\sqrt{Q}\|\mathbf{c}_{s,v}^{\text{reg}}\|_2 \tag{49}$$

Thus,

$$\left|\sum_{i \in S_{svw}} \langle \mathbf{c}_{s,v,i}, \mathbf{X}_{v,i}^{(1)} \rangle\right| \leq \sum_{j \in \pi_v(S_{svw})} \sum_{i \in S_{svw} \cap \pi_v^{-1}(j)} \|\mathbf{c}_{s,v,i}\|_1 \quad \leq \quad \tau^3 K\ell\sqrt{Q}\|\mathbf{c}_{s,v}^{\text{reg}}\|_2$$

$$\leq \quad (\tau^{3/2}/4)\|\mathbf{c}_{s,v}^{\text{reg}}\|_2 \tag{50}$$

using (49) and the size bound $|\pi_v(S_{svw})| \leq |\pi_w(C_{\neq}^{\leq K}(w))| \leq K\ell$, and by a small enough choice of $\varepsilon_0$ in (9). The derivation of (50) also implies,

$$\sum_{i \in S_{svw}} \|\mathbf{c}_{s,v,i}\|_2^2 \quad \leq \quad \sum_{i \in S_{svw}} \|\mathbf{c}_{s,v,i}\|_1^2$$

$$\leq \quad \left(\sum_{j \in \pi_v(S_{svw})} \sum_{i \in S_{svw} \cap \pi_v^{-1}(j)} \|\mathbf{c}_{s,v,i}\|_1\right)^2 \leq (\tau^3/16)\|\mathbf{c}_{s,v}^{\text{reg}}\|_2^2 \tag{51}$$

First, using (47), (50)

$$\left|h_{s,v}^{(2)}(\mathbf{X}^{(1)})\right| \leq (\tau^{3/2}/2)\|\mathbf{c}_v^{\text{reg}}\|_2, \tag{52}$$

and (48) and (51) also yield the following bound on the sum $\ell_2^2$ masses,

$$\sum_{i \in S_{svv} \cup S_{svw}} \|\mathbf{c}_{s,v,i}\|_2^2 \leq (\tau^3/8)\|\mathbf{c}_v^{\text{reg}}\|_2^2 \tag{53}$$

Thus,

$$\gamma := \sum_{i \in C_\tau^{\text{reg}}(h_{s,v}) \setminus (S_{svv} \cup S_{svw})} \|\mathbf{c}_{s,v,i}\|_2^2 \quad = \quad \|\mathbf{c}_{s,v}^{\text{reg}}\|_2^2 - \sum_{i \in S_{svv} \cup S_{svw}} \|\mathbf{c}_{s,v,i}\|_2^2$$

$$\geq \quad \left(1 - \tau^3/8\right)\|\mathbf{c}_{s,v}^{\text{reg}}\|_2^2$$

$$\geq \quad (1/2)\|\mathbf{c}_{s,v}^{\text{reg}}\|_2^2. \tag{54}$$

Since $\{\|\mathbf{c}_{s,v,i}\|_2 \mid i \in C_\tau^{\text{reg}}(h_{s,v})\}$ is $\tau$-regular by definition (refer to Sec. B.1), the above (and an analogous analysis for $h_{s,w}$) implies the following lemma.

**Lemma C.7.** *The following sequences of coefficients – of $h_{s,v}^{(3)}$ and $h_{s,w}^{(3)}$ respectively – are $(2\tau)$-regular:*

$$\{\|\mathbf{c}_{s,v,i}\|_2 \mid i \in C_\tau^{\text{reg}}(h_{s,v}) \setminus (S_{svv} \cup S_{svw})\},$$
$$\{\|\mathbf{c}_{s,w,i}\|_2 \mid i \in C_\tau^{\text{reg}}(h_{s,w}) \setminus (S_{sww} \cup S_{swv})\},$$

*for $s \in [\ell]$.*

Note that $h_{s,v}^{(0)}$ and $h_{s,v}^{(2)}$ depend only on $\mathbf{X}_{v,i}^{(1)}$ satisfying $\pi_v(i) \in \pi_v\left(C_{\neq}^{\leq K}(v)\right) \cup \pi_w\left(C_{\neq}^{\leq K}(w)\right)$, while $h_{s,v}^{(3)}$ depends only on those for which $\pi_v(i) \notin \pi_v\left(C_{\neq}^{\leq K}(v)\right) \cup \pi_w\left(C_{\neq}^{\leq K}(w)\right)$. Therefore, $h_{s,v}^{(0)} + h_{s,v}^{(2)}$ is independent of $h_{s,v}^{(3)}$.

From a small enough choice of $\varepsilon_0$ in (9), Lemma D.1 along with (79) and Lemma D.3 implies that for any $t$,

$$\Pr\left[h_{s,v}^{(3)}(\mathbf{X}^{(1)}) \in [t, t+\theta]\right] \leq \theta((\varepsilon/16)\gamma)^{-1/2} + O\left(\varepsilon^{1/4}\right), \tag{55}$$

where the probability is taken only on the setting of the variables $\mathbf{X}_i$ on which $h^{(3)}$ depends. Letting $t$ denote the independent fixation of $h_{s,v}^{(0)} + h_{s,v}^{(2)} + c_{s,0}$,

$$
\begin{aligned}
\delta_2 \quad &:= \quad \Pr\left[\mathsf{pos}\left(h_{s,v}^{(0)} + h_{s,v}^{(2)} + h_{s,v}^{(3)} + c_{s,0}\right) \neq \mathsf{pos}\left(h_{s,v}^{(0)} + h_{s,v}^{(3)} + c_{s,0}\right)\right] \\
&\leq \quad \Pr\left[h_{s,v}^{(3)} \in \left[t - \left|h_{s,v}^{(2)}\right|, t + \left|h_{s,v}^{(2)}\right|\right]\right] \\
\text{(using (55) and (52))} \quad &\leq \quad \frac{\tau^{3/2}\|\mathbf{c}_{s,v}^{\mathrm{reg}}\|_2}{\sqrt{(\varepsilon/16)\gamma}} + O\left(\varepsilon^{1/4}\right) \\
\text{(using (54))} \quad &\leq \quad 4\tau^{3/2}\sqrt{1/\varepsilon} + O\left(\varepsilon^{1/4}\right) = O\left(\varepsilon^{1/4}\right)
\end{aligned}
\tag{56}
$$

by our setting of parameters and small enough $\varepsilon_0$ in (9). The above along with (46) implies the lemma. $\qquad\square$

Define:

$$
\tilde{f}_{v,H}(\mathbf{X}_v) := \prod_{s\in H}\mathsf{pos}(\tilde{g}_{s,v}(\mathbf{X}_v)), \qquad \tilde{f}_{w,H}(\mathbf{X}_w) := \prod_{s\in H}\mathsf{pos}(\tilde{g}_{s,w}(\mathbf{X}_w))
\tag{57}
$$

where

$$
\tilde{g}_{s,v}(\mathbf{X}_v) = h_{s,v}^{(0)}(\mathbf{X}_v) + h_{s,v}^{(3)}(\mathbf{X}_v) + c_{s,0}, \qquad \tilde{g}_{s,w}(\mathbf{X}_w) = h_{s,w}^{(0)}(\mathbf{X}_w) + h_{s,w}^{(3)}(\mathbf{X}_w) + c_{s,0}
\tag{58}
$$

for all $s \in [\ell]$. Applying a union bound along with Lemma C.6 we obtain that for any $H, H' \subseteq [\ell]$

$$
\left|\mathsf{E}_{\mathcal{D}}\left[f_{v,H}\left(\mathbf{X}^{(1)}\right)f_{w,H'}\left(\mathbf{X}^{(2)}\right)\right] - \mathsf{E}_{\mathcal{D}}\left[\tilde{f}_{v,H}\left(\mathbf{X}^{(1)}\right)\tilde{f}_{w,H'}\left(\mathbf{X}^{(2)}\right)\right]\right| \leq O(\ell\varepsilon^{1/4})
\tag{59}
$$

Observe that $\{h_{s,v}^{(0)}\}_{s=1}^{\ell}$ depends only on $\mathbf{X}_{v,i}$ for $\pi_v(i) \in \pi_v(C_{\bar{\tau}}^{\leq K}(v))$, and $\{h_{s,w}^{(0)}\}_{s=1}^{\ell}$ depends only on $\mathbf{X}_{w,i}$ for $\pi_w(i) \in \pi_w(C_{\bar{\tau}}^{\leq K}(w))$. By (25) $\pi_v(C_{\bar{\tau}}^{\leq K}(v))$ and $\pi_w(C_{\bar{\tau}}^{\leq K}(w))$ are disjoint. Further, $\{h_{s,v}^{(3)}\}_{s=1}^{\ell}$ depend only on $\mathbf{X}_{v,i}$ such that $\pi_v(i) \notin \pi_v(C_{\bar{\tau}}^{\leq K}(v)) \cup \pi_w(C_{\bar{\tau}}^{\leq K}(w))$, and $\{h_{s,w}^{(3)}\}_{s=1}^{\ell}$ depend only on $\mathbf{X}_{w,i}$ such that $\pi_w(i) \notin \pi_v(C_{\bar{\tau}}^{\leq K}(v)) \cup \pi_w(C_{\bar{\tau}}^{\leq K}(w))$. Thus, we have

1. $\{h_{s,v}^{(0)} = t_{s,v}\}_{s=1}^{\ell}$ is fixed independently of $\{h_{s,v}^{(3)}\}_{s=1}^{\ell}, \{h_{s,w}^{(0)}\}_{s=1}^{\ell}, \{h_{s,w}^{(3)}\}_{s=1}^{\ell}$, by sampling variables corresponding to $j \in \pi_v(C_{\bar{\tau}}^{\leq K}(v))$. This fixes $J^v \cap \pi_v(C_{\bar{\tau}}^{\leq K}(v))$ independent of $J^w \cap \pi_w(C_{\bar{\tau}}^{\leq K}(w))$.

2. $\{h_{s,w}^{(0)} = t_{s,w}\}_{s=1}^{\ell}$ is fixed independently of $\{h_{s,w}^{(3)}\}_{s=1}^{\ell}, \{h_{s,v}^{(0)}\}_{s=1}^{\ell}, \{h_{s,v}^{(3)}\}_{s=1}^{\ell}$, by sampling variables corresponding to $j \in \pi_w(C_{\bar{\tau}}^{\leq K}(w))$. This fixes $J^w \cap \pi_w(C_{\bar{\tau}}^{\leq K}(w))$ independent of $J^v \cap \pi_v(C_{\bar{\tau}}^{\leq K}(v))$

For now we assume the above fixations. Letting $\tilde{J}^v, \tilde{J}^w$ be the restrictions of $J^v, J^w$ to $[m] \setminus \left(\pi_v(C_{\bar{\tau}}^{\leq K}(v)) \cup \pi_w(C_{\bar{\tau}}^{\leq K}(w))\right)$,

$$
\begin{aligned}
&\mathsf{E}_{\mathcal{D}}\left[\tilde{f}_{v,H}\left(\mathbf{X}^{(1)}\right)\tilde{f}_{w,H'}\left(\mathbf{X}^{(2)}\right)\right] \\
= \quad &\mathsf{E}_{\tilde{J}^v, \tilde{J}^w}\left[\mathsf{E}\left[\left(\prod_{s\in H}\mathsf{pos}(t_{s,v} + h_{s,v}^{(3)}(\mathbf{X}^{(1)}) + c_{s,0})\right)\right.\right. \\
&\hspace{4cm}\left.\left.\left(\prod_{s\in H'}\mathsf{pos}(t_{s,w} + h_{s,w}^{(3)}(\mathbf{X}^{(2)}) + c_{s,0})\right) \mid \tilde{J}^v, \tilde{J}^w\right]\right] \\
= \quad &\mathsf{E}_{\tilde{J}^v, \tilde{J}^w}\left[\mathsf{E}\left[\left(\prod_{s\in H}\mathsf{pos}(t_{s,v} + h_{s,v}^{(3)}(\mathbf{X}^{(1)}) + c_{s,0})\right) \mid \tilde{J}^v\right]\right. \\
&\hspace{4cm}\left.\mathsf{E}\left[\left(\prod_{s\in H'}\mathsf{pos}(t_{s,w} + h_{s,w}^{(3)}(\mathbf{X}^{(2)}) + c_{s,0})\right) \mid \tilde{J}^w\right]\right]
\end{aligned}
\tag{60}
$$

since $\{h_{s,v}^{(3)}(\mathbf{X}^{(1)})\}_{s=1}^{\ell}$ are independent of $\mathbf{X}^{(2)}$ once $\tilde{J}^v$ is fixed. As observed earlier in this subsection $\{h_{s,v}^{(3)}(\mathbf{X}^{(1)})\}_{s=1}^{\ell}$ satisfy the regularity conditions in Section D.1 with parameter $2\tau$. With the setting of the parameters we have, and applying Lemma D.11 we obtain that the value of

$$\mathsf{E}\left[\left(\prod_{s\in H'}\mathsf{pos}(t_{s,w}+h_{s,w}^{(3)}(\mathbf{X}^{(2)})+c_{s,0})\right)\mid \tilde{J}^w\right]$$

is within $O(\ell\varepsilon^{1/4})$ of a fixed quantity, for all but $O(\ell^2\sqrt{\varepsilon})$ fraction of the choices of $\tilde{J}^w$. Since the pos function takes values $\{0,1\}$ we can decouple the expectation of products into product of expectations bounding the error as follows.

$$\left|\mathsf{E}_{\tilde{J}^v,\tilde{J}^w}\left[\mathsf{E}\left[\left(\prod_{s\in H}\mathsf{pos}(t_{s,v}+h_{s,v}^{(3)}(\mathbf{X}^{(1)})+c_{s,0})\right)\mid \tilde{J}^v\right]\right.\right.$$
$$\mathsf{E}\left[\left(\prod_{s\in H'}\mathsf{pos}(t_{s,w}+h_{s,w}^{(3)}(\mathbf{X}^{(2)})+c_{s,0})\right)\mid \tilde{J}^w\right]\right]-$$
$$\left.\mathsf{E}_{\tilde{J}^v}\left[\prod_{s\in H}\mathsf{pos}(t_{s,v}+h_{s,v}^{(3)}(\mathbf{X}^{(1)})+c_{s,0})\right]\cdot\mathsf{E}_{\tilde{J}^w}\left[\prod_{s\in H'}\mathsf{pos}(t_{s,w}+h_{s,w}^{(3)}(\mathbf{X}^{(2)})+c_{s,0})\right]\right|$$
$$\leq \quad O\left(\ell\varepsilon^{1/4}+\ell^2\sqrt{\varepsilon}\right)=O\left(\left(\ell\varepsilon^{1/4}\right)\right) \tag{61}$$

by our setting of $\varepsilon$ and small enough choice of $\varepsilon_0$ in (9).

Randomizing over the fixation of variables corresponding to $j\in\pi_v(C_{\tau}^{\leq K}(v))\cup\pi_w(C_{\tau}^{\leq K}(w))$ and using the disjointness of $\pi_v(C_{\tau}^{\leq K}(v))$ and $\pi_w(C_{\tau}^{\leq K}(w))$ allows us to extend the independently sampled $\tilde{J}^v$ and $\tilde{J}^w$ above to independently sampled $J^v$ and $J^w$ in the product of expectations in the LHS of (61). Combining this with bound in (61) and with (60) yields,

$$\left|\mathsf{E}_{\mathcal{D}}\left[\tilde{f}_{v,H}\left(\mathbf{X}^{(1)}\right)\tilde{f}_{w,H'}\left(\mathbf{X}^{(2)}\right)\right]-\mathsf{E}_{\mathcal{D}}\left[\tilde{f}_{v,H}\left(\mathbf{X}^{(1)}\right)\right]\mathsf{E}_{\mathcal{D}}\left[\tilde{f}_{w,H'}\left(\mathbf{X}^{(2)}\right)\right]\right|$$
$$\leq \quad O\left(\ell\varepsilon^{1/4}\right). \tag{62}$$

Using the definitions in (57), the above combined with (59), Lemma C.6 along with a union bound for the product of expectations above (similar to (59)) completes the proof of Lemma C.5. $\square$

## C.4 Critical Index Truncation

In this subsection we consider the distribution given by $\mathcal{D}_X$ (Fig. 1) over $\{X_{i,b,q}\mid i\in[M],b\in\{0,1\},q\in[Q]\}$, with $\{S_j\}_{j=1}^m$ being the partition of $[M]$. For a given linear form $h$ with coefficients $\{c_{i,b,q}\mid i\in[M],b\in\{0,1\},q\in[Q]\}$, let $\tilde{h}$ be a linear form whose coefficient vector $\tilde{\mathbf{c}}$ is given by:

$$\tilde{\mathbf{c}}_i=\begin{cases}\mathbf{0}_{2Q} & \text{if } i\in C_{\tau}(h)\setminus C_{\tau}^{\leq K}(h)\\ \mathbf{c}_i & \text{otherwise.}\end{cases} \tag{63}$$

Note that $C_{\tau}(\tilde{h})=C_{\tau}^{\leq K}(\tilde{h})=C_{\tau}^{\leq K}(h)$. We have the following lemma.

**Lemma C.8.** *If* $\left|C_{\tau}^{\leq K}(h)\cap S_j\right|\leq 1$ *for each* $j\in[m]$ *then except with probability* $\exp(-K\varepsilon/64)$ *over the choice of* $J,J_{\varepsilon},\mathbf{b}:=(b_i)_{i\in[M]}$,

$$\left|\Pr\left[h(\mathbf{X})>t\mid J,J_{\varepsilon},\mathbf{b}\right]-\Pr\left[\tilde{h}(\mathbf{X})>t\mid J,J_{\varepsilon},\mathbf{b}\right]\right|\leq \exp(-K\varepsilon^2/64)+O\left(1/\sqrt{\varepsilon K}\right). \tag{64}$$

*Proof.* If $|C_{\tau}(h)|\leq K$ then $\tilde{\mathbf{c}}=\mathbf{c}$ and the lemma is clearly true. So we may assume that $|C_{\tau}(h)|>K$ and in particular, $C_{\tau}^{\leq K}(h)=K$. Firstly, observe that

$$\left|h(\mathbf{X})-\tilde{h}(\mathbf{X})\right|\leq \sum_{i\in C_{\tau}(h)\setminus C_{\tau}^{\leq K}(h)}\|\mathbf{c}_i\|_1 \tag{65}$$

Note that by Prop. B.7, letting $\sigma$ be the ordering used in Sec. B.1

$$\sum_{i \in C_\tau(h) \setminus C_{\overline{\tau}}^{\leq K}(h)} \|\mathbf{c}_i\|_1 \quad \leq \quad \sqrt{2Q} \sum_{K < r < i_\tau(\mathbf{c})} \|\mathbf{c}_{\sigma(r)}\|_2$$

$$\leq \quad \sqrt{\frac{2Q}{\tau}} \|\mathbf{c}_{\sigma(K/4)}\|_2 \sum_{i \geq 0} (\sqrt{1-\tau})^{3K/4+i}$$

$$\leq \quad \sqrt{\frac{2Q}{\tau}} \|\mathbf{c}_{\sigma(K/4)}\|_2 \left( 2 \cdot (\sqrt{1-\tau})^{3K/4} \right). \tag{66}$$

The above, with our choice of $K$ w.r.t to $\tau$ and small enough $\varepsilon_0$ in (9) yields,

$$\min\{\|\mathbf{c}_i\|_2 \mid i \in C_{\overline{\tau}}^{\leq K/4}(h)\} > \sqrt{K} \sum_{i \in C_\tau(h) \setminus C_{\overline{\tau}}^{\leq K}(h)} \|\mathbf{c}_i\|_1 \tag{67}$$

For each $i \in C_{\overline{\tau}}^{\leq K/4}(h)$, define $Z_i \in \{0,1\}$ as the indicator of the event that $\{i \in S_j$ s.t. $j \in J_\varepsilon$ and $\|\mathbf{c}_{i,b_i}\|_2^2 \geq (1/2)\|\mathbf{c}_i\|_2^2\}$. By the condition of the lemma, each $i \in C_{\overline{\tau}}^{\leq K/4}(h)$ belongs to $S_j$ for a distinct $j$. Thus, $\{Z_i \mid i \in C_{\overline{\tau}}^{\leq K/4}(h)\}$ are independent random variables. Independently for each such $i$ s.t. $i \in S_j$, $j \in J$ w.p. $1/2$. With a further probability of $\varepsilon$, $j \in J_\varepsilon$; and $b_i$ satisfies $\|\mathbf{c}_{i,b_i}\|_2^2 \geq (1/2)\|\mathbf{c}_i\|_2^2$ independently w.p. at least $1/2$. Thus, $\Pr[Z_i = 1] \geq \varepsilon/4$. Applying the Chernoff-Hoeffding (Thm. B.3) bound we obtain that

$$\Pr\left[ \sum_{i \in C_{\overline{\tau}}^{\leq K/4}(h)} Z_i < \frac{K\varepsilon}{32} \right] \leq 2 \cdot \exp\left( -\left(\frac{7K\varepsilon}{32}\right)^2 K^{-1} \right) \tag{68}$$

In particular, (using (9) and small enough choice of $\varepsilon_0$) except for probability

$$\exp(-K\varepsilon^2/64) \tag{69}$$

over the choice of $J, J_\varepsilon, \mathbf{b}$

$$\sum_{i \in C_{\overline{\tau}}^{\leq K/4}(h)} Z_i \geq \frac{K\varepsilon}{32}. \tag{70}$$

Fixing such a choice of $J, J_\varepsilon, \mathbf{b}$, and letting $K' := \sum_{i \in C_{\overline{\tau}}^{\leq K/4}(h)} Z_i$, by the definition of $Z_i$s and by (67)

$$\min\{|\mathbf{c}_{i,b_i}\|_2 \mid i \in C_{\overline{\tau}}^{\leq K/4}(h), Z_i = 1\} > \sqrt{K'} \sum_{i \in C_\tau(h) \setminus C_{\overline{\tau}}^{\leq K}(h)} \|\mathbf{c}_i\|_1 \tag{71}$$

Note that each $\{\mathbf{X}_{i,b_i} \mid i \in C_{\overline{\tau}}^{\leq K/4}(h), Z_i = 1\}$ are iid distributed as $\widehat{\mathcal{U}}_Q$ (uniformly over $\{0,1\}^Q$), and independently from all other variables which whose contribution to $h$ can be independently fixed such that it equals $\theta + t$ for some $\theta$. Thus, applying the Lemma B.4 to $\{\mathbf{c}_{i,b_i} \mid i \in C_{\overline{\tau}}^{\leq K/4}(h), Z_i = 1\}$ yields that over the random choice of $\{\mathbf{X}_{i,b_i} \mid i \in C_{\overline{\tau}}^{\leq K/4}(h), Z_i = 1\}$,

$$\Pr\left[ \mathbb{1}\{h(\mathbf{X}) > t\} \neq \mathbb{1}\{\tilde{h}(\mathbf{X}) > t\} \right]$$

$$\leq \quad \Pr\left[ \left| \sum_{i \in C_{\overline{\tau}}^{\leq K/4}(h) \mid Z_i = 1} \langle \mathbf{c}_{i,b_i}, \mathbf{X}_{i,b_i} \rangle + \theta \right| \leq \left| h(\mathbf{X}) - \tilde{h}(\mathbf{X}) \right| \right]$$

$$(\text{using (65)}) \quad \leq \quad \Pr\left[ \left| \sum_{i \in C_{\overline{\tau}}^{\leq K/4}(h) \mid Z_i = 1} \langle \mathbf{c}_{i,b_i}, \mathbf{X}_{i,b_i} \rangle + \theta \right| \leq \sum_{i \in C_\tau(h) \setminus C_{\overline{\tau}}^{\leq K}(h)} \|\mathbf{c}_i\|_1 \right]$$

$$(\text{using (71)}) \quad \leq \quad \Pr\left[ \left| \sum_{i \in C_{\overline{\tau}}^{\leq K/4}(h) \mid Z_i = 1} \langle \mathbf{c}_{i,b_i}, \mathbf{X}_{i,b_i} \rangle + \theta \right| \leq \frac{\min\{|\mathbf{c}_{i,b_i}\|_2 \mid i \in C_{\overline{\tau}}^{\leq K/4}(h), Z_i = 1\}}{\sqrt{K'}} \right]$$

$$\leq \quad O\left(1/\sqrt{K'}\right) \leq O\left(1/\sqrt{\varepsilon K}\right). \tag{72}$$

The above, along with (65) and collecting the error probabilities from the above and in (69), (70) completes the proof. $\qquad\square$

# D   Invariance for the regular parts of the linear forms

The results of this section are used in the proof of Lemma C.5 applied to $h^{(3)}_{s,v}$ and $h^{(3)}_{s,w}$ ($s \in [\ell]$) – for a good triple $(u, v, w)$ – whose coefficients are shown to be $2\tau$-regular (Lemma C.7).

## D.1   Invariance for single regular LTF

We abstract out the properties of $h^{(3)}_{s,v}$ (resp. $h^{(3)}_{s,w}$) implied by the $2\tau$-regularity as mentioned above and the Top-$K$ Top-reg bijective condition (Defn. C.2) with the same setting of parameters given in (9) as follows.

Let $h := \sum_{i=1}^{M} \sum_{b \in \{0,1\}} \sum_{q \in [Q]} c_{i,b,q} X_{i,b,q}$ be a homogeneous linear form. For a parameter $\tau$, we say that $h$ is $(\tau, d^8)$-*nice* if:

$$\|\mathbf{c}_i\|_2^2 \leq 2\tau \|\mathbf{c}\|_2^2 \qquad \qquad \forall i \in [M], \qquad (73)$$

$$\left| \{ i \in S_j \mid \|\mathbf{c}_i\|_2^2 > (\tau/(4d^8))\|\mathbf{c}\|_2^2 \} \right| \leq 1 \qquad \qquad \forall j \in [m]. \qquad (74)$$

For a parameter $\varepsilon$ to be decided later, consider the distribution $\mathcal{D}_X$ over $\{ X_{i,b,q} \mid i \in [M], b \in \{0,1\}, q \in [Q] \}$ as given in Figure 1. Let us denote by $\mathcal{D}_X[h]$ the distribution of $h(\mathbf{X})$ over $\mathbf{X} \leftarrow \mathcal{D}_X$, and by $\mathcal{D}_X[h|J]$ when $\mathbf{X} \leftarrow \mathcal{D}_X$ given $J$ For a given $j \in [m]$ let us define the random variable $h^{(j)} := \sum_{i \in S_j} \sum_{b \in \{0,1\}} \sum_{q \in [Q]} c_{i,b,q} X_{i,b,q}$. Let us consider the expectation and variance of $h^{(j)}$ conditioned on the event $\mathcal{E}_j = \{ j \in J \}$ in Fig. 1 denoted by

$$E_j = \mathsf{E}_{\mathcal{D}_X} \left[ h^{(j)} \mid \mathcal{E}_j \right], \qquad V_j = \mathrm{Var}_{\mathcal{D}_X} \left[ h^{(j)} \mid \mathcal{E}_j \right] \qquad (75)$$

Since each $j$ is included in $J$ independently w.p. $1/2$, it is easy to see that $E_j$ and $V_j$ are independent of the choice of $J$. Further, given $J$, $\{ h^{(j)} \mid j \in [m] \}$ are independent random variables. Thus, we have the following quantities:

$$E^{(J)} := \mathsf{E}_{\mathcal{D}_X} \left[ h \mid J \right] = \sum_{j=1}^{m} \mathbb{1}_{\mathcal{E}_j} E_j \qquad (76)$$

$$V^{(J)} := \mathrm{Var}_{\mathcal{D}_X} \left[ h \mid J \right] = \sum_{j=1}^{m} \mathbb{1}_{\mathcal{E}_j} V_j \qquad (77)$$

Further, over the choice of $J$ let

$$\tilde{V}_j := \mathrm{Var}_J \left[ \mathbb{1}_{\mathcal{E}_j} E_j \right] = \frac{E_j^2}{2} - \frac{E_j^2}{4} = \frac{E_j^2}{4} \qquad (78)$$

so that from (76) and the independence of $\{\mathcal{E}_j\}_{j=1}^{m}$ with $\mathsf{E}_J[\mathcal{E}_j] = 1/2$ we have

$$\mathsf{E}_J \left[ E^{(J)} \right] = \frac{1}{2} \sum_{j=1}^{m} E_j, \quad \mathsf{E}_J \left[ V^{(J)} \right] = \frac{1}{2} \sum_{j=1}^{m} V_j, \quad \text{and} \quad \mathrm{Var}_J \left[ E^{(J)} \right] = \sum_{j=1}^{m} \tilde{V}_j. \qquad (79)$$

We wish to prove the following lemma.

**Lemma D.1.** *With the choice of parameters and small enough $\varepsilon_0$ in (9), for a $(\tau, d^8)$-nice $h$, except with probability $O\left(\sqrt{\varepsilon}\right)$ over the choice of $J$ in Figure 1 the following holds:*

$$\left| \Pr\left[ \mathcal{G} > t \right] - \Pr_{\mathcal{D}_X} \left[ h(\mathbf{X}) > t \mid J \right] \right| \leq O\left( \varepsilon^{1/4} \right), \qquad (80)$$

*where $\mathcal{G} \sim N\left( \mathsf{E}_J \left[ E^{(J)} \right], \mathsf{E}_J \left[ V^{(J)} \right] \right)$. In particular, applying Gaussian anti-concentration (5),*

$$\Pr_{\mathcal{D}_X} \left[ h(\mathbf{X}) \in [t, t+\delta] \mid J \right] \leq \delta \left( \mathsf{E}_J \left[ V^{(J)} \right] \right)^{-1/2} + O\left( \varepsilon^{1/4} \right), \qquad (81)$$

*for any $t \in \mathbb{R}$ and $\delta \geq 0$.*

## D.2 Proof of Lemma D.1

The main idea of the proof is to use $(\tau, d^8)$-niceness of $h$ to argue that w.h.p over choice of $J$, $\mathcal{D}_X[h|J]$ is close to a fixed Gaussian distribution $\mathcal{G}$.

We first explicitly calculate $E_j$, $V_j$ and $\tilde{V}_j$ for a given $j \in J$.

If $j \notin J_\varepsilon$ then $E_j = \sum_{i \in S_j} \sum_{b \in \{0,1\}} \Pr[b_i = b](1/Q) \sum_{q=1}^{Q} c_{i,b,q} = (1/2Q) \sum_{i \in S_j} \sum_{b \in \{0,1\}} \sum_{q=1}^{Q} c_{i,b,q}$, otherwise there is an additional term of $\sum_{i \in S_j} \sum_{b \in \{0,1\}} \Pr[b_i = 1 - b](1/2) \sum_{q=1}^{Q} c_{i,b,q} = (1/4) \sum_{i \in S_j} \sum_{b \in \{0,1\}} \sum_{q=1}^{Q} c_{i,b,q}$. Therefore,

$$
\begin{aligned}
E_j &= (1 - \varepsilon)\left(\frac{1}{2Q}\right) \sum_{i \in S_j} \sum_{b \in \{0,1\}} \sum_{q=1}^{Q} c_{i,b,q} \\
&\quad + \varepsilon\left[\left(\frac{1}{2Q}\right) \sum_{i \in S_j} \sum_{b \in \{0,1\}} \sum_{q=1}^{Q} c_{i,b,q} + \left(\frac{1}{4}\right) \sum_{i \in S_j} \sum_{b \in \{0,1\}} \sum_{q=1}^{Q} c_{i,b,q}\right] \\
&= \left(\frac{\varepsilon}{4} + \frac{1}{2Q}\right) \Delta_j,
\end{aligned}
\tag{82}
$$

where $\Delta_j := \sum_{i \in S_j} \sum_{b \in \{0,1\}} \sum_{q=1}^{Q} c_{i,b,q}$. For $V_j$ we use apply the law of total variance on (75) over the random variable $\mathbb{1}_{\{j \in J_\varepsilon\}}$ as follows

$$
\begin{aligned}
V_j &= \mathrm{Var}\left[h^{(j)} \mid \mathcal{E}_j\right] \\
&= \mathrm{Var}\left[\mathsf{E}\left[h^{(j)} \mid \mathbb{1}_{\{j \in J_\varepsilon\}}\right] \mid \mathcal{E}_j\right] + \mathsf{E}\left[\mathrm{Var}\left[h^{(j)} \mid \mathbb{1}_{\{j \in J_\varepsilon\}}\right] \mid \mathcal{E}_j\right]
\end{aligned}
\tag{83}
$$

We note the both the terms on the RHS are non-negative. For the first term observe that given $j \in J$, $j \notin J_\varepsilon$ w.p. $(1 - \varepsilon)$ in which case the expectation of $h^{(j)}$ is $(1/2Q)\Delta_j$ while $j \in J_\varepsilon$ w.p. $\varepsilon$ in which case there is an additional term of $(1/4)\Delta_j$. The overall expectation is just $E_j$. Thus,

$$
\begin{aligned}
&\mathrm{Var}\left[\mathsf{E}\left[h^{(j)} \mid \mathbb{1}_{\{j \in J_\varepsilon\}}\right] \mid \mathcal{E}_j\right] \\
&= \left[(1 - \varepsilon)\left(\frac{1}{2Q}\right)^2 + \varepsilon\left(\frac{1}{4} + \frac{1}{2Q}\right)^2 - \left(\frac{\varepsilon}{4} + \frac{1}{2Q}\right)^2\right] \Delta_j^2 \tag{84} \\
&\geq \left[\frac{\varepsilon}{16} - \left(\frac{\varepsilon}{4} + \frac{1}{2Q}\right)^2\right] \Delta_j^2 \\
\text{(using (78), (82))} \quad &= \left[\frac{\varepsilon}{16} - \left(\frac{\varepsilon}{4} + \frac{1}{2Q}\right)^2\right] \cdot \left(\frac{\varepsilon}{4} + \frac{1}{2Q}\right)^{-2} \cdot 4 \cdot \tilde{V}_j \tag{85}
\end{aligned}
$$

The choice of $Q$ and $\varepsilon$ in (9) guarantees $\varepsilon \leq 1/8$ and $Q \geq 2/\varepsilon$. This, along with Equations (83) and (85) imply the following lemma.

**Lemma D.2.** $V_j \geq (1/(32\varepsilon))\tilde{V}_j$.

Next we concentrate on the second term on the RHS of (83).

When $j \in J \setminus J_\varepsilon$ then it is easy to see that independently for each $i \in S_j$ due to the random choice of $b_i$, exactly one of the $2Q$ variables $\cup_{b \in \{0,1\}} \cup_{q=1}^{Q} \{X_{i,b,q}\}$ is set to 1 and the rest to 0, with equal probability $1/(2Q)$ over all such choices. Thus, in this case, by independence over $i$ the variance of $h^{(j)}$ is the sum over $i \in S_j$ of the variance of $\Gamma_i := \sum_{b \in \{0,1\}} \sum_{q=1}^{Q} c_{i,b,q} X_{i,b,q}$, which for each $i$ can be calculated to be

$$
\left(\frac{1}{2Q}\right) \|\mathbf{c}_i\|_2^2 - \left(\frac{\langle \mathbf{c}_i, \mathbf{1}_{2Q} \rangle}{2Q}\right)^2,
$$

where $\mathbf{1}_{2Q}$ is the all 1s vector of dimension $2Q$. Thus,

$$
\mathrm{Var}\left[h^{(j)} \mid j \in J \setminus J_\varepsilon\right] = \sum_{i \in S_j} \left[\left(\frac{1}{2Q}\right) \|\mathbf{c}_i\|_2^2 - \left(\frac{\langle \mathbf{c}_i, \mathbf{1}_{2Q} \rangle}{2Q}\right)^2\right].
\tag{86}
$$

For the case when $j \in J_\varepsilon$, again the distributions of $\mathbf{X}_i$ ($i \in S_j$) are independent, so the variance of $h^{(j)}$ is the sum over $i \in S_j$ of the variance of $\Gamma_i$. For a given $i \in S_j$, $j \in J_\varepsilon$ we compute this variance by the law of total variance over conditioning on the choice of $b_i$ as follows:

$$
\begin{aligned}
\mathsf{E}\left[\Gamma_i \mid i \in S_j, j \in J_\varepsilon, b_i = b = 1 - a\right] &= \mathsf{E}_{\mathbf{X}_{i,a} \sim \overline{\mathcal{U}}_Q}[\langle \mathbf{c}_{i,a}, \mathbf{X}_{i,a}\rangle] + \mathsf{E}_{\mathbf{X}_{i,b} \sim \widehat{\mathcal{U}}_Q}[\langle \mathbf{c}_{i,b}, \mathbf{X}_{i,b}\rangle] \\
&= \frac{\langle \mathbf{c}_{i,a}, \mathbf{1}_Q\rangle}{Q} + \frac{\langle \mathbf{c}_{i,b}, \mathbf{1}_Q\rangle}{2},
\end{aligned}
\tag{87}
$$

and therefore,

$$
\begin{aligned}
&\mathrm{Var}\left[\mathsf{E}\left[\Gamma_i \mid b_i\right] \mid i \in S_j, j \in J_\varepsilon\right] \\
&= \frac{1}{2}\left[\sum_{b=1-a\in\{0,1\}}\left(\frac{\langle \mathbf{c}_{i,a}, \mathbf{1}_Q\rangle}{Q} + \frac{\langle \mathbf{c}_{i,b}, \mathbf{1}_Q\rangle}{2}\right)^2\right] - \left(\frac{1}{2Q} + \frac{1}{4}\right)^2 \cdot \langle \mathbf{c}_i, \mathbf{1}_{2Q}\rangle^2.
\end{aligned}
\tag{88}
$$

Similarly,

$$
\begin{aligned}
&\mathrm{Var}\left[\Gamma_i \mid i \in S_j, j \in J_\varepsilon, b_i = b = 1 - a\right] \\
&= \mathrm{Var}_{\mathbf{X}_{i,a} \sim \overline{\mathcal{U}}_Q}[\langle \mathbf{c}_{i,a}, \mathbf{X}_{i,a}\rangle] + \mathrm{Var}_{\mathbf{X}_{i,b} \sim \widehat{\mathcal{U}}_Q}[\langle \mathbf{c}_{i,b}, \mathbf{X}_{i,b}\rangle] \\
&= \left(\frac{1}{Q}\right)\|\mathbf{c}_{i,a}\|_2^2 - \left(\frac{\langle \mathbf{c}_{i,a}, \mathbf{1}_Q\rangle}{Q}\right)^2 + \frac{\|\mathbf{c}_{i,b}\|_2^2}{4}.
\end{aligned}
\tag{89}
$$

and therefore,

$$
\mathsf{E}\left[\mathrm{Var}\left[\Gamma_i \mid b_i\right] \mid i \in S_j, j \in J_\varepsilon\right] = \left(\frac{1}{8} + \frac{1}{2Q}\right)\|\mathbf{c}_i\|_2^2 - \left(\frac{1}{2Q^2}\right)\sum_{b\in\{0,1\}}\langle \mathbf{c}_{i,b}, \mathbf{1}_Q\rangle^2.
\tag{90}
$$

Combining (88) and (90) and using the law of total variance we obtain,

$$
\begin{aligned}
&\mathrm{Var}\left[\Gamma_i \mid i \in S_j, j \in J_\varepsilon\right] \\
&= \mathrm{Var}\left[\mathsf{E}\left[\Gamma_i \mid b_i\right] \mid i \in S_j, j \in J_\varepsilon\right] + \mathsf{E}\left[\mathrm{Var}\left[\Gamma_i \mid b_i\right] \mid i \in S_j, j \in J_\varepsilon\right] \\
&= \left(\frac{1}{8} + \frac{1}{2Q}\right)\|\mathbf{c}_i\|_2^2 + \frac{1}{2}\left[\sum_{b=1-a\in\{0,1\}}\left(\frac{\langle \mathbf{c}_{i,a}, \mathbf{1}_Q\rangle}{Q} + \frac{\langle \mathbf{c}_{i,b}, \mathbf{1}_Q\rangle}{2}\right)^2\right] \\
&\quad - \left(\frac{1}{2Q^2}\right)\sum_{b\in\{0,1\}}\langle \mathbf{c}_{i,a}, \mathbf{1}_Q\rangle^2 - \left(\frac{1}{2Q} + \frac{1}{4}\right)^2 \cdot \langle \mathbf{c}_i, \mathbf{1}_{2Q}\rangle^2 \\
&= \left(\frac{1}{8} + \frac{1}{2Q}\right)\|\mathbf{c}_i\|_2^2 + \left(\frac{1}{4} - \frac{1}{Q} - \frac{1}{Q^2}\right)\frac{\langle \mathbf{c}_i, \mathbf{1}_{2Q}\rangle^2}{4} - \left(\frac{1}{4} - \frac{1}{Q}\right)\langle \mathbf{c}_{i,a}, \mathbf{1}_Q\rangle\langle \mathbf{c}_{i,b}, \mathbf{1}_Q\rangle \\
&= \left(\frac{1}{8} + \frac{1}{2Q}\right)\|\mathbf{c}_i\|_2^2 + \left(\frac{1}{16} - \frac{1}{4Q}\right)(\langle \mathbf{c}_{i,a}, \mathbf{1}_Q\rangle - \langle \mathbf{c}_{i,b}, \mathbf{1}_Q\rangle)^2 - \left(\frac{\langle \mathbf{c}_i, \mathbf{1}_{2Q}\rangle}{2Q}\right)^2.
\end{aligned}
\tag{91}
$$

Therefore,

$$
\begin{aligned}
&\mathrm{Var}\left[h^{(j)} \mid j \in J_\varepsilon\right] \\
&= \sum_{i\in S_j}\left[\left(\frac{1}{8} + \frac{1}{2Q}\right)\|\mathbf{c}_i\|_2^2 + \left(\frac{1}{16} - \frac{1}{4Q}\right)(\langle \mathbf{c}_{i,a}, \mathbf{1}_Q\rangle - \langle \mathbf{c}_{i,b}, \mathbf{1}_Q\rangle)^2 \right. \\
&\qquad\qquad\left. - \left(\frac{\langle \mathbf{c}_i, \mathbf{1}_{2Q}\rangle}{2Q}\right)^2\right].
\end{aligned}
\tag{92}
$$

Combining (86) and (92) and using the fact that given $\Pr[j \in J_\varepsilon \mid j \in J] = \varepsilon$, we obtain,

$$\mathsf{E}\left[\mathrm{Var}\left[h^{(j)} \mid \mathbb{1}_{\{j \in J_\varepsilon\}}\right] \mid \mathcal{E}_j\right]$$

$$= \sum_{i \in S_j}\left[\left(\frac{\varepsilon}{8} + \frac{1}{2Q}\right)\|\mathbf{c}_i\|_2^2 + \left(\frac{\varepsilon}{16} - \frac{\varepsilon}{4Q}\right)(\langle\mathbf{c}_{i,a}, \mathbf{1}_Q\rangle - \langle\mathbf{c}_{i,b}, \mathbf{1}_Q\rangle)^2\right.$$

$$\left. - \left(\frac{\langle\mathbf{c}_i, \mathbf{1}_{2Q}\rangle}{2Q}\right)^2\right] \quad (93)$$

$$\geq \sum_{i \in S_j}\left[\left(\frac{\varepsilon}{8}\right)\|\mathbf{c}_i\|_2^2 + \left(\frac{\varepsilon}{16} - \frac{\varepsilon}{4Q}\right)(\langle\mathbf{c}_{i,a}, \mathbf{1}_Q\rangle - \langle\mathbf{c}_{i,b}, \mathbf{1}_Q\rangle)^2\right] \quad (94)$$

where the final inequality follows from $\langle\mathbf{c}_i, \mathbf{1}_{2Q}\rangle \leq \|\mathbf{c}_i\|_1 \leq \sqrt{2Q}\|\mathbf{c}_i\|_2$. Using (94) in conjunction with (83) along with the setting of $Q \geq 4$ in (9) directly yields the following lemma.

**Lemma D.3.** $V_j \geq (\varepsilon/8)\sum_{i \in S_j}\|\mathbf{c}_i\|_2^2$. In particular, $\sum_{j \in [m]} V_j \geq (\varepsilon/8)\|\mathbf{c}\|_2^2$.

Next, we upper bound $V_j^2$ in the following lemma.

**Lemma D.4.** $\sum_{j \in [m]} V_j^2 \leq 2\tau\|\mathbf{c}\|_2^4$.

*Proof.* It can be seen from (9) that $Q \geq 4, \varepsilon \leq 1/4$ and $\tau \leq 1/(20Q^2)$. From this, (i) (84) implies an upper bound of $(1/2)\Delta_j^2 \leq (1/2)\left(\sum_{i \in S_j}\|\mathbf{c}_i\|_1\right)^2$ on first term on the RHS of (83), and (ii) (93) upper bounds the second term on the RHS of (83) by

$$\frac{1}{4}\sum_{i \in S_j}\left(\|\mathbf{c}_i\|_2^2 + (|\langle\mathbf{c}_{i,a}, \mathbf{1}_Q\rangle| + |\langle\mathbf{c}_{i,b}, \mathbf{1}_Q\rangle|)^2\right) \leq \frac{1}{4}\sum_{i \in S_j}\left(\|\mathbf{c}_i\|_1^2 + \|\mathbf{c}_i\|_1^2\right)$$

$$= \frac{1}{2}\sum_{i \in S_j}\|\mathbf{c}_i\|_1^2 \leq \frac{1}{2}\left(\sum_{i \in S_j}\|\mathbf{c}_i\|_1\right)^2 \quad (95)$$

where the first inequality follows from $\|\mathbf{c}_i\|_2 \leq \|\mathbf{c}_i\|_1$, and $|\langle\mathbf{c}_{i,a}, \mathbf{1}_Q\rangle| + |\langle\mathbf{c}_{i,b}, \mathbf{1}_Q\rangle| \leq \|\mathbf{c}_{i,a}\|_1 + \|\mathbf{c}_{i,b}\|_1 = \|\mathbf{c}_i\|_1$.

Combining the two upper bounds on the terms on the RHS of (83), we have

$$V_j \leq \left(\sum_{i \in S_j}\|\mathbf{c}_i\|_1\right)^2 \leq (2Q)\left(\sum_{i \in S_j}\|\mathbf{c}_i\|_2\right)^2 \quad (96)$$

Let $S_j \ni i^* := \arg\max_{i \in S_j}\|\mathbf{c}_i\|_2$. Then, (73) and (74) imply that

$$\|\mathbf{c}_{i^*}\|_2^2 \leq 2\tau\|\mathbf{c}\|_2^2, \quad \text{and} \quad \|\mathbf{c}_i\|_2^2 \leq (\tau/(4d^8))\|\mathbf{c}\|_2^2, \quad \forall i \in S_j \setminus \{i^*\} \quad (97)$$

Recall that $|S_j| \leq d$, and consider the expansion of $\left(\sum_{i \in S_j}\|\mathbf{c}_i\|_2\right)^4$ into terms of the form $\|\mathbf{c}_{i_1}\|_2\|\mathbf{c}_{i_2}\|_2\|\mathbf{c}_{i_3}\|_2\|\mathbf{c}_{i_4}\|_2$ for $(i_1, i_2, i_3, i_4) \in S_j \times S_j \times S_j \times S_j$. There is one term $\|\mathbf{c}_{i^*}\|_2^4 \leq 4\tau^2\|\mathbf{c}_{i^*}\|_2^2\|\mathbf{c}\|_2^2$ (using (97)) corresponding to $i_1 = i_2 = i_3 = i_4 = i^*$. Any other term has at least one of $i_1, i_2, i_3$ or $i_4$ distinct from $i^*$ and – using the definition of $i^*$ and the conditions in (97) – can be bounded by $\|\mathbf{c}_{i^*}\|_2^3(\tau/(4d^8))\|\mathbf{c}\|_2 \leq (\tau^2/(4d^8))\|\mathbf{c}_{i^*}\|_2^2\|\mathbf{c}\|_2^2$. The total number of terms is $d^4$, and summing all of them up we obtain that $\left(\sum_{i \in S_j}\|\mathbf{c}_i\|_2\right)^4$ is at most $\tau^2(4 + 1/(4d^4))\|\mathbf{c}_{i^*}\|_2^2\|\mathbf{c}\|_2^2 \leq 5\tau^2\|\mathbf{c}_{i^*}\|_2^2\|\mathbf{c}\|_2^2$. Using the choice of $\tau \leq 1/(20Q^2)$, (96) then yields,

$$V_j^2 \leq 2\tau\|\mathbf{c}_{i^*}\|_2^2\|\mathbf{c}\|_2^2 \Rightarrow \sum_{j \in [m]} V_j^2 \leq 2\tau\|\mathbf{c}\|_2^2\sum_{j \in [m]}\|\mathbf{c}_{i^*}\|_2^2 \leq 2\tau\|\mathbf{c}\|_2^4.$$

$\square$

We are now ready to show that $V^{(J)}$ is concentrated around its mean $(1/2)\sum_{j=1}^{m} V_j$ (by (79)).

**Lemma D.5.**

$$\Pr_J \left[ \left| V^{(J)} - \mathsf{E}_J\left[V^{(J)}\right] \right| = \left| V^{(J)} - (1/2)\sum_{j=1}^{m} V_j \right| > \tau^{1/4}\sum_{j=1}^{m} V_j = \frac{\tau^{1/4}}{2}\mathsf{E}_J\left[V^{(J)}\right] \right]$$

$$\leq \quad 2 \cdot \exp(-1/\tau^{1/4}). \tag{98}$$

*Proof.* First, observe that the setting in (9) implies $\tau \leq (\varepsilon/4)^8$. From Lemma D.3 we obtain that the LHS of (98) is at most,

$$\Pr_J \left[ \left| V^{(J)} - \mathsf{E}_J\left[V^{(J)}\right] \right| > \tau^{1/4}(\varepsilon/8)\|\mathbf{c}\|_2^2 \right] \quad \leq \quad 2\exp\left( -\frac{2\tau^{1/2}(\varepsilon/8)^2\|\mathbf{c}\|_2^4}{\sum_{j=1}^{m} V_j^2} \right)$$

$$\leq \quad 2\exp\left( -\frac{2\tau^{1/2}(\varepsilon/8)^2\|\mathbf{c}\|_2^4}{2\tau\|\mathbf{c}\|_2^4} \right)$$

$$\leq \quad 2\exp(-1/\tau^{1/4}) \tag{99}$$

using our bound on $\tau$ where the first inequality uses the Chernoff-Hoeffding inequality (Theorem B.3) and the second inequality uses Lemma D.4. $\qquad\square$

Using the above we have the following lemma showing that $E^{(J)}$ is also highly concentrated around its mean $(1/2)\sum_{j=1}^{m} E_j$ (by (79)).

**Lemma D.6.** *Except with probability* $2\exp(-1/\tau^{1/4}) + \sqrt{\varepsilon}$ *over the choice of J,*

$$\left| E^{(J)} - \mathsf{E}_J\left[E^{(J)}\right] \right| = \left| E^{(J)} - (1/2)\sum_{j=1}^{m} E_j \right| < 16\varepsilon^{1/4}\sqrt{V^{(J)}}.$$

*Proof.* Using $\Pr_J\left[ V^{(J)} < (1/4)\sum_{j=1}^{m} V_j \right] \leq 2\exp(-1/\tau^{1/4})$ obtained from Lemma D.5 we have,

$$\Pr_J \left[ \left| E^{(J)} - (1/2)\sum_{j=1}^{m} E_j \right| \geq 16\varepsilon^{1/4}\sqrt{V^{(J)}} \right]$$

$$\leq 2\exp(-1/\tau^{1/4}) + \Pr_J \left[ \left| E^{(J)} - \mathsf{E}_J\left[E^{(J)}\right] \right| \geq 8\varepsilon^{1/4}\sqrt{\sum_{j\in[m]} V_j} \right] \qquad \text{(using (79))}$$

$$\leq 2\exp(-1/\tau^{1/4}) + \Pr_J \left[ \left| E^{(J)} - \mathsf{E}_J\left[E^{(J)}\right] \right| \geq \varepsilon^{-1/4}\sqrt{\sum_{j\in[m]} \tilde{V}_j} \right] \qquad \text{(using Lemma D.2)}$$

$$\leq 2\exp(-1/\tau^{1/4}) + \sqrt{\varepsilon} \qquad\qquad\qquad\qquad \text{(using (79) and (4))} \quad (100)$$

$\qquad\qquad\qquad\qquad\qquad\qquad\qquad\qquad\qquad\qquad\qquad\qquad\qquad\qquad\qquad\qquad\qquad\square$

We now show that w.h.p. over the choice of $J$, the distribution of $h(\mathbf{X})$ is close to a Gaussian distribution with mean $E^{(J)}$ and variance $V^{(J)}$.

**Lemma D.7.** *For a small enough choice of $\varepsilon_0$ in (9), except with probability $2\exp(-1/\tau^{1/4})$ over the choice of J, for all $t \in \mathbb{R}$:*

$$|\Pr\left[h(\mathbf{X}) > t \mid J\right] - \Pr\left[\mathcal{G} > t\right]| \leq O\left( \sqrt{\tau(Q/\varepsilon)^3} \right),$$

*where $\mathcal{G} \sim N\left(E^{(J)}, V^{(J)}\right)$.*

*Proof.* For $j \in [m]$, let $\rho_j := \mathsf{E}_{\mathcal{D}_X}\left[\left|h^{(j)} - E_j\right|^3 \mid \mathcal{E}_j\right]$. It is easy to see that the maximum vaue of $\left|h^{(j)} - E_j\right|$ is at most $2\sum_{i \in S_j}\|\mathbf{c}_i\|_1 \leq \sqrt{8Q}\sum_{i \in S_j}\|\mathbf{c}_i\|_2$. Using the definition of $i^*$ as in the proof of Lemma D.4 and following a similar set of arguments, we upper bound the expansion of $\left(\sum_{i \in S_j}\|\mathbf{c}_i\|_2\right)^3$ with the sum of one term $\|\mathbf{c}_{i^*}\|_2^3 \leq \sqrt{2\tau}\|\mathbf{c}_{i^*}\|_2^2\|\mathbf{c}\|_2$ and at most $d^3$ other terms each of value at most $\|\mathbf{c}_{i^*}\|_2^2(\tau/(4d^8))\|\mathbf{c}\|_2$. This sum therefore is at most $2\sqrt{\tau}\|\mathbf{c}_{i^*}\|_2^2\|\mathbf{c}\|_2$. Using this and Lemma D.3 we obtain the following,

$$\rho_j \leq 64\sqrt{Q^3\tau}\|\mathbf{c}_{i^*}\|_2^2\|\mathbf{c}\|_2 \leq 2^9\left(\sqrt{Q^3\tau}/\varepsilon\right)\|\mathbf{c}\|_2 V_j \leq 2^{12}\left(\sqrt{\tau(Q/\varepsilon)^3}\right)V_j\sqrt{\frac{1}{4}\sum_{j=1}^{m}V_j} \quad (101)$$

From Lemma D.5 we obtain that $\Pr_J\left[V^{(J)} < (1/4)\sum_{j=1}^{m}V_j\right] \leq 2\exp(-1/\tau^{1/4})$, and except for this probability over the choice of $J$ by (101),

$$\rho_j \leq 2^{12}\left(\sqrt{\tau(Q/\varepsilon)^3}\right)V_j\sqrt{V^{(J)}}, \quad (102)$$

for each $j \in [m]$. Using the above along with (75), and applying the Berry-Esseen theorem (Theorem B.1) we obtain the error in (2) to be,

$$O\left(\frac{1}{\sqrt{V^{(J)}}} \cdot \max_{j \in J}\frac{\rho_j}{V_j}\right) = O\left(\sqrt{\tau(Q/\varepsilon)^3}\right),$$

completing the proof. $\qquad\square$

From Theorem B.6 we have,

$$\mathsf{TV}\left(N\left(E^{(J)}, V^{(J)}\right), N\left(\mathsf{E}_J\left[E^{(J)}\right], \mathsf{E}_J\left[V^{(J)}\right]\right)\right)$$

$$\leq \quad \frac{3\left|V^{(J)} - \mathsf{E}_J\left[V^{(J)}\right]\right|}{V^{(J)}} + \frac{\left|E^{(J)} - \mathsf{E}_J\left[E^{(J)}\right]\right|}{2\sqrt{V^{(J)}}}. \quad (103)$$

Lemma D.5 implies that the first term on the RHS above is at most $3\tau^{1/4}/(2 - \tau^{1/4}) \leq 4\tau^{1/4}$ except w.p. $2\exp(-1/\tau^{1/4})$ over the choice of $J$. On the other hand, Lemma D.6 upper bounds the second term on the RHS of (103) by $8\varepsilon^{1/4}$ except w.p. $2\exp(-1/\tau^{1/4}) + \sqrt{\varepsilon}$ over the choice of $J$. Combining this with Lemma D.7 yields that except with probability $6\exp(-1/\tau^{1/4}) + \sqrt{\varepsilon}$ over the choice of $J$, for all $t \in \mathbb{R}$:

$$\left|\Pr\left[h(\mathbf{X}) > t \mid J\right] - \Pr\left[N\left(\mathsf{E}_J\left[E^{(J)}\right], \mathsf{E}_J\left[V^{(J)}\right]\right) > t\right]\right|$$

$$\leq \quad O\left(\sqrt{\tau(Q/\varepsilon)^3}\right) + O\left(\tau^{1/8}\right) + O\left(\varepsilon^{1/4}\right) \quad (104)$$

which implies Lemma D.1 using our choice of parameters and small enough $\varepsilon_0$ in (9).

### D.3 Invariance of product of regular LTFs

We will use the setup of Sec. D.1 and instead of one linear form, we consider $\ell$ linear forms $h_1, \ldots, h^\ell$ as: $h_s := \sum_{i=1}^{M}\sum_{b \in \{0,1\}}\sum_{q \in [Q]}c_{s,i,b,q}X_{i,b,q}$, $(s \in [\ell])$ which are all $(\tau, d^8)$-*nice* as defined in (73) and (74). Without loss of generality we shall assume that the sum of squares of coefficients for each of $h_s$ $(s \in [\ell])$ is 1 i.e.,

$$\|\mathbf{c}_s\|_2^2 = 1, \quad \forall s \in [\ell] \quad (105)$$

As in previous section $h_s^{(j)}$ will denote the contribution to $h_s$ from the variables corresponding to $S_j$ $(j \in [m])$. Also, $E_{s,j}, V_{s,j}, E_s^{(J)}$ and $V_s^{(J)}$ shall denote the quantities defined in (75), (76), (77) corresponding to $h_s$ $(s \in [\ell])$. Along with this, we will reuse some of the derivations in the previous subsection for a single linear form $h$.

Let us define the following random (given a fixed $\mathbf{X}$) versions of the above linear forms, with independent Gaussian noise added to them.

$$\tilde{h}_s(\mathbf{X}) := \sum_{j=1}^{m}\left(\sum_{i \in S_j}\sum_{b \in \{0,1\}}\sum_{q \in [Q]}c_{s,i,b,q}X_{i,b,q} + \zeta_{s,j}\right), \quad \forall s \in [\ell], \quad (106)$$

where $\zeta_{s,j}$ are independent mean-zero Gaussian random variables given by

$$\zeta_{s,j} \sim N\left(0, \frac{\varepsilon^2}{64} \sum_{i \in S_j} \|\mathbf{c}_{s,i}\|_2^2\right) \tag{107}$$

Note that the above along with (105) implies that $\zeta_s := \sum_{j=1}^m \zeta_{s,j}$, $s = 1, \ldots, \ell$, are iid $N(0, (\varepsilon^2/64))$ Gaussians. Let $\boldsymbol{\zeta}$ denote the choices of $\zeta_{s,j}$ ($s \in [\ell], j \in [m]$ The following lemma is derived from a union bound application of Lemma D.1.

**Lemma D.8.** *For any $t_1, \ldots, t_\ell$, with except with probability $O(\ell\sqrt{\varepsilon})$ over the choice of $J$ in $\mathcal{D}_X$ (Fig. 1)*

$$\left| \Pr_{\mathcal{D}_X}\left[ \bigwedge_{s=1}^\ell (h_s(\mathbf{X}) > t_s) \,|J\right] - \Pr_{\mathcal{D}_X, \boldsymbol{\zeta}}\left[ \bigwedge_{s=1}^\ell \left(\tilde{h}_s(\mathbf{X}) > t_s\right) |J\right] \right| \leq O\left(\ell\varepsilon^{1/4}\right). \tag{108}$$

*Proof.* We first observe that all $\zeta_s$ ($s \in [\ell]$) will, by Chebyshev's inequality, have magnitude at most $\varepsilon^{3/4}/8$ except with probability $\ell\sqrt{\varepsilon}$. Further, using union bound over Lemma D.1, except with probability $O(\ell\sqrt{\varepsilon})$ over the choice of $J$, (81) holds for each $h_s$. Now, from (79) and Lemma D.3 $\mathsf{E}_J\left[V_s^{(J)}\right]$ is at least $(\varepsilon/16)$. Thus, taking $\delta$ in (81) to be the (high probability) upper bound of $\varepsilon^{3/4}/8$ on the magnitude of $\zeta_s$ for each $h_s$, we obtain the error on the RHS of (81) to be $O(\varepsilon^{1/4})$ for each $s \in [\ell]$. A further union bound completes the proof. $\square$

### D.4 Concentration of covariance

Fix for this subsection fix $s, r \in [\ell]$ (not necessarily distinct). We will show a high probability bound on the concentration on $\mathrm{Cov}(\tilde{h}_s, \tilde{h}_r | J)$.

For a given $J$, the variables corresponding to $S_j$ are independent of any $S_{j'}, j' \neq j$. Further the $\{\zeta_{s,j} \mid$ variables are independent. Thus, we have

$$C^{(J)} := \mathrm{Cov}(\tilde{h}_s, \tilde{h}_r \mid J) = \sum_{j=1}^m \mathrm{Cov}\left(h_s^{(j)} + \zeta_{s,j}, h_r^{(j)} + \zeta_{r,j} \mid J\right)$$

$$= \sum_{j=1}^m \mathrm{Cov}\left(h_s^{(j)}, h_r^{(j)} \mid J\right) = \sum_{j=1}^m C_j \mathbb{1}_{\mathcal{E}_j}, \tag{109}$$

where

$$C_j := \mathrm{Cov}\left(h_s^{(j)}, h_r^{(j)} \mid \mathcal{E}_j\right) \tag{110}$$

Note that by linearity of expectation,

$$\mathsf{E}_J\left[C^{(J)}\right] = (1/2) \sum_{j=1}^m C_j \tag{111}$$

Using the fact that $2\,\mathrm{Cov}(\mathbf{A}, \mathbf{B}) = \mathrm{Var}(\mathbf{A} + \mathbf{B}) - \mathrm{Var}(\mathbf{A}) - \mathrm{Var}(\mathbf{B})$ we have,

$$\sum_{j=1}^m C_j^2 \leq \frac{1}{4} \sum_{j=1}^m \mathrm{Var}\left(h_s^{(j)} + h_r^{(j)} \mid \mathcal{E}_j\right)^2 \tag{112}$$

Using the same arguments used in the proof of Lemma D.4, we can obtain analogous to (96) the following:

$$C_j^2 \leq \frac{1}{4}(2Q)^2 \left(\sum_{i \in S_j} (\|\mathbf{c}_{s,i}\|_2 + \|\mathbf{c}_{r,i}\|_2)\right)^4$$

$$\leq 2 \cdot (2Q)^2 \left[\left(\sum_{i \in S_j} \|\mathbf{c}_{s,i}\|_2\right)^4 + \left(\sum_{i \in S_j} \|\mathbf{c}_{r,i}\|_2\right)^4\right] \tag{113}$$

where the second inequality follows Holder's inequality (in particular $(a + b)^4 \leq 8(a^4 + b^4)$. The rest of the arguments from the proof of Lemma D.4) lead to,

$$\sum_{j=1}^{m} C_j^2 \leq 4\tau(\|\mathbf{c}_s\|_2^4 + \|\mathbf{c}_r\|_2^4) \leq 8\tau \tag{114}$$

Applying the Chernoff-Hoeffding bound (Thm. B.3) along the lines in Lemma D.5 we obtain the following,

$$\Pr_J\left[\left|\left|C^{(J)} - \mathsf{E}_J\left[C^{(J)}\right]\right|\right| = \left|C^{(J)} - (1/2)\sum_{i=1}^{m} C_j\right| > 2\tau^{1/4}\right]$$

$$\leq \quad 2 \cdot \exp(-8\tau^{1/2}/8\tau) \leq 2 \cdot \exp(-1/\sqrt{\tau}). \tag{115}$$

Note that the above also gives a slightly different, as compared to Lemma D.5, bound for $s = r$ with $C^{(J)} = V_s^{(J)}$ and $C_j = V_{s,j}$.

Next we apply the multi-dimensional version of Berry Esseen theorem on $\tilde{h}_s$ ($s \in [\ell]$) together.

### D.5 Applying multi-dimensional Berry-Esseen

From (115), the analysis in Lemma D.6 (the last three inequalities in (100)), and from Lemma D.5 along with the choice of parameters and small enough $\varepsilon_0$ in (9), we obtain using union bound over all pairs $s, r \in [\ell]$, and all $s \in [\ell]$ that except with probability $O(\ell^2\sqrt{\varepsilon})$ over the choice of $J$,

$$\left|\text{Cov}(\tilde{h}_s, \tilde{h}_r \mid J) - \mathsf{E}_J\left[\text{Cov}(\tilde{h}_s, \tilde{h}_r \mid J)\right]\right| \leq 2\tau^{1/4} \tag{116}$$

is satisfied for all pairs $s, r \in [\ell]$ (including $s = r$) and

$$\left|E_s^{(J)} - \mathsf{E}_J\left[E_s^{(J)}\right]\right| \leq 8\varepsilon^{1/4}\sqrt{\sum_{j\in[m]} V_{s,j}}, \tag{117}$$

and,

$$\left|V_s^{(J)} - (1/2)\sum_{i=1}^{m} V_{s,j}\right| \leq \tau^{1/4}\sum_{i=1}^{m} V_{s,j}, \tag{118}$$

are satisfied for all $s \in [\ell]$. For the remainder of this subsection we shall fix such a choice of $J$. All expectations of the form $\mathsf{E}$ are after fixing $J$, while $\mathsf{E}_J$ are expectations over choices of $J$ in $\mathcal{D}_X$

Consider independent random vectors $\{\mathbf{Z}_j \in \mathbb{R}^\ell \mid j \in [m]\}$ where $Z_{j,s} = h_s^{(j)}(\mathbf{X})$. Let $\mathbf{Z} := \sum_{j=1}^{m} \mathbf{Z}_j$. Further, let $\{\tilde{\mathbf{Z}}_j \in \mathbb{R}^\ell \mid j \in [m]\}$ independent random vectors such that $\tilde{Z}_{j,s} := Z_{j,s} + \zeta_{s,j} = h_s^{(j)}(\mathbf{X}) + \zeta_{s,j}$. By the definition of $\tilde{h}_s$ we have

$$\tilde{\mathbf{Z}} := \sum_{j=1}^{m} \tilde{\mathbf{Z}}_j = (h_1(\mathbf{X}) + \zeta_1, \ldots, h_\ell(\mathbf{X}) + \zeta_\ell)^\mathsf{T} = \left(\tilde{h}_1(\mathbf{X}), \ldots, \tilde{h}_\ell(\mathbf{X})\right)^\mathsf{T}, \tag{119}$$

First let $\boldsymbol{\Sigma}, \tilde{\boldsymbol{\Sigma}}$ be the the covariance matrices of $\mathbf{Z}, \tilde{\mathbf{Z}}$ respectively. Since $\{\zeta_s\}_{s=1}^{\ell}$ are iid $N(0, (\varepsilon^2/64))$ (ref. (107)) and independent of $\mathbf{Z}$, we have that,

$$\tilde{\boldsymbol{\Sigma}} = \boldsymbol{\Sigma} + \frac{\varepsilon^2}{64}\mathbf{I}. \tag{120}$$

As a pre-requisite to applying the multi-dimensional Berry-Esseen theorem the we need to upper bound $\sum_{j=1}^{m} \mathsf{E}\left[\|\tilde{\mathbf{Z}}_j - \mathsf{E}[\tilde{\mathbf{Z}}_j]\|_2^3\right]$. We first observe that

$$\left\|\tilde{\mathbf{Z}}_j - \mathsf{E}[\tilde{\mathbf{Z}}_j]\right\|_2^3 = \left(\sum_{s=1}^{\ell} \left|\tilde{Z}_{j,s} - \mathsf{E}[\tilde{Z}_{j,s}]\right|^2\right)^{3/2}$$

$$\text{(By Holder's Ineq.)} \leq \left(\ell^{1/3}\left(\sum_{s=1}^{\ell} \left|\tilde{Z}_{j,s} - \mathsf{E}[\tilde{Z}_{j,s}]\right|^3\right)^{2/3}\right)^{3/2}$$

$$= \sqrt{\ell}\sum_{s=1}^{\ell} \left|\tilde{Z}_{j,s} - \mathsf{E}[\tilde{Z}_{j,s}]\right|^3 \tag{121}$$

So we need to only bound $\mathsf{E}\left[\left|\tilde{Z}_{j,s} - \mathsf{E}[\tilde{Z}_{j,s}]\right|^3\right]$, for which we consider the following two cases.

Case 1: $j \in J$. In this case,

$$\left|\tilde{Z}_{j,s} - \mathsf{E}[\tilde{Z}_{j,s}]\right|^3 \leq (|Z_{j,s} - \mathsf{E}[Z_{j,s}]| + |\zeta_{j,s}|)^3$$

$$\text{(By Holder's Ineq.)} \leq \left(2^{2/3}\left(|Z_{j,s} - \mathsf{E}[Z_{j,s}]|^3 + |\zeta_{j,s}|^3\right)^{1/3}\right)^3$$

$$\leq 4 \cdot \left(|Z_{j,s} - \mathsf{E}[Z_{j,s}]|^3 + |\zeta_{j,s}|^3\right) \tag{122}$$

Note that $|Z_{j,s} - \mathsf{E}[Z_{j,s}]| = \left|h_s^{(j)} - E_{s,j}\right|$. Thus, from the first inequality of (101) in the proof of Lemma D.7,

$$\mathsf{E}\left[|Z_{j,s} - \mathsf{E}[Z_{j,s}]|^3\right] \leq 64\left(\sqrt{Q^3\tau}/\varepsilon\right)\|\mathbf{c}_s\|_2 \sum_{i \in S_j}\|\mathbf{c}_{s,i}\|_2^2. \tag{123}$$

For the remaining term $|\zeta_{j,s}|^3$ using the known upper bound of $2\sigma^3$ on the third moment of $N(0, \sigma^2)$ we obtain:

$$\mathsf{E}\left[|\zeta_{j,s}|^3\right] \leq 2\left(\frac{\varepsilon^2}{64}\sum_{i \in S_j}\|\mathbf{c}_{s,i}\|_2^2\right)^{3/2} \leq 4\left(\frac{\varepsilon^2}{64}\sum_{i \in S_j}\|\mathbf{c}_{s,i}\|_2^2\right)\sqrt{\tau}\|\mathbf{c}_s\|_2 \tag{124}$$

using (74) along the lines of previous analyses.

Case 2: $j \notin J$. In this case $\tilde{Z}_{j,s} = \zeta_{j,s}$. Thus we have,

$$\mathsf{E}\left[\left|\tilde{Z}_{j,s} - \mathsf{E}[\tilde{Z}_{j,s}]\right|^3\right] = \mathsf{E}\left[|\zeta_{j,s}|^3\right] \leq 4\left(\frac{\varepsilon^2}{64}\sum_{i \in S_j}\|\mathbf{c}_{s,i}\|_2^2\right)\sqrt{\tau}\|\mathbf{c}_s\|_2 \tag{125}$$

Combining the above and with our setting parameters in (9) we have our desired bound as:

$$\sum_{j=1}^m \mathsf{E}\left[\|\tilde{\mathbf{Z}}_j - \mathsf{E}[\tilde{\mathbf{Z}}_j]\|_2^3\right] \leq 2^9 \cdot \left(\sqrt{\tau Q^3 \ell}/\varepsilon\right)\sum_{s=1}^\ell\left(\|\mathbf{c}_s\|_2\sum_{j=1}^m\sum_{i \in S_j}\|\mathbf{c}_{s,i}\|_2^2\right)$$

$$= 2^9 \cdot \left(\sqrt{\tau Q^3 \ell}/\varepsilon\right)\sum_{s=1}^\ell\left(\|\mathbf{c}_s\|_2\sum_{i=1}^M\|\mathbf{c}_{s,i}\|_2^2\right)$$

$$= 2^9 \cdot \left(\sqrt{\tau Q^3 \ell}/\varepsilon\right)\sum_{s=1}^\ell\|\mathbf{c}_s\|_2\|\mathbf{c}_s\|_2^2$$

$$= 2^9 \cdot \left(\sqrt{\tau Q^3 \ell^3}/\varepsilon\right) \tag{126}$$

where we use the normalization $\|\mathbf{c}_s\|_2 = 1$ ($s \in [\ell]$). From (120) we get that the minimum eigenvalue of $\tilde{\boldsymbol{\Sigma}}$ is at least $\varepsilon^2/64$. Thus, the maximum eigenvalue of $\tilde{\boldsymbol{\Sigma}}^{-1/2}$ is at most $8/\varepsilon$. This implies,

$$\sum_{j=1}^m \mathsf{E}\left[\left\|\tilde{\boldsymbol{\Sigma}}^{-1/2}\left(\tilde{\mathbf{Z}}_j - \mathsf{E}[\tilde{\mathbf{Z}}_j]\right)\right\|_2^3\right] \leq \sum_{j=1}^m \mathsf{E}\left[\left((8/\varepsilon)\left\|\tilde{\mathbf{Z}}_j - \mathsf{E}[\tilde{\mathbf{Z}}_j]\right\|_2\right)^3\right]$$

$$= \left(\frac{8}{\varepsilon}\right)^3\sum_{j=1}^m \mathsf{E}\left[\|\tilde{\mathbf{Z}}_j - \mathsf{E}[\tilde{\mathbf{Z}}_j]\|_2^3\right]$$

$$\text{(by (126))} \leq 2^{21} \cdot \left(\sqrt{\tau Q^3 \ell^3/\varepsilon^7}\right) := \xi \tag{127}$$

Using the above and applying the multi-dimensional Berry-Esseen theorem (Theorem B.2) we obtain that for any convex set $A$ in $\mathbb{R}^\ell$,

$$\left|\Pr\left[\tilde{\mathbf{Z}} \in A\right] - \Pr\left[\tilde{\boldsymbol{\Upsilon}} \in A\right]\right| \leq O\left(\ell^{1/4}\xi\right) \tag{128}$$

where $\xi$ is as in (127) and $\tilde{\boldsymbol{\Upsilon}} \in \mathbb{R}^\ell$ is distributed as $N\left(\mathsf{E}[\tilde{\mathbf{Z}}], \tilde{\boldsymbol{\Sigma}}\right)$.

In the above, the parameters of $N\left(\mathsf{E}[\tilde{\mathbf{Z}}], \tilde{\boldsymbol{\Sigma}}\right)$ depend on the specific choice of $J$. Next we shall replace it with a distribution independent of $J$ using the concentration of the means and covariances that we assumed while fixing $J$ in this subsection.

### D.5.1 Making the joint distribution independent of $J$

Consider a matrix $\boldsymbol{\Sigma}'$ the $(s,r)$th entry being $\mathsf{E}_J\left[\mathrm{Cov}(\tilde{h}_s, \tilde{h}_r \mid J)\right]$. It is easy to see that $\boldsymbol{\Sigma}' = \mathsf{E}_J[\boldsymbol{\Sigma}]$ and therefore $\boldsymbol{\Sigma}'$ is a symmetric positive semidefinite matrix. Now, define

$$\overline{\boldsymbol{\Sigma}} := \boldsymbol{\Sigma}' + \frac{\varepsilon^2}{64}\mathbf{I}. \tag{129}$$

Clearly, $\overline{\boldsymbol{\Sigma}}$ is positive-definite. Further, from (120) and our assumption on the choice of $J$ in (116),

$$\left\|\overline{\boldsymbol{\Sigma}} - \tilde{\boldsymbol{\Sigma}}\right\|_F^2 = \|\boldsymbol{\Sigma}' - \boldsymbol{\Sigma}\|_F^2 \leq 2\ell^2 \tau^{1/4}. \tag{130}$$

Since the minimum eigenvalue of $\tilde{\boldsymbol{\Sigma}}$ is at least $(\varepsilon^2/64)$, the sum of squares of eigenvalues of $\tilde{\boldsymbol{\Sigma}}^{-1}\left(\overline{\boldsymbol{\Sigma}} - \tilde{\boldsymbol{\Sigma}}\right)$ is (using (130)) at most

$$\ell\left(\frac{64}{\varepsilon^2}\right)\|\boldsymbol{\Sigma}' - \boldsymbol{\Sigma}\|_F^2 \leq (128\ell^3 \tau^{1/4}/\varepsilon^2).$$

Thus, applying Thm. B.5 we have the following lemma.

**Lemma D.9.** *For any convex subset $A \subseteq \mathbb{R}^\ell$,*

$$\left|\Pr\left[\overline{\boldsymbol{\Upsilon}} \in A\right] - \Pr\left[\tilde{\boldsymbol{\Upsilon}} \in A\right]\right| \leq (3/2)\sqrt{(128\ell^3 \tau^{1/4}/\varepsilon^2)} \leq (16\ell^{3/2}/\varepsilon)\tau^{1/8}. \tag{131}$$

*where $\tilde{\boldsymbol{\Upsilon}}$ is as in (128) and $\overline{\boldsymbol{\Upsilon}}$ is distributed as $N\left(\mathsf{E}[\tilde{\mathbf{Z}}], \overline{\boldsymbol{\Sigma}}\right)$*

While the covariance matrix $\overline{\boldsymbol{\Sigma}}$ is independent of the choice of $J$, the expectation vector $\mathsf{E}[\tilde{\mathbf{Z}}]$ still is. We use (117) and (118) to eliminate this dependence as follows.

**Lemma D.10.** *Let $\boldsymbol{\mu} \in \mathbb{R}^\ell$ be such that its $s$th entry is $\mathsf{E}_J\left[E_s^{(J)}\right]$. Let $\overline{\boldsymbol{\Upsilon}}$ be as in Lemma D.9, and let $\boldsymbol{\Upsilon}$ be distributed as $N\left(\boldsymbol{\mu}, \overline{\boldsymbol{\Sigma}}\right)$. Then, for any $t_1, \ldots, t_\ell$,*

$$\left|\Pr\left[\bigwedge_{s=1}^\ell (\Upsilon_s > t_s)\right] - \Pr\left[\bigwedge_{s=1}^\ell (\overline{\Upsilon}_s > t_s)\right]\right| \leq O(\ell\varepsilon^{1/4}) \tag{132}$$

*Proof.* First observe that since $\boldsymbol{\Upsilon}$ and $\overline{\boldsymbol{\Upsilon}}$ have identical covariance matrices, we can take

$$\boldsymbol{\Upsilon} = \overline{\boldsymbol{\Upsilon}} + \boldsymbol{\mu} - \mathsf{E}[\tilde{\mathbf{Z}}] \tag{133}$$

Next, since $\zeta_{s,j}$ are mean zero ($s \in [\ell], j \in [m]$), we have

$$\mathsf{E}[\tilde{\mathbf{Z}}] = \mathsf{E}[\mathbf{Z}] = (\mathsf{E}[h_1|J], \ldots, \mathsf{E}[h_\ell|J])^{\mathsf{T}|} = \left(E_1^{(J)}, \ldots, E_\ell^{(J)}\right)^{\mathsf{T}}.$$

For any $s \in [\ell]$ from (117) we have that,

$$\left|\mu_s - E_s^{(J)}\right| \leq 8\varepsilon^{1/4}\sqrt{\sum_{j\in[m]} V_{s,j}} =: \alpha_s,$$

while $s$th diagonal entry of $\overline{\boldsymbol{\Sigma}}$ i.e., $\mathrm{Var}[\Upsilon_s|J]$ is, by (129) and (118)

$$\mathsf{E}_J\left[\mathrm{Var}[h_s] \mid J\right] \geq \frac{1}{4}\sum_{j\in[m]} V_{s,j} =: \beta_s.$$

Combining the above and applying the Gaussian anticoncentration (5), we get that for any $t_s$,

$$\begin{aligned}
\Pr\left[\mathbb{1}_{\Upsilon_s > t_s} \neq \mathbb{1}_{\overline{\Upsilon}_s > t_s}\right] &\leq \Pr\left[\Upsilon_s \in [t_s - \alpha_s, t_s + \alpha_s]\right] \\
&\leq \alpha_s/\sqrt{\beta_s} \leq O(\varepsilon^{1/4})
\end{aligned} \tag{134}$$

Taking a union bound over all $s \in [\ell]$ yields the lemma. $\qquad\square$

### D.5.2 Main Lemma

Combining Lemma D.8, the analysis in Sec. D.5 culminating in (128), along with Lemmas D.9 and D.10 and the setting of our parameters and choice of small enough $\varepsilon_0$ in (9), we obtain the following main result of this section

**Lemma D.11.** *Let $\boldsymbol{\Upsilon}$ be distributed as $N\left(\boldsymbol{\mu}, \overline{\boldsymbol{\Sigma}}\right)$ (as in Lemma D.10 which is independent of any of the choices of $\mathcal{D}_X$. Then, except with probability $O(\ell^2\sqrt{\varepsilon})$ over the choice of $J$, for any $t_1, \ldots, t_\ell$,*

$$\left| \Pr\left[ \bigwedge_{s=1}^{\ell} (\Upsilon_s > t_s) \right] - \Pr_{\mathcal{D}_X}\left[ \bigwedge_{s=1}^{\ell} (h_s(\mathbf{X}) > t_s) \mid J \right] \right| \leq O(\ell\varepsilon^{1/4}) \tag{135}$$

## E    Generalization Error for Minimizing Unsatisfied Bags of size $2$

The work of [8] showed a generalization error bound for the proportion function $f$ which maps a vector $\mathbf{y}$ of $r$ binary labels in a bag to $\mathbb{R}$. The sufficient property of $f$ used is that $f$ is 1-Lipschitz w.r.t. the infinity norm of $\mathbf{y}$.

To apply their analysis for bounding the generalization error when minimizing the unsatisfied bags of size at most 2 we first observe there are three proportions possible: $0, 0.5$ and $1$. These are the bag labels given by the feature-vector classifier. Let us assume all bags are of size 2 by replicating the feature vectors in the bags of size $1$.

Now consider $f$ which maps (i) the 0-bags to $0$, and (ii) the 0.5-bags and 1-bags to $1$. Clearly, $f$ is 1-Lipshitz w.r.t. to the infinity norm. Applying the results of [8] we get the generalization error bound when classifying between 0-bags and the $\{0.5, 1\}$-bags. Further, by flipping the labels of the feature-vectors we obtain the same generalization error bound when classifying between the 1-bags and the $\{0.5, 0\}$-bags since the negation of a classifier has the same VC-dimension. Additionally we have the constraint that the proportions of the 0-bags, 0.5-bags and 1-bags sum to 1. Combining these arguments we obtain the same (up to a factor of 3) generalization error bound for minimizing the number of unsatisfied bags of size at most 2 by a feature-vector level classifier.