# OpenReview forum: "Learnability of Linear Thresholds from Label Proportions"
_NeurIPS.cc/2021/Conference — NeurIPS 2021 Poster_

### Official Review · Reviewer_myvN · 2021-07-14

**Rating:** 5
**Confidence:** 1

**Summary:**

This paper intends to make a few theoretical contributions for the learning from label proportions framework. In this framework training data arrives in the form of bags that contain several feature vectors and the proportion of positive labels in the bag as ground truth. The authors focus on learning linear threshold functions, while making a restriction to bags of size two.

A few theoretical results are presented. A first theorem gives a lower bound on the number of bags that can be "satisfied" with a polynomial time algorithm -- the authors define a bag as satisfied if the predicted proportion of positive labels equals the real proportion in the bag. A second theorem states that finding a linear threshold function that satisfies all bags is NP-hard in the worst case.

**Ethical Concerns:**

No concerns.

**Limitations And Societal Impact:**

No comments.

**Main Review:**

Frankly speaking, I found it very hard to follow the reasoning of the authors. This is probably due to my limited knowledge of the topic. This is not a paper I voted for, and it probably ended up in my batch because nobody else voted for it. In any case my review should have a low weight in taking a final decision whether this paper should be accepted or not.

I believe that the presentation could be substantially improved. Let me mention the main problems that appeared while reading the paper:
- I was unable to follow the problem setting that is described in Section 1.2. I had to consult the paper "On learning from label proportions" Yu et al. Arxiv 2014 to understand the problem setting. The authors should better describe that a training dataset consists of tuples (B,sigma) where B is a bag of feature vectors, and sigma is the proportion of positive instances in the bag.
-L100: "with the guarantee that there exists an LTF that satisfies all the bags". It is not clear to me why such an LTF always exists. More explanation is needed here. Perhaps a reference or a short proof?
-L103: "when the guaranteed LTF is an OR". The acronym OR is never defined in the paper. I have no idea what it is, but this is very crucial to understand the remainder of the paper. The acronym is already mentioned in the abstract and the introduction, but also there it is not explained what it is.
-L135: What does the acronym WLOG mean?

As a result it is not possible to appreciate the impact of Theorem 1.2, as well as the implications that are discussed in Section 1.4 and 1.5.

Theorem 1.2 looks like a nice result, but a lot of restrictions are needed. The restriction to bags of size two, which is omnipresent in the whole paper, is to my opinion a very restrictive condition, which limits the practical relevance of the results. Moreover, even with this restriction, the result is quite weak, because only half of the bags can be satisfied in polynomial time.

Overall, I think that the presentation of this paper has to be improved before it can be accepted. Yet, as mentioned above, I am not an expert on the topic, and my opinion should have a low weight.

**Time Spent Reviewing:**

3

---

> ### Author Response · Authors · 2021-08-05
> **Response to Reviewer myvN by Paper6429 Authors**
>
> We thank the Reviewer for their efforts and feedback. However, as mentioned by the Reviewer, perhaps due to lack of familiarity with the topic, the paper’s contributions have not been fully conveyed to the Reviewer. We therefore wish to address the specific problems and concerns mentioned by the Reviewer in detail, beginning with the questions on the acronyms used.
>
> *Reviewer*: “ L103: "when the guaranteed LTF is an OR". The acronym OR is never defined in the paper. I have no idea what it is, but this is very crucial to understand the remainder of the paper. The acronym is already mentioned in the abstract and the introduction, but also there it is not explained what it is.”
> *Response*: OR denotes the OR function over boolean variables i.e., the disjunction of boolean variables. It is well known that OR is a special case of Linear Threshold Function (LTF) i.e., one can write any OR over boolean variables as an LTF over those variables. In the abstract, L10 introduces it as “..  the special case of OR over the $d$-dimensional boolean vectors..” , and L65-66 refers to it as “.. special case of the feature vectors being $d$-dimensional boolean and the bags being consistent with an OR..”. While we think our usage of OR is reasonably clear, we will add an explicit definition of OR as the boolean disjunction.
>
>
> *Reviewer*: “L135: What does the acronym WLOG mean?”
> *Response*: WLOG is a standard abbreviation of “without loss of generality”, and is commonly used in mathematical arguments.
>
>
> *Reviewer*: “I was unable to follow the problem setting that is described in Section 1.2. I had to consult the paper "On learning from label proportions" Yu et al. Arxiv 2014 to understand the problem setting. The authors should better describe that a training dataset consists of tuples (B,sigma) where B is a bag of feature vectors, and sigma is the proportion of positive instances in the bag.”
> *Response*: The problem setting in Section 1.2 abstracts out the optimization problem at the core of LLP learning LTFs on bags of size at most 2. We also wish to respectfully point out that lines 36-37 of the paper state:  “..the training data consists of bags (subsets) of feature-vectors along with the proportion of labels in each bag.”.  Nevertheless, we will add an explanation for the abstraction and a formal definition, as suggested by the Reviewer, of the LLP training dataset as part of Section 1.2.
>
>
> *Reviewer*: “L100: "with the guarantee that there exists an LTF that satisfies all the bags". It is not clear to me why such an LTF always exists. More explanation is needed here. Perhaps a reference or a short proof?”
> *Response*: This guarantee flows from the standard computational learning framework where we assume that the observed data is consistent with some (unknown) function from a given classifier class (LTFs in this case). In particular, the Probably Approximately Correct (PAC) [Valiant ’84] model of learning has this assumption and is widely used to study the computational learnability of concept classes. We will add a brief explanation and a reference to PAC learning.
>
> Next we address the following concern.
> *Reviewer*: “Theorem 1.2 looks like a nice result, but a lot of restrictions are needed.”
> *Response*: Theorem 1.2 is a NP-hardness result. Proving that a special/restricted case of a problem is hard also implies that the problem in general is hard. Therefore, these restrictions are not required, and their presence in fact yields a stronger result.
> In the case of Theorem 1.2, we show a hardness result for satisfying an LLP-OR instance with non-monochromatic bags of size $2$ using as hypothesis an arbitrary function of a constant number of LTFs. In this, the concept class is the OR function and so this is a special case of satisfying an LLP-LTF instance with non-monochromatic bags of size $2$. The latter is a special case of LLP-LTF with bags (not necessarily non-monochromatic) of unrestricted sizes. Therefore, the hardness result of Theorem 1.2 holds for satisfying LLP-LTF with unrestricted bag sizes using an arbitrary function of a constant number of LTFs.
>
>
> Finally, we address the following concern.
> *Reviewer*: “The restriction to bags of size two, which is omnipresent in the whole paper, is to my opinion a very restrictive condition, which limits the practical relevance of the results. Moreover, even with this restriction, the result is quite weak, because only half of the bags can be satisfied in polynomial time.”
> *Response*: We wish to emphasize that prior to this work, there was no algorithmic guarantee whatsoever for bag size $> 1$. The main thrust of this work is a deeper understanding of this problem. In particular, we show that its difficulty changes drastically from bag size $= 1$ (i.e., traditional supervised learning) to bag size $\leq 2$ in terms of the stronger techniques used to obtain non-trivial accuracy guarantees and a bound on the best accuracy possible, as evidenced by Theorems 1.1 and 1.2.  As mentioned in the Conclusions and Future Work (Sec. 4) extending the algorithmic results to bag sizes $> 2$ remains an open problem, and possibly requires more sophisticated techniques.
>
>
> We hope that the above clarifications effectively address the concerns of the Reviewer and reinforce the novel contributions of this work. If so, we sincerely request the Reviewer to consider improving their score.

---

### Official Review · Reviewer_w9nx · 2021-07-17

**Rating:** 7
**Confidence:** 2

**Summary:**

This paper studies the learnability of LTFs from label proportions, on bags of size at most two.

**Ethical Concerns:**

No ethical concern was found.

**Limitations And Societal Impact:**

The authors adequately addressed the limitations, and they do not see any negative social impact of this work.

**Main Review:**

Pros:
+ An algorithm is proposed which efficiently produces an LTF that satisfies at least (2/5)-fraction of the bags of an LLP-LTF instance, and (1/2)-fraction of them if all bags are non-monochromatic. Additionally, For LLP-OR over d-dimensional Boolean vectors, the authors improve these factors to (2/5 + Ω(1/d)) and (1/2 + Ω(1/d)) respectively.
+ The authors also proved that  it is NP-hard, given a collection of non-monochromatic bags which are all satisfied 16 by some monotone OR, to compute any function of constantly many LTFs that 17 satisfies (1/2 + $\epsilon$)-fraction of the bags for any constant $\epsilon> 0$. The mathematics are concise.

Cons:
-  It could be better if some experiments can be added to validate the proposed theoretical results.


**Time Spent Reviewing:**

7

---

> ### Author Response · Authors · 2021-08-10
> **Response to Reviewer w9nx by Paper6429 Authors**
>
> We thank the Reviewer for their efforts and encouraging evaluation. We address their concern below.
>
> *Reviewer*: It could be better if some experiments can be added to validate the proposed theoretical results.
> *Response*: We accept the suggestion of the Reviewer and propose to add an experimental evaluation of Algorithm $\mathcal{A}$ (Fig. 1 on pg. 7) which is at the core of the algorithmic contributions of this work. We evaluated the performance on synthetically generated LLP-LTF instances over dimensions $d \in$ {$10, 40, 100$} with number of bags $m\in$ {$50, 100, 200$}. For each $(d, m)$, the algorithm solved $100$ independently generated LLP-LTF instances. For each instance, the SDP in equations (1) and (2) (Fig. 1) was solved and the best $h^*$ was chosen using $100$ independent Gaussian samples ${\bf g}=(g_1,\dots,g_{d+1})$. For each instance, $m_{non}$ and $m_{mon}$ denote the number of non-monchromatic and monochromatic bags (summing up to $m$). The theoretical performance threshold is given by $t =  \frac{m_{non}}{2} + \frac{m_{mon}}{4}$ (Lemma 2.1). We measure $s$ as the number of bags satisfied by $h^*$ and the average and minimum values of $(s/t)$ over the $100$ instances, both of which are quite a bit larger than $1$. This indicates that the algorithm performs much better than than the theoretical guarantee on random instances. The experimental results are presented in the table below.
>
> |   $d$ |   $m$ |   ${\rm avg}\ m_{non}$ |   ${\rm avg}\ m_{mon}$ |   ${\rm avg}\ t$       |   ${\rm avg}\ s$ |   ${\rm avg}\ s/t$ |   $\min\ s/t$ |
> |----:|----:|------------------------:|------------------:|--------------------:|---------------:|------------------:|------------------:|
> |  10 |  50 |                   23.12 |               26.88 |         18.28   |          39.36 |              2.16 |              1.7  |
> |  10 | 100 |                   46.39 |               53.61 |         36.5975 |          85.94 |              2.35 |              2.01 |
> |  10 | 200 |                   92.78 |              107.22 |         73.195  |         182.94 |              2.5  |              2.25 |
> |  40 |  50 |                   24.61 |               25.39 |         18.6525 |          29.51 |              1.58 |              1.35 |
> |  40 | 100 |                   50    |               50    |         37.5    |          54.27 |              1.45 |              1.31 |
> |  40 | 200 |                   99.33 |              100.67 |         74.8325 |         105.07 |              1.4  |              1.28 |
> | 100 |  50 |                   25.11 |               24.89 |         18.7775 |          28.53 |              1.52 |              1.39 |
> | 100 | 100 |                   49.08 |               50.92 |         37.27   |          51.48 |              1.38 |              1.27 |
> | 100 | 200 |                   99.23 |              100.77 |         74.8075 |          96.29 |              1.29 |              1.18 |

---

### Official Review · Reviewer_pByS · 2021-07-18

**Rating:** 7
**Confidence:** 2

**Summary:**

This paper studies proper learning of linear threshold functions (LLFs) in the learning from label proportions model.  There are two main contributions.  The first is a polynomial time algorithm that, given a collection of bags each of size at most two consistent with some (unknown) LLF, identifies an LLF that satisfies at least $2/5$ of the bags.  The second main contribution, which complements the first, shows that it is NP-hard to compute, given a collection of bags all satisfied by some monotone OR, to compute any function of constantly many LLFs that satisfies $(1/2+\epsilon)$ of the bags for any positive $\epsilon$.


**Ethical Concerns:**

None.

**Limitations And Societal Impact:**

Yes, I think so.

**Main Review:**

Unfortunately, the proof techniques of Theorem 1.2 lie outside my area of expertise, so despite my best reading I am not absolutely sure whether the proof is overall correct.  It certainly looks credible enough though.

Minor comments/suggestions:

- Page 5, line 195: feature -> features.  Also, double occurrence of "supported".

- Page 8, line 315: tranformed -> transformed

- Page 8, line 317: one modify -> one modifies

- Page 8, line 321: Should it be Appendix A rather than B?  (It seems the observations in the two bullet points appearing 5 lines after Fact A.1 were being used here.)

- Page 9, line 343: "it [is] NP-Hard..."

- Page 11, reference [29]: Title of the paper seems to be "On learning [from] label proportions".

- Supplementary, page 1, paragraph after Fact A.1: Should it be $\langle z_1,z_2\rangle$ rather than $\langle z_1,z_1\rangle$ ?

**Time Spent Reviewing:**

16

---

> ### Author Response · Authors · 2021-08-09
> **Response to Reviewer pByS by Paper6429 Authors**
>
> We are grateful for the Reviewer's efforts and encouraging feedback. We thank the Reviewer for their comments and suggestions and our response is given below.
>
> *Reviewer*: Page 5, line 195: feature -> features. Also, double occurrence of "supported".
> *Response*: On page 5, line 195, instead of “feature”, it should be in fact be “feature-vectors”. We will correct this and the double occurrence of “supported”.
>
> *Reviewer*: Page 8, line 315: tranformed -> transformed
> Page 8, line 317: one modify -> one modifies
> Page 9, line 343: "it [is] NP-Hard..."
> *Response*: We will correct the above typos.
>
> *Reviewer*: Page 8, line 321: Should it be Appendix A rather than B? (It seems the observations in the two bullet points appearing 5 lines after Fact A.1 were being used here.)
> *Response*: Yes, it should be Appendix A on Line 321 rather than Appendix B. We will correct it.
>
> *Reviewer*: Page 11, reference [29]: Title of the paper seems to be "On learning [from] label proportions".
> *Response*: Yes, we will correct the title of reference [29] to “On learning from label proportions”.
>
> *Reviewer*: Supplementary, page 1, paragraph after Fact A.1: Should it be $\langle z_1, z_2\rangle$ rather than $\langle z_1, z_1\rangle$ ?
> *Response*:  It should indeed be  $\langle z_1, z_2\rangle$ rather than $\langle z_1, z_1\rangle$. All occurrences of this typo in the paragraph after Fact A.1 will be fixed.

---

> > ### Comment · Reviewer_pByS · 2021-08-30
> > **Response to Response to Reviewer pByS**
> >
> > Thank you very much for the additional explanations and for carrying out an experimental evaluation of Algorithm $\mathcal{A}$.  I still think this paper should be accepted (though I am not absolutely sure that the proofs are correct due to lack of expertise), and will keep my present score.

---

### Official Review · Reviewer_GBZR · 2021-09-01

**Rating:** 7
**Confidence:** 4

**Summary:**

This work considers the problem of learning a linear threshold function (LTF) from label proportions. In particular they consider the special case where both single (as in the standard PAC learning setting) labeled (as 0 or 1) examples are observed or pairs of points $(x_1, x_2)$ together with their average label (which can be 0, 1, or 1/2) are observed.  They give two main results: a polynomial time algorithm that finds an LTF that satisfies at least 2/5 of the labeled examples (pairs and singletons together) and a lower bound that shows that finding an LTF that satisfies more than 1/2 of the labeled examples is NP-Hard.  For the special case where only mixed pairs, i.e., with label 1/2, are observed, the provided algorithm satisfies 1/2 of the labeled examples matching the lower bound (which also assumes only mixed pairs).

**Limitations And Societal Impact:**

Does not apply.

**Main Review:**

Learning from label proportions (LLP) is an interesting generalization of PAC learning that has attracted attention both from the applied and the theoretical learning communities.  This work considers the fundamental problem of learning linear threshold functions in the LLP setting.  It is interesting that even without noise it is NP-Hard to find a halfspace in the distribution free LLP setting (with bags of size 2) that satisfies more than 1/2 of the bags.  In the standard noiseless PAC setting (corresponding to bags of size 1) the same problem admits a polynomial LP algorithm that finds a halfspace satisfying all the bags.  Therefore, this work establishes a provable separation between PAC and LLP learning of halfspaces.  Taken together the upper and lower bounds of this work provide a nice (and in some interesting cases tight) picture of learning LTFs in the distribution free LLP setting.  Moreover, the results of this work (mainly the reduction) are technically interesting and non-trivial.  The paper is also well-written and the provided sketches of the proofs were very useful.
Overall, this work provides interesting results on a fundamental learning problem in the LLP setting (which is well motivated) and I believe that it meets the standards of NeurIPS.

**Time Spent Reviewing:**

8

---

### Official Review · Reviewer_Cri6 · 2021-09-02

**Rating:** 7
**Confidence:** 4

**Summary:**

This work studies the problem of learning from label proportions (LLP), that is given bags of vectors and given the average of the positive labels in each bag, we need to find an LTF that satisfies most of the bags. The authors study the case where the bags contain at most 2 vectors. They study two cases, one where the vectors are taken from $\mathcal X\subseteq \mathbb{R}^d$ and they are searching for an LTF; in the other case the vectors are taken from the $d$-dimensional hypercube $\{0,1\}^d$, and they are looking for an OR-LTF. In the first case, they provide an algorithm that satisfies $2/5$ of the bags and if all the bags are non-monochromatic (contain vectors with different labels) then the algorithm satisfies $1/2$ of the bags, for the other case they provide an algorithm that satisfies $(2/5 + \gamma_0/d)$ bags and if all the bags are non-monochromatic it satisfies $(1/2+\gamma_0/d)$ of the bags. Furthermore, for the last case, the authors provide an NP-hardness reduction to the Label Covering problem that shows that their result is qualitatively tight and that in general, we cannot hope for an algorithm that satisfies all the bags.

**Limitations And Societal Impact:**

This work does not seem to have any negative societal impact.

**Main Review:**

First, this paper is very well-written and organized, with minor typos. This work shows that learning with LLP is difficult without any assumptions which does not hold in the standard (noiseless) supervised PAC learning (bags of size one); where the problem admits a polynomial-time algorithm that satisfies all the bags. In addition, I find this model very interesting as this model can have applications in privacy (i.e., when you do not want to reveal the labels of individual people) and a natural generalization of supervised PAC learning. Their algorithm uses the structural property that the $w\cdot x_1\times w\cdot x_2$ is negative when the bag is non-monochromatic and then using this property they formulate SDP along with standard rounding technique.
Their lower bound (which is an NP-hardness reduction t) is technically challenging, which is the main contribution of this work and shows that their algorithm for the case of the non-monochromatic bags is tight, even if we allow the hypothesis to consists a union of LTFs.

The weakness of this work is that the structural property, which the algorithm heavily relies on, does not seem to extend on bags with sizes greater than two but still, this upper bound complements the lower bound for the tightness result.

Overall, I recommend this work for acceptance.

**Time Spent Reviewing:**

5

---

### Decision · Program_Chairs · 2021-09-27

**Decision:**

Accept (Poster)

**Comment:**

This paper studies the problem of learning linear threshold functions (LTFs) in the following model, termed "learning from label proportions"(LLP). (Recall that in the standard statistical learning setting, one observes i.i.d. labeled examples (x, y).) In the LLP model considered in this work, in addition to labeled examples (x, y), one observes *pairs* of examples together with their *average* label. The main results of the paper are: (1) an efficient proper learner satisfying at least 2/5 fraction of the pairs, and (2) an NP-hardness of proper learning result ruling out approximation better than 1/2. For a special case of the problem, the authors show that 1/2 is in fact the optimal approximation ratio for efficient proper learning. The algorithmic result uses SDP, while the hardness result uses a PCP-style reduction from label cover. Overall, the reviewers agreed that this is a technically worthy and well-written theory paper that should be accepted to NeurIPS.